# Endogenous formaldehyde scavenges cellular glutathione resulting in redox disruption and cytotoxicity

Carla Umansky [1,6], Agustín E. Morellato [1,6], Matthias Rieckher[2,6], Marco A. Scheidegger[1], Manuela R. Martinefski [3], Gabriela A. Fernández[3], Oleg Pak[4], Ksenia Kolesnikova[2], Hernán Reingruber[1], Mariela Bollini[3], Gerry P. Crossan [5], Natascha Sommer [4], María Eugenia Monge [3], Björn Schumacher [2] & Lucas B. Pontel [1✉]

Formaldehyde (FA) is a ubiquitous endogenous and environmental metabolite that is thought to exert cytotoxicity through DNA and DNA-protein crosslinking, likely contributing to the onset of the human DNA repair condition Fanconi Anaemia. Mutations in the genes coding for FA detoxifying enzymes underlie a human inherited bone marrow failure syndrome (IBMFS), even in the presence of functional DNA repair, raising the question of whether FA causes relevant cellular damage beyond genotoxicity. Here, we report that FA triggers cellular redox imbalance in human cells and in *Caenorhabditis elegans*. Mechanistically, FA reacts with the redox-active thiol group of glutathione (GSH), altering the GSH:GSSG ratio and causing oxidative stress. FA cytotoxicity is prevented by the enzyme alcohol dehydrogenase 5 (ADH5/GSNOR), which metabolizes FA-GSH products, lastly yielding reduced GSH. Furthermore, we show that GSH synthesis protects human cells from FA, indicating an active role of GSH in preventing FA toxicity. These findings might be relevant for patients carrying mutations in FA-detoxification systems and could suggest therapeutic benefits from thiol-rich antioxidants like N-acetyl-L-cysteine.

[1] Instituto de Investigación en Biomedicina de Buenos Aires (IBioBA), CONICET - Partner Institute of the Max Planck Society, C1425FQD Buenos Aires, Argentina. [2] Institute for Genome Stability in Ageing and Disease, Medical Faculty, University of Cologne, and Cologne Excellence Cluster for Cellular Stress Responses in Aging-Associated Diseases (CECAD), and Center for Molecular Medicine Cologne (CMMC), 50931 Cologne, Germany. [3] Centro de Investigaciones en Bionanociencias (CIBION), Consejo Nacional de Investigaciones Científicas y Técnicas (CONICET), C1425FQD Buenos Aires, Argentina. [4] Justus-Liebig University, Excellence Cluster Cardio-Pulmonary Institute (CPI), Universities of Giessen and Marburg Lung Center (UGMLC), Member of the German Center for Lung Research (DZL), Giessen, Germany. [5] MRC Laboratory of Molecular Biology, Cambridge Biomedical Campus, Francis Crick Avenue, Cambridge CB2 0QH, UK . [7] These authors contributed equally: Carla Umansky, Agustín E. Morellato, Matthias Rieckher. ✉email: lpontel@ibioba-mpsp-conicet.gov.ar

F A is a potent genotoxin classified by the World Health Organization (WHO) as a human carcinogen[1]. Physiologically, FA is constantly generated through cellular metabolism, i.e., histone and DNA demethylation reactions or the one-carbon cycle, but it can also arise from the diet and it is ubiquitously found in the environment[2–5]. Indeed, this aldehyde is more abundant in the human body than previously thought; different works have reported FA quantification in healthy human blood samples with concentrations around 50 μmol L$^{-1}$ [6, 7]. Endogenous FA has been suggested as a causative agent for the DNA repair-deficient condition Fanconi Anaemia[8–10]. This disease is caused by mutations in genes coding for factors that conform the Fanconi Anaemia DNA repair pathway. When this DNA repair pathway is not functional, cells are more sensitive to compounds that can damage the DNA such as DNA interstrand-crosslinking (ICL) agents and simple aldehydes[11]. Mice with inactivating mutations in the gene coding for the enzyme alcohol dehydrogenase 5 (Adh5) and in the gene coding for the factor Fanconi Anaemia Complementation Group D2 (Fancd2) accumulate FA-DNA adducts and present hematopoietic stem cell attrition, which correlates with genome instability in the hematopoietic compartment[9]. Moreover, these mice develop leukaemia and liver cancer and present widespread karyomegaly[9], overall indicating that endogenous FA can drive cancer initiation and Fanconi Anaemia phenotypes through causing genome instability. Additionally, when DNA repair is impaired as a consequence of mutations in the gene coding for the tumour suppressor Breast Cancer 2 (BRCA2), environmental FA can drive genome instability and contribute with cancer initiation[12]. Furthermore, human Fanconi Anaemia patients carrying a mutation in the gene coding for the acetaldehyde/FA catabolizing enzyme aldehyde dehydrogenase 2 (ALDH2) present accelerated progression of bone marrow failure (BMF)[13]. In the absence of mutations in DNA repair-coding genes, FA can also underly a variant of a human inherited bone marrow failure syndrome (IBMFS) caused by mutations in both ALDH2 and ADH5[10, 14]. This syndrome has been termed IBMFS/AMeD (Aplastic anaemia, Mental retardation, and Dwarfism) and it is manifested by early myelodysplasia, mental and growth retardation, and dwarfism[10, 14]. The simultaneous inactivation of Adh5 and Aldh2 in mice results in FA-driven postnatal lethality and hematopoietic phenotypes, recapitulating some aspects of the human IBMFS/AMeD[10], and thus defining a human syndrome originated in a failure in FA metabolism.

The phenotypes observed in patients with mutations in the main FA clearance systems but proficient for DNA repair suggest that FA can cause cellular damage beyond genotoxicity[4]. In vitro, it was reported that the spontaneous electrophilic attack of the FA carbonyl group to the thiol-group of GSH leads to the formation of the covalent product S-hydroxymethyl-GSH (HSMGSH)[15]. This reaction might be strongly favoured inside cells, where GSH levels are in the millimolar range[16]. Accordingly, ADH5 metabolizes HSMGSH, lastly yielding formate, which is directed to the one-carbon cycle for nucleotide synthesis[3]. The reaction between FA and GSH might alter the ratio between GSH and the oxidized GSH disulfide form (GSSG)-GSH:GSSG, which has recently been reported to determine the GSH redox buffer capability[17]. Interestingly, mild to severe alterations in GSH homeostasis have been reported in multiple pathologies such as haemolytic anaemia, diabetes, liver diseases, cystic fibrosis, neurodegeneration, and cancer[18–21]. GSH not only neutralizes reactive oxygen species (ROS), but can also promote chemoresistance by forming GSH-xenobiotic conjugates that are pumped out of the cell via multiple resistance-associated protein transporters (MRP)[22]. To replenish intracellular GSH, cells synthesize GSH in a two-step metabolic pathway centred on the rate-limiting enzyme glutamate cysteine

ligase (GCL), which is composed of a catalytic (GCLC) and a regulatory or modifier (GCLM) subunit, and the GSH synthetase (GS)[22]. Hence, cells might also need to maintain the balance between GSH and GSSG to limit free FA and to prevent redox disruption.

We report here that in cells FA reacts with the redox-active thiol group present in GSH, leading to an imbalance of the ratio between GSH and GSSG and to oxidative stress. Moreover, FA cytotoxicity can be prevented by supplying the GSH-precursors N-acetyl-L-cysteine (NAC) or GSH-monoethyl ester (GSH-MEE), and it is exacerbated by inactivating cellular GSH synthesis. Hence, our data also support a previously unrecognized function of GSH in the protection against FA toxicity, and an evolutionary conserved mechanism that maintains GSH:GSSG balance by salvaging reduced GSH from FA-GSH products. These data might have wide implications for the biology underlying the FA-driven IBMFS/AMeD syndrome, for Fanconi Anaemia patients, for BRCA2-mutations carriers, and for cancer cells that would have to cope with blood FA[9, 12].

## Results

**ADH5 limits FA cytotoxicity even in the presence of DNA repair.** It is still unresolved how FA compromises cellular function, especially in cells proficient for DNA repair, such as those from patients harbouring mutations in the genes coding for FA-metabolizing enzymes. To gain insight into the cellular response to FA, we obtained human pre-B lymphoblastic leukemic Nalm6 cells deficient in the FA-metabolizing enzyme ADH5 (ΔADH5) and explored the tolerance to FA, the accumulation of the DNA double-strand break (DSB) marker γ-H2AX, and the phosphorylation of the tumour suppressor p53. These phenotypes were assessed in comparison to isogenic cells deficient in the ICL DNA-repair pathway Fanconi Anaemia (ΔFANCB) (Fig. 1a–c and Supplementary Fig. 1a). Despite that ΔADH5 and ΔFANCB cells were significantly sensitive to physiological levels of FA, γ-H2AX was detected in the presence of FA in ΔFANCB but not in ΔADH5 cells (Fig. 1b, c), indicating that FA also impairs cell viability when DNA repair is intact, but without triggering the DNA damage marker γ-H2AX. In contrast, p-p53 and γ-H2AX were induced in ΔADH5 and ΔFANCB cells by the ICL-DNA damaging drug cisplatin (CPt) (Fig. 1b, c). These observations are in agreement with previous findings reporting that γ-H2AX was induced by FA in ΔFANCD2 but not in wild type (WT) or ΔADH5 chicken DT40 cells[11]. To further corroborate this observation, we inactivated the ADH5 gene in the non-blood derived human colorectal carcinoma cell line HCT116 by CRISPR/Cas9 (Supplementary Fig. 1b). HCT116 ADH5-deficient cells were sensitive to FA and were not able to form tumour-spheroids when exposed to FA (Fig. 1d, e). Moreover, in the presence of FA, ΔADH5 cells showed the early apoptosis marker phosphatidylserine (measured by Annexin V), a blockage of the cell cycle at G2/M phase, and sub-G1 DNA accumulation (Supplementary Fig. 1c, d). Concordantly with the observation in pre-B leukemic Nalm6 cells, H2AX was not phosphorylated in WT or ΔADH5 HCT116 cells upon FA exposure (Fig. 1f and Supplementary Fig. 2a). In contrast to FA treatment, exposure to the DNA-damaging drugs Cpt, hydroxyurea (HU) or mitomycin C (MMC) resulted in a profound induction of γ-H2AX (Fig. 1f and Supplementary Fig. 2a). Interestingly, despite neither WT nor ΔADH5 cells presented a significant phosphorylation of the upstream kinases CHK1 and CHK2 in response to FA, we detected a mild phosphorylation of p53 (Ser 15) in ΔADH5 but not in WT HCT116 cells (Fig. 1f and Supplementary Fig. 2b, c). p53 is one of the main factors that orchestrates apoptosis in response to DNA damage, and it controls the fate of hematopoietic cells deficient in both the Fanconi Anaemia ICL DNA repair pathway and the aldehyde-metabolizing enzyme Aldh2 in mice[13]. Therefore, to address

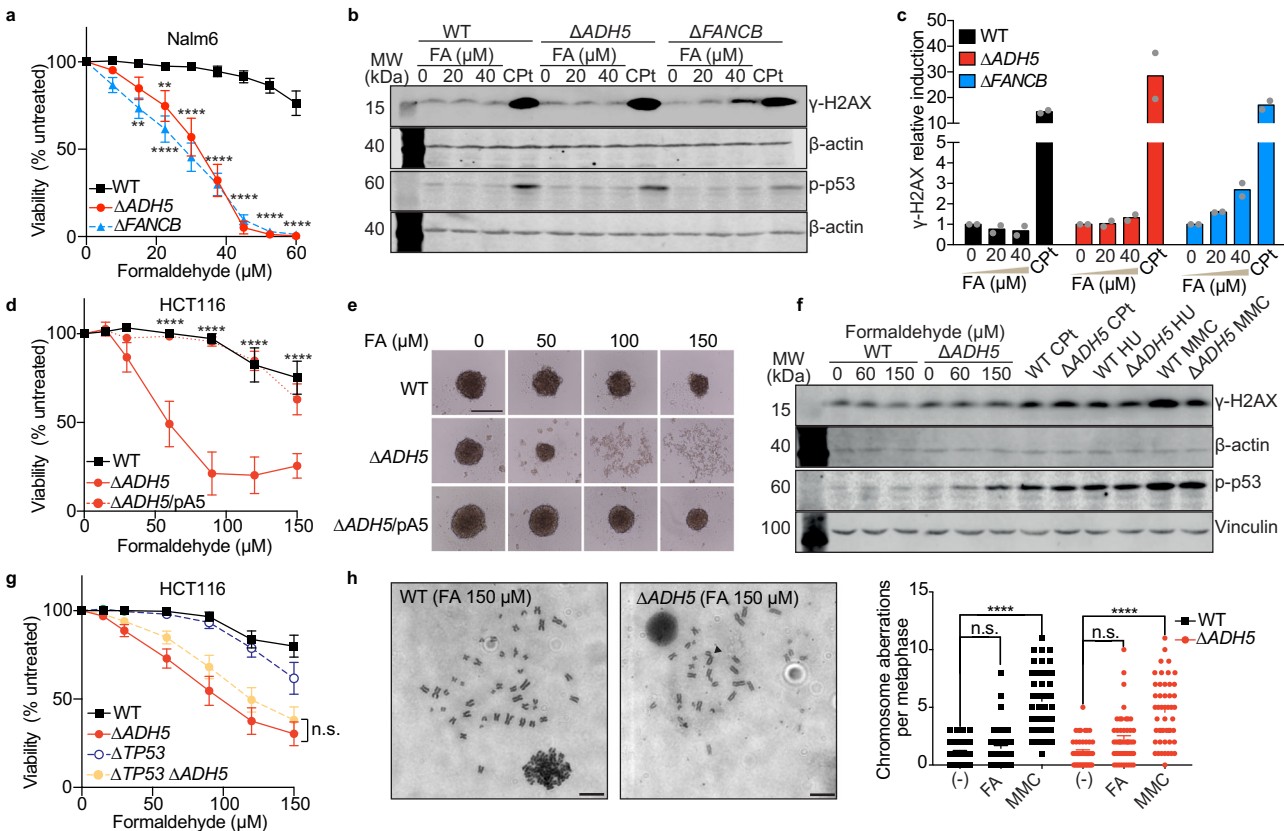

**Fig. 1 ADH5 prevents formaldehyde toxicity in human cancer cells. a** Viability assay for Nalm6 cells exposed to formaldehyde (FA) (mean ± SEM, $n = 6$, one-way ANOVA for multiple comparison using a Tukey-corrected test between $\Delta ADH5$ or $\Delta FANCB$ and Wild type (WT) cell lines, **$P = 0.0011$ (WT vs $\Delta FANCB$ 15 μmol L$^{-1}$ FA), **$P = 0.004$ (WT vs $\Delta ADH5$ 22.5 μmol L$^{-1}$ FA), ****$P < 0.0001$. **b** Western blots depicting γ-H2AX and p-p53 signals after 48 h of exposure to the indicated concentrations of FA, or after 24 h to 8 μmol L$^{-1}$ cisplatin (CPt) in Nalm6 cells. **c** Quantification of γ-H2AX western blots ($n = 2$, mean relative to same cell line untreated). **d** Viability assay for HCT116 cells. Data are presented as mean (% of untreated) ± SEM, $n = 6$, two-way ANOVA for multiple comparison using a Tukey-corrected test between WT and $\Delta ADH5$ HCT116 cells, ****$P < 0.0001$. **e** Representative images of 3D-sphere formation for WT, $\Delta ADH5$ and $\Delta ADH5/p$ADH5 (pA5) HCT116 cells (scale bar 0.5 mm). μM refers to μmol L$^{-1}$ and mM refers to mmol L$^{-1}$. **f** Western blots showing the induction of p-p53 and γ-H2AX in HCT116 cell lines (48 h for FA, 24 h for Cpt (4 μmol L$^{-1}$), hydroxyurea (HU, 1 mmol L$^{-1}$) and mitomycin C (MMC, 1.5 μmol L$^{-1}$). **g** Viability assay in the indicated HCT116 cell lines in the presence of FA (% of untreated, $n = 8$, mean ± SEM, n.s. not significant between $\Delta ADH5$ and $\Delta TP53 \Delta ADH5$, two-way ANOVA using a Tukey-corrected for multiple comparisons). **h** Representative images of metaphases obtained from WT and $\Delta ADH5$ HCT116 cells exposed to 150 μmol L$^{-1}$ FA (scale bar 1 μm) and the quantification (MMC (1.5 μmol L$^{-1}$) or FA (150 μmol L$^{-1}$). Data are presented as mean ± SEM, $n = 49$, one-way ANOVA, Tukey´s multiple comparison test, ****$P < 0.0001$, n.s.: not significant).

whether p53 is involved in controlling the cellular tolerance to FA, we inactivated the gene coding for this factor (TP53) and the one coding for ADH5 in HCT116 cells (Supplementary Fig. 2d, e). In the presence of functional DNA repair, TP53 inactivation did not rescue the toxicity inflicted by FA (Fig. 1g), supporting that p53 is not the main factor driving FA-induced cell death. To further confirm that a 48 h exposure to micromolar levels of FA is not lethally genotoxic for cells proficient in DNA repair, we addressed genome instability by direct visualization of single chromosome damage in HCT116 cells (Fig. 1h and Supplementary Fig. 2f). Concordantly, we found that most of the metaphases in WT as well as in $\Delta ADH5$ cells were normal and only few of them presented chromosome damage. In stark contrast, significant chromosome damage was evident upon treatment with the DNA crosslinking agent MMC, thus suggesting that when DNA repair is functional, FA might be causing cell death by damaging other cellular components rather than DNA.

ADH5 was reported to operate preventing FA toxicity in bacteria, mice, and humans, suggesting a widely conserved function of this enzyme[8]. To further support the evolutionary role of ADH5 in preventing FA toxicity, we carried out a phylogenetic analysis in eukaryotes looking for genes coding for ADH5-like proteins (Fig. 2a). In the metazoan model *Caenorhabditis elegans*,

the uncharacterized gene H24K24.3 codes for the ortholog of the human ADH5 enzyme. Transgenic expression of H24K24.3 (ADH-5) fused with GFP under the control of the endogenous *adh-5* promoter (Ex[$p_{adh-5}$ADH-5::GFP; $p_{myo-2}$tdTomato]) presented a ubiquitous cytoplasmic expression in larvae and in the adult nematode (Fig. 2b). To assess whether H24K24.3 participates in the prevention of FA toxicity in worms, we generated a null mutant via CRISPR/Cas9 by introducing multiple stop codons in all three reading frames[23]. Animals lacking H24K24.3 showed an extreme hypersensitivity to FA (Fig. 2c), indicating that H24K24.3 is the ortholog of *ADH5* in *C. elegans*. We thus refer to H24K24.3 from now on as *adh-5*. Survival of *adh-5(sbj21)* mutants was restored upon transgenic expression of ADH-5::GFP in adult worms (Supplementary Fig. 2g). Altogether, these data indicate that ADH5 is a widely conserved enzyme dedicated to prevent FA cytotoxicity even in the presence of operating DNA repair mechanisms.

**FA causes oxidative stress.** Despite not detecting DNA damage, FA was evidently cytotoxic for cells lacking ADH5 (Fig. 1a, d). We reasoned that the strong avidity of the FA-carbonyl group

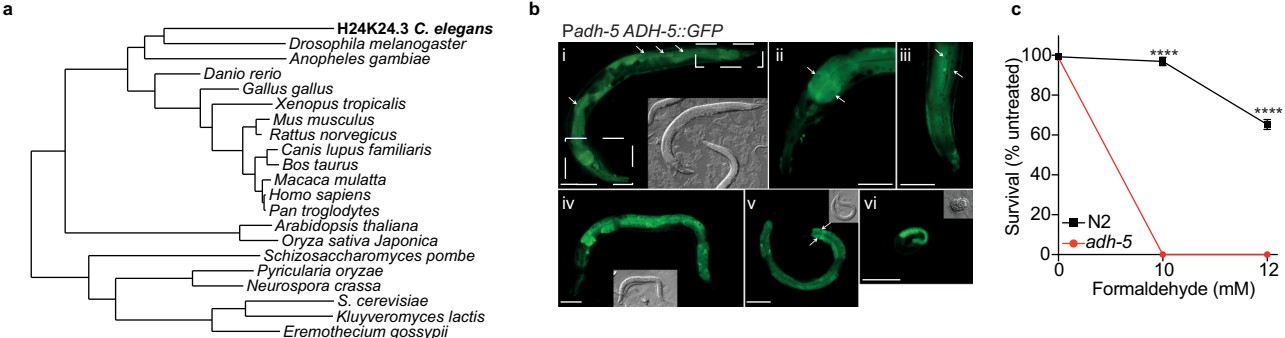

**Fig. 2 H24K24.3 is the *C. elegans* ortholog gene of human *ADH5*. a** Phylogenetic analysis of *ADH5*-homologue genes in eukaryote highlighting the ortholog gene (gi_71997431/H24K24.3) found in *C. elegans*. **b** Various developmental stages of a *C. elegans* transgenic line expressing $Ex[p_{adh-5}$ADH-5::GFP; $p_{myo-2}$tdTomato]. White arrows point to nuclei. Scale bars = 50 μm. Grey inlays show the corresponding DIC image. i Adult stage, boxes refer to (ii and iii). ii Head area with focus on the intestine of the adult depicted in (i). iii Tail area with focus on the cuticle of the animal shown in (i). (iv) L4 stage. (v) L3 stage. (vi) Late embryo. **c** Survival of L1-staged WT (N2) and *adh-5* mutant *C. elegans* upon exposure to the indicated formaldehyde (FA) concentrations measured directly after the treatment (data are presented as mean ± SEM, *n* = 3, one-way ANOVA, Tukey´s multiple comparison test, ****$P < 0.0001$).

toward electron-rich thiol groups might affect the antioxidant GSH. The reaction between FA and the thiol group in GSH would block the redox capability of GSH, impairing its redox function. Moreover, the abundance of GSH (1–10 mmol L$^{-1}$) might favour the spontaneous reaction between GSH and FA, which, if not limited, could alter the GSH:GSSG ratio leading to oxidative stress. We, therefore, measured the level of oxidant species by quantifying the oxidation of the probe 2′,7′-dichlorodihydro-fluorescein diacetate (H2DCFDA). In the cells, this probe is converted into dichlorodihydrofluorescein (H2DCF) that in the presence of oxidant species is oxidized into fluorescent dichlorofluorescein (DCF). Interestingly, FA induced a significant oxidation of H2DCF to DCF in Δ*ADH5* cells (Fig. 3a, b) that could be reverted by expressing ADH5 in trans. As positive control, we exposed cells to the GSH-synthesis inhibitor L-buthionine-sulfoximine (L-BSO), which was shown to induce oxidative stress but not cytotoxicity in HCT116 cells[24, 25]. Accordingly, in Δ*ADH5* cells, FA was also able to induce the oxidation of the dye dihydroethidium (DHE), which reports the presence of superoxide and non-specific oxidant species (Fig. 3c). This induction was comparable to the one observed with anti-mycin A, a drug that blocks the complex III in the mitochondrial electron chain transport causing an increase in superoxide[26] (Fig. 3c).

In order to test the induction of oxidative stress by FA more thoroughly, we incorporated the genetically-encoded cytosolic sensor roGFP. This sensor reports on the GSH redox potential ($E_{GSH}$) established by the ratio GSH:GSSG through a redox relay involving cysteines in cellular glutaredoxins, as an indirect measurement of ROS[27]. Exposure to FA induced a population of cells in which the sensor is oxidized in the absence of ADH5 (Fig. 3d, e). Importantly, iodoacetamide (IAA, a known thiol-reactive compound) was not able to oxidize roGFP when cells were exposed to non-lethal or subtoxic concentrations of this compound, indicating that an electrophilic attack on roGFP is not sufficient to cause a spectral change consistent with its oxidation (Supplementary Fig. 3a–c). To conclusively show that FA can induce oxidative stress and ROS, we performed Electron Spin Resonance (ESR) spectroscopy using the spin probe CMH (1-hydroxy-3-methoxycarbonyl-2,2,5,5-tetramethylpyrrolidine). This probe detects superoxide ion, peroxyl radical, peroxynitrite and nitrogen dioxide[28]. Interestingly, in the absence of ADH5, FA induces a significant accumulation of ROS at non-toxic FA concentrations, which is further exacerbated by blocking GSH synthesis (Fig. 3f and Supplementary Fig. 3d). We next assessed

the chemical probe pentafluorobenzenesulfonyl fluorescein (PBSF) that was significantly induced in Δ*ADH5* HCT116 cells exposed to FA (Supplementary Fig. 3e). This probe fluoresces at $\lambda_{em}$ = 530 nm upon peroxidation of a sulfonyl bond, indicating the presence of cellular peroxides[29]. In *C. elegans*, the peroxide sensor HyPer in *C. elegans*[30] was ratiometrically induced by FA (Supplementary Fig. 3f). In order to further support these findings, we addressed the activation of the gene coding for the glutathione-S-transferase gst-4 (controlled by SKiNhead1 (SKN-1), the Nuclear factor erythroid 2-related factor 2 (NRF2) ortholog) in adult *C. elegans* (Fig. 3g, h). P*gst-4*::GFP presented a significant induction by FA in control worms, further exacerbated by silencing *adh-5*. NAC was able to significantly restrict the FA-dependent P*gst-4*::GFP induction (Fig. 3h). Altogether these data indicate that FA induces oxidative stress in the absence of ADH5 and triggers a SKN-1 dependent response in *C. elegans*.

**Thiol-rich antioxidants prevent FA cytotoxicity.** To address the causal contribution of FA-induced oxidative stress and GSH imbalance to cell death, we set out to test whether FA toxicity could be rescued by the antioxidants NAC, GSH-MEE, or Trolox (water-soluble vitamin E). The death phenotype and the 3D-sphere formation defect observed in HCT116 cells could be almost fully prevented by incubating with NAC or GSH-MEE, indicating that an increase in free-thiols can limit FA cytotoxicity (Fig. 4a, b and Supplementary Fig. 4a, b). FA may react with thiols present in the culture medium, limiting the amount of FA available to enter cells. To reduce this effect, we pre-treated HCT116 cells with NAC for 24 h, and then replaced the medium for NAC-free medium containing FA. This experimental design revealed a significant, NAC dose-dependent prevention of FA toxicity (Fig. 4a). In contrast, Trolox, a non-thiol drug, was unable to revert FA cytotoxicity (Supplementary Fig. 4a, b), in agreement with early results showing that Trolox is cytotoxic in colorectal cancer cells, despite of reducing intracellular H2O2 level[31]. More recently, it was shown that NAC but not Trolox can reverse cell death induced by the combination of L-BSO and MI-2 or Eeyarestatin I (EERI), which are inhibitors of deubiquitinases (DUB)[32]. Remarkably, both GSH-MEE and NAC led to an overgrowth of WT 3D spheres (Fig. 4b, c and Supplementary Fig. 4b). To further interrogate the suppressive effect observed with the antioxidant NAC, we combined it with L-BSO. Blocking GSH synthesis limited the overgrowth phenotype observed in 3D-

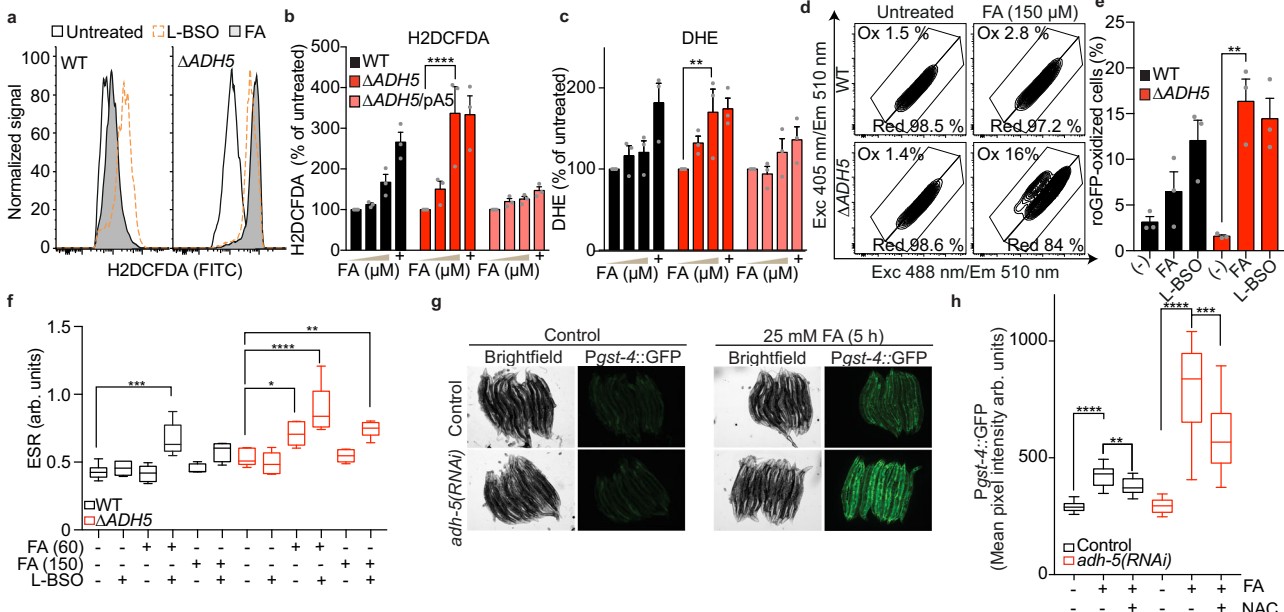

**Fig. 3 Formaldehyde induces oxidative stress.** Oxidative stress determination by 2′,7′-dichlorodihydrofluorescein diacetate (H2DCFDA) (**a, b**) and dihydroethidium (DHE) (**c**) in Wild type (WT), ΔADH5, and complemented ΔADH5 (ΔADH5/pA5) HCT116 cells upon 48 h exposure to 0, 60 and 150 μmol L$^{-1}$ formaldehyde (FA). Data are presented as the % untreated mean ± SEM, $n = 3$, one-way ANOVA, Tukey´s multiple comparison test, ****$P < 0.0001$ (**b**), **$P = 0.0017$ (**c**). μM refers to μmol L$^{-1}$, mM refers to mmol L$^{-1}$. The positive controls are 100 μmol L$^{-1}$ L-buthionine sulfoximine (L-BSO) (48 h for H2DCFDA) and 10 μmol L$^{-1}$ antimycin A (30 min for DHE). **d** Flow cytometry representative plots obtained from WT and ΔADH5 HCT116 cells harbouring the cytoplasmic-roGFP reporter and exposed to 150 μmol L$^{-1}$ FA for 48 h. **e** Quantification of experiments shown in (**d**) ($n = 3$, mean ± SEM, one-way ANOVA, Tukey´s multiple comparison test, **$P = 0.0013$). Positive control: The signal obtained incubating cells for 48 h with 100 μmol L$^{-1}$ L-BSO. **f** Data showing Electron spin resonance (ESR) spectroscopy of reactive oxygen species (ROS) by the spin probe CMH (1-hydroxy-3-methoxycarbonyl-2,2,5,5-tetramethylpyrrolidine). Measures are from independent biological replicates exposed to the conditions stated in the figure for 48 h ($n = 6$ except for ΔADH5 (−), ΔADH5 + 150 μmol L$^{-1}$ FA, ΔADH5 + 60 μmol L$^{-1}$ FA + L-BSO, WT + 150 μmol L$^{-1}$ FA and WT 60 μmol L$^{-1}$ FA + L-BSO with $n = 5$), mean ± SEM, one-way ANOVA, Tukey´s multiple comparison test. WT: ***$P = 0.0008$; ΔADH5: *$P = 0.0485$ ((−) vs FA (60)), **$P = 0.0065$ ((−) vs FA (60) + L-BSO), ****$P < 0.0001$. **g** Representative images of transgenic *C. elegans* expressing the transcriptional reporter P*gst-4*::GFP upon 5 h exposure to 0 and 25 mmol L$^{-1}$ FA. The *adh-5* plot refers to same strain carrying P*gst-4*::GFP, in which *adh-5* expression was silenced by an specific RNAi. **h** Boxplot with Tukey's whiskers (1.5 inter-quartile distance, box centre at median, hinges at quartiles Q1 and Q3) depicting the quantification of mean pixel intensities of animals expressing P*gst-4*::GFP ($n = 25$, two-tailed Mann Whitney test ****$P < 0.0001$, **$P = 0.0019$, ***$P = 0.0002$). In the indicated groups, animals were co-incubated with 10 mmol L$^{-1}$ N-acetyl-L-cysteine (NAC) and 25 mM FA for 5 h.

spheres exposed to NAC (Fig. 4b, c). Moreover, GSH synthesis inhibition partially reduced the NAC-rescue of 3D-sphere formation in ΔADH5 cells exposed to FA from 96.5 to 75 % (Fig. 4d), indicating that part of the protective effect observed with NAC requires GSH synthesis.

*adh-5(sbj21)* mutant *C. elegans* larvae did not survive FA exposure (Fig. 2c). However, a treatment with 10 mmol L$^{-1}$ NAC significantly restored survival of *adh-5* mutants and also allowed the majority of the animals to develop into adulthood as assessed 72 h post FA treatment (Fig. 4e–g and Supplementary Fig. 4c, d). Remarkably, treatment of L1 larvae with a sublethal FA concentration and simultaneous exposure to the prooxidant paraquat (PQ), which generates ROS in *C. elegans*[33], further affected the development of L1 *adh-5* larvae (Fig. 4h, i). However, we could not detect a reversion of the FA cytotoxic phenotype by pre-treating adult animals for 2 h with NAC (Supplementary Fig. 4e). *C. elegans* animals might not be able to maintain high levels of intracellular NAC. Thus, the protection against FA can only be achieved by co-incubating the antioxidant and FA. Accordingly, a co-incubation of FA and NAC significantly limited P*gst-4*::GFP activation (Fig. 3g, h). Altogether, these results indicate that providing a thiol-based antioxidant can reduce FA toxicity, whereas additional oxidative damage increases FA toxicity in nematodes. Thus, cumulatively supporting that in the presence of DNA repair, FA is causing redox balance disruption, which can contribute to its cytotoxicity.

**GSH biosynthesis limits FA toxicity.** Exogenous GSH precursors can prevent FA toxicity; we, therefore, predicted that limiting endogenous GSH should increase FA toxicity even in the presence of ADH5. First, we selected concentrations of the GSH synthesis inhibitor L-BSO that were not cytotoxic to the human cancer cells Nalm6 and HCT116 (Supplementary Fig. 5a). The viability of WT HCT116 and Nalm6 cells treated with FA was significantly reduced in the presence of L-BSO, indicating that GSH synthesis contributes to cellular FA tolerance (Fig. 5a and Supplementary Fig. 5b). Moreover, the human hepatocellular carcinoma HepG2 and the chronic myeloid leukaemia HAP1 cell lines were more sensitive to FA in the presence of non-toxic L-BSO levels (Supplementary Fig. 5c, d). In Nalm6 cells, which grow in suspension, the treatment with L-BSO increased the sensitivity of ΔADH5 cells to FA (Fig. 5a), suggesting that GSH biosynthesis and ADH5 independently contribute to prevent FA toxicity in this lymphoblastic human cancer cell line. Although L-BSO is neither cytotoxic to HCT116 nor Nalm6 cells at the concentrations used in our experiments (Supplementary Fig. 5a), it is still a pharmacological avenue that might have off-target effects. Therefore, we set out to genetically inactivate GSH biosynthesis (*GCLM*) by CRISPR/Cas9 in HCT116 cells (Supplementary Fig. 5e). Concordantly with the pharmacological experiments, *GCLM* deficiency reduced cellular tolerance to FA, which could be reverted by expressing *GCLM* in trans (Fig. 5b).

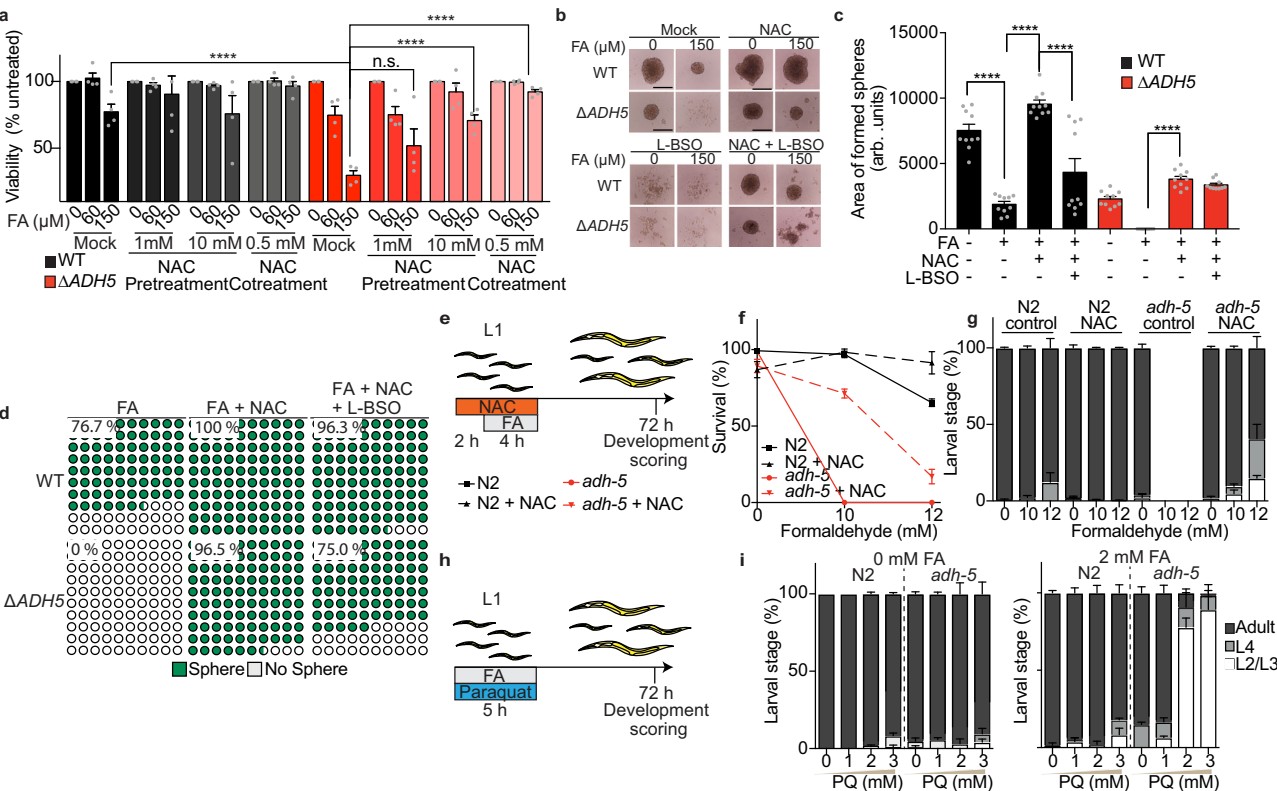

**Fig. 4 Thiol-containing antioxidants prevent formaldehyde cytotoxicity. a** Resazurin-based viability at 0, 60 and 150 µmol L$^{-1}$ formaldehyde (FA) in in Wild type (WT) and ΔADH5 HCT116 cells pre-loaded during 24 h with 1 or 10 mmol L$^{-1}$ N-acetyl-L-cysteine (NAC, pretreatment group), or simultaneously exposing cells to 500 µmol L$^{-1}$ NAC and the indicated concentrations of FA (cotreatment group). The data represent the mean ± SEM, n = 4, ****$P < 0.0001$, two-way ANOVA, Tukey´s multiple comparison test). µM refers to µmol L$^{-1}$ and mM refers to mmol L$^{-1}$. **b** Representative images of the 3D-tumour spheroid formation phenotype of WT and ΔADH5 HCT116 cells in the presence of the indicated FA concentrations and 500 µmol L$^{-1}$ NAC or 100 µmol L$^{-1}$ L-buthionine sulfoximine (L-BSO) (scale bar 0.5 mm). **c** Quantification of 3D-sphere area from 10 formed spheres at day 5 after seeding (mean ± SEM, one-way ANOVA, Tukey´s multiple comparison test, ****$P < 0.0001$). **d** Quantification of sphere formation phenotype at day 5 after seeding 2000 cell/well of WT or ΔADH5 cells in the presence of FA (150 µmol L$^{-1}$); FA and NAC (500 µmol L$^{-1}$); or FA, NAC and L-BSO (100 µmol L$^{-1}$). The plots correspond to a part of the whole representation (WT + FA, n = 30; ΔADH5 + FA, n = 28; WT + FA + NAC, n = 29; ΔADH5 + FA + NAC, n = 29; WT + FA + NAC + L-BSO, n = 27; ΔADH5 + FA + NAC + L-BSO, n = 28). **e** Scheme depicting the protocol used to treat C. elegans with FA and NAC. **f** Survival of L1-staged WT (N2) and adh-5 mutant upon exposure to the indicated FA concentrations and 10 mmol L$^{-1}$ NAC measured directly after treatment (mean ± SD, n = 3). **g** Quantification of developmental stages of surviving C. elegans animals 72 h after FA and NAC exposure (mean ± SD, n = 3). **h** Scheme depicting the protocol used to treat C. elegans with FA and paraquat (PQ). **i** Developmental stages of C. elegans 72 h after 2 mmol L$^{-1}$ FA and PQ exposure (mean ± SD, n = 3).

The simultaneous inactivation of *ADH5* and *GCLM* did not further affect viability compared to *ADH5*-single knockout cells (Fig. 5b). This observation indicates that *ADH5* is epistatic to *GCLM* in the FA viability assay performed in HCT116 cells, although this phenotype might also be a consequence of partial GSH depletion in ΔGCLM cells compared to the full GSH depletion achieved by L-BSO (Supplementary Fig. 5f). On the other hand, the 3D-sphere formation phenotype was affected by the sole inactivation of *GLCM* (Fig. 5d), concordantly with the results observed using the GSH-synthesis inhibitor L-BSO (Supplementary Fig. 5g). The formation of colonies was further impaired in ΔADH5 ΔGCLM cells compared to *ADH5*-single knockout counterparts, thus revealing an independent contribution of GSH biosynthesis and ADH5 to this phenotype (Fig. 5d, e). Altogether, these data indicate that FA metabolism and GSH synthesis are required to limit FA toxicity, sustaining cell proliferation and growth.

**Endogenous FA reacts with GSH yielding HSMGSH.** GSH and FA metabolisms are linked as FA spontaneously reacts with GSH

yielding HSMGSH (Fig. 6a), a substrate of ADH5. We hypothesized that this reaction might occur in vivo, affecting endogenous GSH as well as limiting the reactivity of free FA. By in-house synthesis and reaction monitoring using ultraperformance liquid chromatography coupled to high resolution mass spectrometry (UPLC-HRMS), we first confirmed that GSH and FA react in vitro yielding HSMGSH (Supplementary Fig. 6a–d). Should cellular metabolism generate sufficient endogenous FA, we might be able to detect the formation of HSMGSH in cells. By UPLC-HRMS, we were able to detect this compound in cell extracts (Fig. 6b, c and Supplementary Fig. 7a–c). HSMGSH accumulated in ΔADH5 cells when measured relative to GSH (Fig. 6b). However, the net amount of total GSH and HSMGSH was lower in ΔADH5 cells (Fig. 6d–f and Supplementary Fig. 7d–l), thus we cannot rule out the participation of efflux mechanism(s) pumping out HSMGSH when this product accumulates. Concordantly, FA was shown to increase the export of GSH in cultured neurons[34]. The lower amount of GSH detected in ΔADH5 cells suggests that this protein is necessary to recover reduced GSH from HSMGSH generated by the spontaneous reaction of FA and GSH. Indeed, by using an indirect fluorescent reagent, we corroborated that

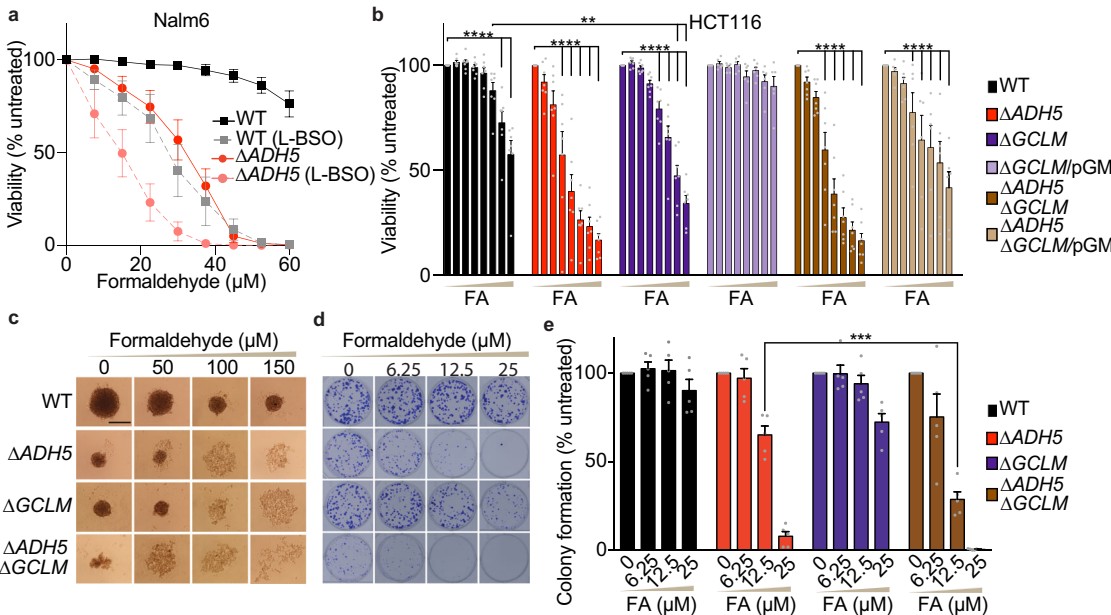

**Fig. 5 GSH biosynthesis limits formaldehyde toxicity. a** Resazurin-based viability assay in Wild type (WT) and ΔADH5 Nalm6 cells in the presence of different concentrations of formaldehyde (FA) with or without 50 μmol L⁻¹ L-buthionine-sulfoximine (L-BSO) for five days (mean ± SEM, $n = 6$). μM refers to μmol L⁻¹. **b** Resazurin-based viability assay performed with WT, ΔADH5, ΔGCLM and ΔADH5 ΔGCLM ($n = 7$) HCT116 cells and with the complemented cell lines ΔGCLM/pGCLM (pGM) and ΔADH5 ΔGCLM/pGCLM (pGM) ($n = 6$) in response to 0, 15, 30, 60, 90, 120, 150 and 180 μmol L⁻¹ FA ($n = 6$, mean ± SEM, two-way ANOVA, Tukey´s multiple comparison test, WT vs ΔGCLM: **$P = 0.0038$ (150 μmol L⁻¹ FA), **$P = 0.0097$ (180 μmol L⁻¹ FA); ****$P < 0.0001$. **c** Representative images of HCT116 3D-spheroid formation for WT, ΔADH5, ΔGCLM, and ΔADH5 ΔGCLM cells in the presence of FA at day 5 after seeding (scale bar 0.5 mm). **d** Representative images of the colony formation assay performed with WT, ΔADH5, ΔGCLM, and ΔADH5 ΔGCLM HCT116 cells in six-well plates in the presence of the indicated concentration of FA. **e** Quantification of the colony formation assay shown in (**d**) ($n = 5$, mean ± SEM, one-way ANOVA using a Tukey´s multiple comparison test, ***$P = 0.0003$).

cells lacking *ADH5* presented 17.9 % less reduced GSH than the WT counterparts (Supplementary Fig. 5f). The genetic inactivation of the regulatory component in the rate-limiting step of GSH biosynthesis (*GCLM*) or the treatment with L-BSO further depleted endogenous GSH in both ΔADH5 and WT cells, denoting that the mechanism by which ADH5 contributes to GSH homeostasis is downstream GSH synthesis. Exposing cells to FA significantly affected cellular GSH, GSSG and total GSH (GSH + GSSG) levels. In WT cells, FA slightly stimulated the accumulation of GSH and GSSG (Fig. 6d–f), which, in contrast, significantly dropped in the absence of ADH5 (Fig. 6d–f). The net level of HSMGSH in ΔADH5 cells exposed to FA was below the detection limit of the UPLC-HRMS analytic method. This limitation prevented us from calculating HSMGSH relative to GSH in ΔADH5 cells exposed to FA. In conclusion, the data presented in this section indicate that FA alters the GSH:GSSG balance and that ADH5-centred metabolism contributes to cellular GSH quota.

**HSMGSH metabolism prevents GSH:GSSG imbalance.** In the cytosol, GSH and GSSG levels have been reported to be around 10 mmol L⁻¹ and 200 nmol L⁻¹, respectively, determining a cytosolic GSH redox potential ($E_{GSH}$) of −320 mV[16]. Despite the high level of reduced GSH, a small change in the ratio between the reduced and the oxidized GSH form (GSH:GSSG) can substantially affect $E_{GSH}$, thus impairing cellular redox balance[17]. We reasoned that blocking the GSH supply through ADH5 would affect the GSH:GSSG ratio, which might consequently lead to redox imbalance. Therefore, we calculated the GSH:GSSG ratio from UPLC-HRMS data, observing a 5.9-fold reduction in ΔADH5 compared to WT HCT116 cells (Fig. 6g). The GSH:GSSG ratio was further affected by FA in ΔADH5 but not in WT cells

(Fig. 6g).To further interrogate the role of *ADH5* in maintaining the GSH:GSSG ratio upon FA stress, we incorporated the cytoplasmic version of the reporter Grx1-roGFP2[35] in WT and ΔADH5 HCT116 cells. Grx1-roGFP2 is a ratiometric reporter generated by the combination of roGFP2 and the human glutaredoxin 1 (GRX1). This set up facilitates the equilibrium between roGFP2 and the GSH redox couple, thus reporting $E_{GSH}$[35]. In a more oxidant environment, the ratio between GSH and GSSG will decrease, leading to a more oxidized Grx1-roGFP2 sensor. The fraction of the oxidized sensor (OxD Grx1-roGFP2) can be calculated from the ratio between the Grx1-roGFP2 emission at $\lambda = 510$ nm when it is excited at $\lambda = 405$ and $\lambda = 488$ nm (R405/488)[35]. We found that ADH5 prevented the FA-dependent oxidation of Grx1-roGFP2 in the cytosol (Fig. 6h, i), concordantly with the detection of H2DCFDA, DHE, and roGFP oxidation (Fig. 3a–d). Interestingly, the mitochondrial version of Grx1-roGFP2 was also oxidized by FA, but independently of ADH5, which is in accordance with the cytoplasmic localization of this enzyme (Fig. 6i). In summary, these results show that HSMGSH metabolization by ADH5 can prevent cytoplasmic GSH:GSSG imbalance by regenerating cellular GSH.

**GSH metabolism contributes to FA tolerance in DNA-repair deficient cells.** Finally, we reasoned that the reaction between FA and GSH forming HSMGSH molecules and their metabolization by ADH5 might limit free FA. It has been shown that cells lacking the ICL-DNA repair pathway Fanconi Anaemia are very sensitive to FA[9,13,36]. We, therefore, predict that GSH will be required to prevent FA toxicity in Fanconi Anaemia by limiting endogenous free FA level. To assess our hypothesis, we exposed chicken DT40 cells deficient in the *FANCI* and *BRCA2* genes to FA in the presence of L-BSO. *FANCI* codes for a protein that participates in

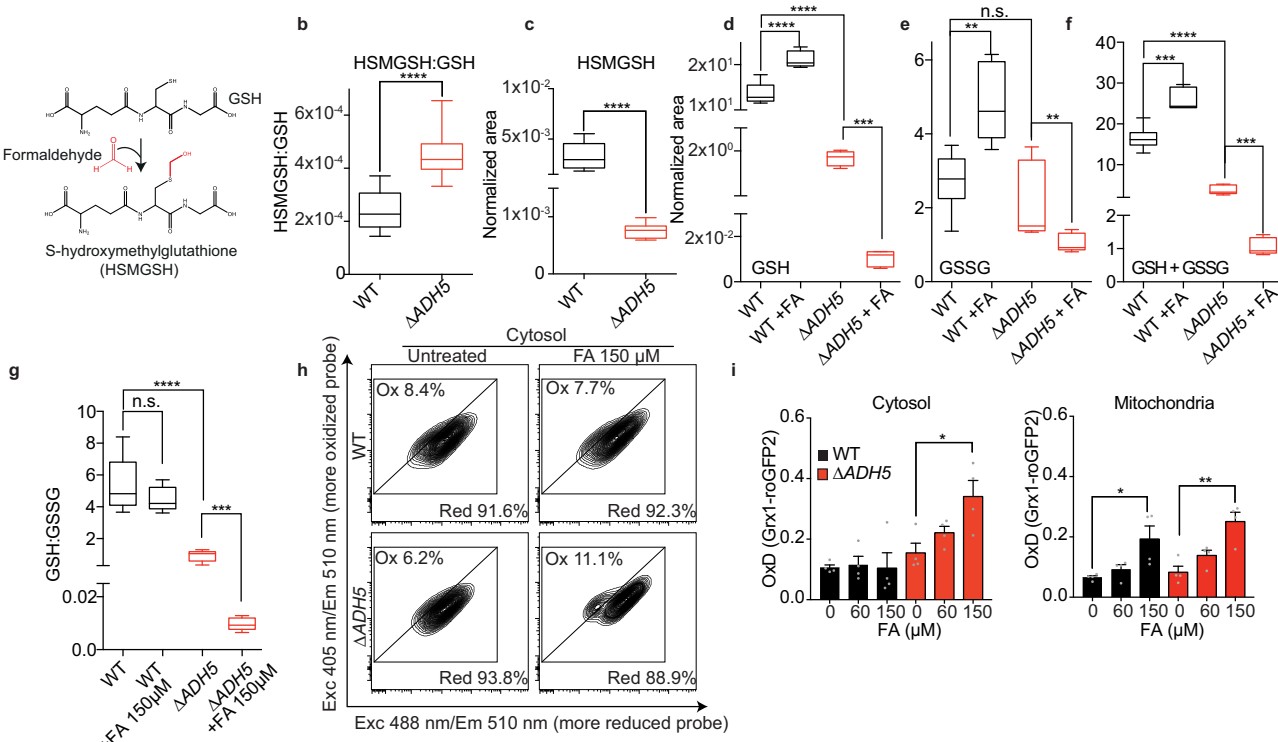

**Fig. 6 Endogenous formaldehyde reacts with GSH altering the GSH:GSSG ratio. a** Scheme showing the spontaneous reaction between formaldehyde (FA) and glutathione (GSH) yielding S-hydroxymethylglutathione (HSMGSH). **b** HSMGSH content relative to GSH in Wild type (WT) and ΔADH5 HCT116 cells ($n = 9$, two-tailed Mann–Whitney test, ****$P < 0.0001$). The box and whiskers plots: the line in the box corresponds to the median; the edges are the 25th and 75th percentiles, and the whiskers extend to the most extreme values. **c** Net HSMGSH content in WT and ΔADH5 cells calculated as normalized peak areas relative to the number of viable cells ($n = 9$, two-tailed Mann–Whitney test, ****$P < 0.0001$). Box and whiskers plotted as (**b**). **d** GSH content in WT and ΔADH5 HCT116 cells exposed to 150 µmol L$^{-1}$ FA over 48 h ($n = 9$ for untreated (UT) WT and ΔADH5 samples, $n = 5$ for WT and ΔADH5 FA-treated samples, two-tailed Mann–Whitney test, ***$P = 0.001$, ****$P < 0.0001$). Box and whiskers plotted as (**b**). **e** GSH disulfide (GSSG) content in WT and ΔADH5 HCT116 cells exposed to 150 µmol L$^{-1}$ FA over 48 h ($n = 9$, WT and ΔADH5 UT, $n = 5$ WT and ΔADH5 FA-treated samples, two-tailed Mann–Whitney test, **$P = 0.002$ (WT vs WT + FA), **$P = 0.0055$ (ΔADH5 vs ΔADH5 + FA), n.s.: not significant). Box and whiskers plotted as (**b**). **f** Total GSH (GSSG plus GSH) content in WT and ΔADH5 HCT116 cells exposed to 150 µmol L$^{-1}$ FA over 48 h ($n = 9$ for untreated WT and ΔADH5 samples, $n = 5$ for WT and ΔADH5 FA-treated samples, two-tailed Mann–Whitney test, ***$P = 0.001$ (WT vs WT + FA), ***$P = 0.001$ (ΔADH5 vs ΔADH5 + FA), ****$P < 0.0001$ (WT vs ΔADH5)). Box and whiskers plotted as (**b**). **g** GSH:GSSG ratio in WT and ΔADH5 HCT116 cells ($n = 9$ for untreated WT and ΔADH5 samples, $n = 5$ for WT and ΔADH5 FA-treated samples, two-tailed Mann–Whitney test, ****$P < 0.0001$ (WT vs ΔADH5), ***$P = 0.001$ (ΔADH5 vs ΔADH5 + FA). Box and whiskers plotted as (**b**). **h** Flow cytometry representative plots from WT and ΔADH5 HCT116 cells harbouring the cytosolic Grx1-roGFP2 reporter. **i** Quantification of oxidized Grx1-roGFP2 [OxD (Grx1-roGFP2)] ($n = 4$, mean ± SEM, one-way ANOVA, Tukey's test). Left: Cytoplasmic Grx1-roGFP2 (ΔADH5 0 vs ΔADH5 + 150 µmol L$^{-1}$ FA: *$P = 0.0186$). Right: Mitochondrial Grx1-roGFP2 (WT vs WT + 150 µmol L$^{-1}$ FA: *$P = 0.023$, ΔADH5 0 vs ΔADH5 + 150 µmol L$^{-1}$ FA: **$P = 0.002$).

the early events of the Fanconi Anaemia DNA crosslink repair pathway, while BRCA2 (FANCD1) is involved in the final steps of the resolution of the ICL, and in homologous recombination DNA repair[37]. As predicted, Δ*FANCI* and Δ*BRCA2* cells were sensitive to FA and this phenotype was exacerbated by blocking GSH synthesis (Fig. 7a). Subsequently, we addressed the effect of GSH synthesis inhibition on the tolerance to FA of transformed fibroblast from Fanconi Anaemia patients harbouring a mutation in the gene *FANCD2*, which codes for another factor of the Fanconi Anaemia DNA repair pathway (PD20, *FANCD2*$^{-/-}$)[38]. In agreement with published data, these cells were hardly sensitive to FA when compared to the isogenic cell line expressing a wild type copy of FANCD2 (PD20-wtD2)[39] (Fig. 7b). Interestingly, PD20 cells became significantly sensitive to FA in the presence of L-BSO, indicating GSH synthesis can protect these cells lacking the Fanconi Anaemia DNA repair pathway from FA (Fig. 7c). We next exposed human Nalm6 cells deficient in *FANCB*, another Fanconi Anaemia DNA crosslink repair gene, to FA in the presence of L-BSO. Δ*FANCB* cells were sensitive to FA and this phenotype was largely exacerbated by blocking GSH

synthesis (Fig. 7d). Furthermore, Nalm6 cells lacking *FANCB* were significantly sensitive to GSH inhibition even in the absence of FA, which might be a consequence of an increase in endogenous free FA, overall suggesting that GSH supply might be fundamental for Fanconi Anaemia patients (Fig. 7e).

## Discussion

In this work we revealed that FA alters the GSH:GSSG redox balance, triggers oxidative stress, and causes cytotoxicity in cells and organisms without any deficiency in DNA repair (Figs. 1, 3, and 6). Our results shed light into early observations of oxidative damage in tissues and cells exposed to FA[40, 41], revealing further cytotoxic consequences of FA, in addition to its widely characterized roles as mutagen and genotoxin. Mechanistically, we determined that FA reacts with GSH, affecting the GSH:GSSG ratio and the cellular redox balance, and described a conserved mechanism to salvage GSH from FA-GSH products (HSMGSH) limiting FA cytotoxicity in human cancer cells and in *C. elegans*. This pathway is centred on the enzyme ADH5 and it is downstream of the de novo GSH synthesis pathway (Fig. 7f).

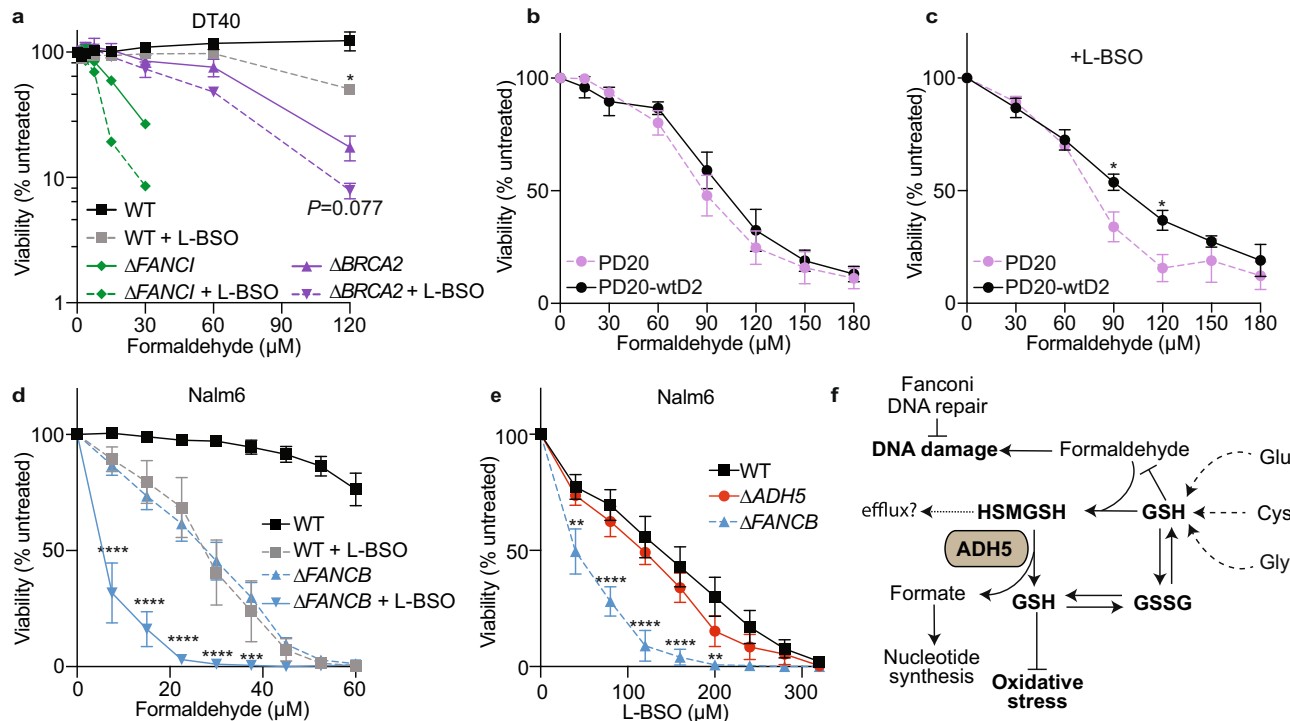

**Fig. 7 ADH5 and GSH synthesis determine formaldehyde tolerance in DNA repair-deficient cells. a** Viability assay for Wild type (WT), ΔBRCA2 and ΔFANCI DT40 cells exposed to formaldehyde (FA) and 50 μmol L⁻¹ L-buthionine-sulfoximine (L-BSO) for 3 days ($n = 3$, mean ± SEM, for WT and ΔBRCA2, two-tailed unpaired t-test, *$P = 0.0264$ (WT), $n = 2$ for ΔFANCI). For ΔFANCI cells at 60 and 120 μmol L⁻¹ FA, the viability values after subtraction of the blank were cero and were excluded to perform the logarithmic plot. μM refers to μmol L⁻¹. **b** Resazurin-based viability assay for PD20 (FANCD2⁻/⁻) cells and the complemented version expressing a WT copy of FANCD2 (PD20-wtD2) exposed to increasing concentrations of FA for three days ($n = 6$ for PD20 and $n = 5$ for PD20-wtD2, mean ± SEM). **c** Viability assay for PD20 and PD20-wtD2 cells exposed to FA in the presence of 500 μmol L⁻¹ L-BSO for three days ($n = 6$, mean ± SEM, two-way ANOVA, using a Bonferroni test for multiple comparison, *$P = 0.0417$ (90 μmol L⁻¹ FA), *$P = 0.0226$ (120 μmol L⁻¹ FA). **d** Viability assay for Nalm6 cells exposed to FA and 50 μmol L⁻¹ L-BSO ($n = 5$, mean ± SEM, two-way ANOVA for multiple comparison using a Bonferroni test between ΔFANCB and ΔFANCB + L-BSO 50 μmol L⁻¹, ****$P < 0.0001$, ***$P = 0.0007$). **e** Viability assay for Nam6 cells exposed to L-BSO ($n = 6$, mean ± SEM, two-way ANOVA for multiple comparison using a Bonferroni test between WT and ΔFANCB cells, **$P = 0.0056$ (40 μmol L⁻¹), ****$P < 0.0001$, **$P = 0.0030$ (200 μmol L⁻¹)). **f** Model for FA metabolism and the crosstalk with GSH metabolism highlighting the formation of HSMGSH products and their metabolization through ADH5. GSH supply by ADH5 limits oxidative stress and sustains the balance between GSH and GSH disulfide (GSSG).

FA is a well-established genotoxin that can reach blood levels close to 50 μmol L⁻¹[6]. The genotoxicity of FA can be prevented by ICL-DNA repair factors, and indeed FA can lead to bone marrow failure and cancer when those DNA repair mechanisms fail[9]. The discovery of patients carrying mutations in FA clearance systems that develop hematopoietic phenotypes has called into question whether genotoxicity is the sole FA-damaging consequence in cells[10, 14]. Indeed, the data presented here indicate that FA can cause cytotoxicity when FA clearance fails, even in the presence of DNA repair (Fig. 1a–c). We speculate that in the presence of functional DNA repair mechanisms and physiological levels of FA, FA-caused DNA damage would be alleviated before it reaches the threshold required for triggering apoptosis. Accordingly, we detected a mild phosphorylation of p53 in ΔADH5 HCT116 cells exposed to FA (Fig. 1f), and an increase in p-CHK2 (Supplementary Fig. 2c), suggesting that in the presence of DNA repair, FA-caused DNA damage can only mildly activate a CHK2-dependent DNA damage response. However, this DNA damage is still not sufficient to significantly damage chromosomes (Fig. 1h), which would trigger p53-dependent apoptosis[13]. Concordantly, the inactivation of p53 did not rescue the viability of ΔADH5 cells in the presence of FA (Fig. 1g).

The detection of HSMGSH in cells not exposed to exogenous FA indicates that cellular metabolism continuously produces sufficient FA to react with GSH yielding HSMGSH (Fig. 6b, c). To

understand the biology underlying FA metabolism, we supplemented the culture medium with FA. Although we were using concentrations near those reported for human blood, the amount of FA that effectively reaches the intracellular space is unknown and becomes a limitation of our study. Remarkably, It was recently reported that in differentiating hematopoietic cells there is a sharp increase in nuclear FA[42]. In this system, the DNA damage detected in the absence of the Fanconi Anaemia DNA repair pathway is comparable to the damage detected by incubating cells with 0.5 mmol L⁻¹ FA, which is close to the level reported for cellular GSH. Thus, it is plausible that a local rise in endogenous FA might alter intracellular GSH leading to redox imbalance. Concordantly, we showed that when HSMGSH metabolization is impaired, FA alters GSH:GSSG ratio (Fig. 6g–i). We proposed this GSH:GSSG imbalance can lead to ROS accumulation and cytotoxicity. FA-dependent oxidative stress induction is supported by five different techniques (roGFP, H2DCFHDA, PBSF, DHE, and ESR). Recently, FA has been described to form a hemithioacetal with cysteine[43], which might potentially affect the functionality of roGFP. To overcome this limitation, we have included probes that do not rely on cysteine oxidation to report ROS and quantified the effect of FA on GSH:GSSG levels by UPLC-HRMS. We also detected a significant drop in total GSH in ΔADH5 cells, which might indicate that the active efflux of

HSMGSH could contribute to a drop in the net cellular GSH thus also affecting cellular redox homeostasis (Fig. 6d–f). Accordingly, in a recent genome-wide CRISPR-screening, it was reported that the inactivation of the gene coding for the ABC-transporter ABCC1, which pumps GSH-conjugates out of the cell, favours cell growth in the presence of FA[44]. Alternatively, due to its strong electrophilicity, FA might react with other abundant cellular thiols such as free cysteine[45] and with sulfhydryl groups present in proteins involved in redox control, thus also contributing to cellular redox disruption. This strong FA electrophilicity also limits the interpretation of the experiments shown in Fig. 3 using antioxidants to suppress FA cytotoxicity. The rescue of cytotoxicity might be because oxidative stress is reduced, or because FA is being quenched by NAC/GSH-MEE, or both. Our data cumulatively support oxidative stress induction upon FA and suggest that cytotoxicity is, at least in part, caused by the accumulation of oxidant species.

Several factors—in addition to GSH—have been implicated in the cellular protection against oxidative stress, most of them being under the control of the master regulator NRF2[46]. In *C. elegans*, *gst-4*, which is controlled by the NRF2 ortholog SKN-1, is upregulated in the presence of FA (Fig. 3g, h), indicating that FA can trigger a SKN-1-dependent response. In humans, upon detecting oxidative stress, the NRF2 partner KEAP1 no longer ubiquitin-labels NRF2 for degradation, resulting in NRF2 stabilization and activation of NRF2-response genes. This mechanism might also operate to mount a FA-dependent response. Indeed, inactivating *Keap1* has been shown to rescue a phenotype of diet-induced steatohepatitis reported in *Adh5*$^{-/-}$ mice[47]. NRF2 is a tumour suppressor gene that also controls stem cell fate and the crosstalk between NRF2 and HSMGSH metabolism might have significant consequences beyond cancer.

ADH5 can also metabolize S-nitrosoglutathione (GSNO) producing ammonia and GSH[15]. This enzymatic activity gave origin to the alternative name of ADH5, GSNOR, proposing it is a protein denitrosylase that can modulate the level of S-nitrosylations by reducing GSNO[48]. S-nitrosylations are post-translational modifications (PTM) that originate from the reaction between the messenger nitric oxide (NO) and a thiol group in a cysteine residue. These PTM were indicated as intermediates toward the formation of disulfide bonds, likely affecting signalling pathways[49]. The indirect role of ADH5 in NO homeostasis has prompted the development of pharmacological inhibitors that might be used for the treatment of inflammatory diseases[50]. Remarkably, GSH is the common product of the enzymatic activity of ADH5 using either HSMGSH or GSNO as substrates. Thus, blocking ADH5 would trigger adverse effects such as GSH redox imbalance and increased toxic endogenous FA.

We showed that FA induces an accumulation of ROS comparable to the one measured upon blocking GSH synthesis with L-BSO (Fig. 3a). However, in HCT116 cells, L-BSO was not cytotoxic even at concentrations as high as 3.2 mmol L$^{-1}$ (Supplementary Fig. 5a), whereas cells died at FA concentrations above 100 μmol L$^{-1}$. This observation might be seen as contradictory because, despite both chemicals induced comparable ROS, only FA was able to trigger cytotoxicity in HCT116. The explanation for this discrepancy can be found in recent papers showing that some cancer cell lines—including HCT116—are not affected by L-BSO or GSH synthesis inactivation because they can compensate GSH depletion by the thioredoxin (TXN) pathway[25], and by maintaining protein homeostasis through deubiquitinating enzymes (DUB)[32]. Further research should clarify the contribution of DUBs enzymes to FA tolerance. In stark contrast, the leukemic Nalm6 cell line was very sensitive to L-BSO and to very low concentrations of FA (Fig. 5a and Supplementary Fig. 5a), and co-treatment with the GSH synthesis inhibitor further

increased FA cytotoxicity (Fig. 5a). Therefore, it is likely that FA-induced oxidative cytotoxicity might be more prevalent in blood-derived cells, which is consistent with the hematopoietic phenotypes reported in IBMFS/AMeD patients.

Our findings may have wide implications for human health. It was recently shown that biallelic mutations in *ADH5* and accumulation of FA can cause a particular type of IBMFS observed in individuals carrying the highly prevalent *ALDH2*2* mutation (IBMFS/AMeD syndrome)[10, 14]. Concordantly, mice deficient in Aldh2 and Adh5 recapitulate some of the hematopoietic phenotypes of IBMFS/AMeD patients[10]. Remarkably, the two top enriched GO terms within the top 100 differentially regulated genes in hematopoietic stem cells from *Aldh2*$^{-/-}$ *Adh5*$^{-/-}$ mice correspond to cellular response to oxygen-containing compounds (GO1901700 and GO1901701)[10], suggesting oxidative stress might play an important role in the onset of IBMFS/AMeD phenotypes. In the human condition Fanconi Anaemia, metabolic ROS were shown to induce DNA damage in hematopoietic stem cells (HSCs) when they start cycling to exit quiescence, which impairs blood production in *FancA*$^{-/-}$ mice[51]. Moreover, mice lacking the cytosolic Cu/Zn superoxide dismutase 1 (Sod1) and the Fanconi Anaemia DNA repair pathway gene *FancC* develop hematopoietic failure and liver steatosis[52]. It is also known that oxygen can exacerbate chromosome aberrations in lymphocytes from Fanconi Anaemia patients[53], that the GSH precursor NAC improves genome stability in these lymphocytes[54], and that nuclear FA increases during hematopoietic cell differentiation[42]. Hence, an increase in ROS might affect the GSH pool, indirectly leading to further accumulation of FA and genome instability. A combined therapy using a FA scavenger such as metformin[55] and GSH-precursors like NAC might succeed in benefiting IBMFS/AMeD and Fanconi Anaemia patients. Furthermore, a diet rich in GSH-precursors might delay cancer onset in healthy *BRCA2*-mutation carriers by limiting FA toxicity. However, additional studies are necessary to assess any therapeutic intervention suggested from our findings.

## Methods

### Experimental model and subject details

*Cells and animals.* HCT116 were obtained from G. Soria's laboratory, originally generated by Vogelstein laboratory. PD20 were from V. Gottifredi laboratory, originally from Coriell repositories. HepG2 were obtained from KJ Patel's laboratory, originally from ATCC collection. These cell lines were maintained in Dulbecco's Modified Eagle's Medium (DMEM) (Thermo Scientific, #12100061)—low glucose for HepG2 (Thermo Scientific, #31600091), supplemented with 1 % Penicillin/Streptomycin (P/S) and 10 % Fetal bovine serum (FBS) (Natocor). Nalm6 cells were obtained from KJ Patel's laboratory, originally from H. Koyama laboratory, and were maintained in Roswell Park Memorial Institute 1640 medium (RPMI) (Thermo Scientific, #31800105) containing 10 % FBS, 1 % P/S, and 50 μmol L$^{-1}$ ß-mercaptoethanol. DT40 cells—from KJ Patel's laboratory stock—were grown in RPMI medium supplemented with 3 % chicken serum (Gibco, # 16010167), 7 % Fetal calf serum (Gibco, # 16110082), 50 μmol L$^{-1}$ β-mercaptoethanol, and P/S[11]. HAP1 cells—from KJ Patel's laboratory, originally from Haplogen—were grown in Iscove's Modified Dulbecco's Medium (IMDM) medium (Thermo Scientific, #12200069) supplemented with 10 % FBS and P/S. *FANCD2*-deficient PD20 cells (GM16633—Coriell Repositories) and the reconstituted PD20-wtD2 (GM16634—Coriell Repositories, a microcell hybrid expressing low levels of wild type FANCD2) were a gift from V. Gottifredi. All the cell lines were regularly tested for mycoplasma infection. *Caenorhabditis elegans* was maintained using standard methods[56]. N2, Bristol *C. elegans* wild isolate was obtained from *Caenorhabditis* Genetics Centre (CGC), Minneapolis, MN, USA. This work did not require ethical approval.

*CRISPR/Cas9 generation of ΔADH5 and ΔGCLM cell lines.* HCT116 ΔADH5, ΔGCLM, ΔTP53 ΔADH5, and ΔADH5 ΔGCLM cell lines were generated by targeting exon 3 of *ADH5* (sgRNA: TGCTGGAATTGTGAAAGTGTT) and exon 1 of *GCLM* (sgRNA: ACGGGGAACCTGCTGAACTG) in the corresponding parental cell lines. Briefly, sgRNAs were cloned into the pX458 vector (Addgene, #48138) and transfected using lipofectamine 3000 (Thermo Scientific, L3000015). GFP-expressing cells were sorted and clonally diluted in 96-well plates. After 20 days, cells were expanded, and inactivation of the desire gene confirmed by western blot using GCLM (Atlas antibodies, #HPA023696, 1:500) or ADH5 (1:500) antibodies[9]. The mutations

generated by Cas9 at the target exons were obtained by preparing genomic DNA from the selected clones and amplifying the exons with the primers: Fwd (hA5-ck1): 5′-TCTTGTATCTGTACCTCTGA-3′; Rv (hA5-ck1rv): 5′-CCTTCAGCTTAGTAA CTC-3′ for ADH5, and Fwd (hGCLM-ck_834Fw): 5′-GAAGCACTTTCTCGGCTA CG-3′; Rv (hGCLM-834_Rv): 5′-TCCTTTACCTGGACAGGGTG-3′ for GCLM. PCR results were analysed by gel electrophoresis, cloned, and sequenced using universal M13 primers.

*Generation of cells stably expressing ADH5 and GCLM.* Cells carrying the ADH5-expressing plasmid pLox-ADH5-FLAG-CT-BSR[3] were selected using 4 µg mL$^{-1}$ Blasticidin (BSR). BSR-resistant cells were clonally diluted and ADH5-expression verified by western blot against FLAG epitope (Abcam, #ab49763, 1:2000).

The human GCLM gene was amplified from the MGC clone MHS6278-202809187 and cloned into the SalI-opened pAAVS1[57] vector fused to mCherry by Gibson assembly. The primers used for GCLM amplification were 5′-TTTTGGCA AAGAATGGTCGACCGCTGCCATGGGCACCGAC-3′ and 5′-GCCCTTGCTCA CCATGTCGATAGAACCCCTTCTTTTAGCTTGTAAAATG-3′. The vector generated (pAAVS1-PC-GCLM::Cherry) was confirmed by sequencing and co-transfected into HCT116 cells together with the AAVS1-TALEN-R/AAVS1-TALEN-L plasmids (Addgene #59026 and #59025, respectively). Puromycin-resistant cells were selected using 5 µg mL$^{-1}$ puromycin. The population expressing GCLM::Cherry was enriched using a Becton Dickinson FACSAria II cell sorter and used for complementation experiments.

*Generation of* C. elegans *lines.* The ADH-5::GFP reporter line *Ex*[p$_{adh-5}$ADH-5::GFP; p$_{myo-2}$tdTomato] was produced via co-injecting the clone (5736523864883943 G06, tagged gene: H24K24.3) of the TransgeneOme fosmid library[58] together with the selection marker for tdTomato expression in the pharynx, using standard *C. elegans* microinjection[59]. The ADH-5 rescue line was produced by injecting the beforementioned fosmid in the *adh-5(sbj21)* mutant, resulting in the line *adh-5(sbj21);Ex*[p$_{adh-5}$ADH-5::GFP]. For imaging, various stages of the transgenic animals were mounted on 5 % agar pads with polystyrene nanoparticles (Polysciences, 2.5 % by volume)[60] and imaged at an AxioImager M.2 fluorescence microscope (Zeiss, Jena, Germany).

The *C. elegans* orthologue of human *adh-5* gene (H24K24.3) was knocked out using the CRISPR/Cas9 system as previously reported: The preassembled CRISPR/Cas9 ribonucleoprotein complexes and linear single stranded DNAs as repair templates were directly injected into the gonad of young adult hermaphrodites[61]. To generate *adh-5* null mutant, we utilized a universal STOP-IN cassette that contained an exogenous Cas9 target site, multiple stop codons in all three reading frames, and the recognition site of the *NheI* restriction enzyme[23]. The *C. elegans adh-5* sgRNA with GGG protospacer adjacent motif was designed using Benchling (https://benchling.com/) and targeted exon 3 of the *adh-5* gene (5′-CTTCATGT CCCAAGACGACA-3′). The *C. elegans* DNA repair oligo included a STOP-IN cassette and two short homology arms identical to the sequences flanking the Cas9 cleavage site (5′- GCCACACGGACGCCTACACCCTCGACGGACACGATCCGG AAGGTCTCTTCCCTGTGGGAAGTTTGTCCAGAGCAGAGGTGACTAAGT GATAAGCTAGCCGTCTTGGGACATGAAGGGTCTGGAATTGTCGAGA-3′). To facilitate screening, a co-conversion strategy with dominant phenotypic roller marker was used[62]. Microinjection was performed using the following injection mix: KCl (25 mmol L$^{-1}$), Hepes pH 7.4 (7.5 mmol L$^{-1}$), tracrRNA (200 ng µL$^{-1}$), *dpy-10* crRNA (150 ng µL$^{-1}$), *dpy-10* ssODN (13.75 ng µL$^{-1}$), *adh-5* sgRNA (300 ng µL$^{-1}$), *adh-5* ssODN (100 ng µL$^{-1}$), Cas9 (416 ng µL$^{-1}$, NEB, USA)[59]. F1 worms carrying roller phenotype were preselected and cloned 4-6 days after the injection. The F2 progeny was subsequently screened for the desired edit by PCR amplification using the *adh-5* forward primer (5′-CGATCCAAGTGGCTCCACC GAA-3′) and the *adh-5* reverse primer (5′-TTCCACATCCCAAAAGCGAAAC C-3′). The presence of the STOP-IN cassette was verified via Sanger sequencing (Eurofins Genomics, Germany) with the *adh-5* sequencing primer (5′-CGATTAACCGACACCCTTGCTC-3′).

*Survival and development assays in* C. elegans. For the combined N-acetyl-L-cysteine (NAC, Sigma-Aldrich, #A7250) and formaldehyde (FA, Pierce, #28908) treatment worm stages were first synchronized via bleach-synchronization. Gravid adult animals and eggs were harvested from NGM plates with 5 mL M9 buffer (3 g KH$_2$PO$_4$, 6 g Na$_2$HPO$_4$, 5 g NaCl, in 1 L H$_2$O; autoclaved and added 1 mL 1 M MgSO$_4$) using a cell scraper and transferred to 15 mL tubes, before adding 1 mL bleach solution (5 M NaOH and sodium hypochlorite in a 1:1 ratio). The tubes were then constantly vortexed for 5 min and centrifuged (Centrifuge 5810R, Eppendorf) at 1000 × *g* for 1 min. After removing the supernatant, the worms were washed three times with 5 mL M9 medium by shaking the tubes and then centrifuging at 1000 × *g* for 1 min. Finally, they were kept in 10 mL M9 medium overnight (16 h) under rotation at 35 rpm (Multiple-Axle-Rotating-Mixer RM10W-80V, CAT) to allow animals to hatch. Prior to FA treatment, L1-staged worms were centrifuged at 200 × *g* for 1 min, and the volume was reduced to 1 mL M9 medium. The number of worms was determined under a stereoscope in a representative volume of 3 µL and a final concentration of approx. 50–100 worms per µL was adjusted. A solution was prepared by pelleting a saturated OP50 *E. coli* bacterial culture, which was first heat-inactivated (60 °C O/N), at 2000 × *g* for 10 min, and then concentrated two-fold in M9 plus cholesterol (5 µg mL$^{-1}$). 5 mL

aliquots were prepared, and 1000 worms were added in a volume of 10–20 µL. NAC (500 mmol L$^{-1}$ stock solution in H$_2$O) was added to a final concentration of 10 mmol L$^{-1}$ to half of the aliquots and incubated under rotation for 2 h. Thereafter, various concentrations of FA (10 and 12 mmol L$^{-1}$) were added to the tubes and incubated under rotation for another 4 h. To this end, methanol-free 16 % FA (w/v; Thermo Scientific) was first adjusted to a 1 M stock solution in H$_2$O, which was prepared fresh for each experiment. After the NAC/FA treatment, the solutions were centrifuged at 200 × *g* for 1 min, followed by two washing steps with 5 mL M9 medium. Finally, worms were pelleted again, and the volume was reduced to 500 µL. A volume of 25–50 µL (approx. 50–100 worms) was transferred to OP50-seeded NGM plates, on which the survival rate was scored under a stereoscope. Worms were qualified as dead when no locomotion could be detected and when stimulation with a wormpick did not cause a response. The survival count was repeated after 24, 48 and 72 h. In parallel, developmental stages of worms were determined under the stereoscope at 48 and 72 h post-treatment and qualified in the categories L1-L3, L4, and adult.

For the pretreatment experiment, NAC was removed by washing the worms three times with M9. Then, FA was applied in the indicated concentrations and the animals were incubated for another 4 h, followed by the quantification of survival right after the treatment.

The survival of the ADH-5 rescue line *adh-5(sbj21);Ex*[p$_{adh-5}$ADH-5::GFP; p$_{myo-2}$tdTomato] was performed in adult worms. Animals were bleach-synchronized as stated earlier, then grown into day-1-adults and the FA-treatment was performed as in L1 animals, with the exception that the FA concentrations were increased (indicated in the Figure).

The combined PQ and FA treatment was performed in the same way as the NAC/FA treatment, with the exception that PQ was added at the same time as FA (2 mmol L$^{-1}$) and incubated together for a total of 5 h. PQ (Methyl viologen dichloride hydrate/paraquat, Sigma-Aldrich, #856177) was always freshly prepared and first adjusted to a 1 M stock in H$_2$O, which was further diluted for the treatment.

*Imaging ROS-induced fluorescence reporters in* C. elegans *upon* adh-5(RNAi). We used the strain JV1, *unc-119(ed3);jrIs1[rpl-17*p::HyPer + *unc-119*(+)] to image the induction of H$_2$O$_2$[30], and the strain CL2166, *dvIs19*[(pAF15)*gst-4*p::GFP::NLS][63] for oxidative stress signalling induction. The animals were grown on plates with HT115 bacteria expressing control RNAi (empty vector) or RNAi against *adh-5* for two generations, respectively, following established RNAi protocols[64]. The RNAi construct was produced by amplifying a part of the *adh-5* gene (exon 3) with the forward primer (5′-ATAGGTACCCGTACCAGTTGAAATTCCAC-3′) and the reverse primer (5′-ATAGCGGCCGCGATTAACCGACACCCTTG-3′). Through the primers we introduced KpnI and NotI restriction sites, which were used to ligate the PCR-amplified gene region into the multiple cloning site of the RNAi vector L4440. Adult animals were treated with 25 mmol L$^{-1}$ FA and/or 10 mmol L$^{-1}$ NAC for 5 h, in the presence of heat-inactivated RNAi-expressing bacteria. Right after, animals were washed twice with M9 buffer, plated to RNAi plates, and grown for 16 h. To image fluorescence intensity in whole animals, >25 animals were immobilized with leva-misole and mounted on 2 % agarose pads, followed by imaging at a Zeiss AxioI-mager.M2 with a 5× lens and the eGFP filter. Images were quantified via Fiji/ImageJ for mean pixel intensity. For ratiometric measurements of the HyPer reporter the animals were imaged at a Zeiss Meta 710 confocal microscope at an excitation of 405 and 488 nm, while emission was detected at 516 nm. Mean pixel intensities at 405 and 488 nm were quantified via Fiji/ImageJ and the ratio 488/405 nm was calculated.

*Viability and survival assays.* For determining cell viability, HCT116, PD20, HepG2, and HAP1 cells were seeded into 96-well plates at a density of 3000 cells per well and allowed to attach for 24 h at 37 °C, 5 % CO$_2$. Then, iodoacetamide (IAA, Sigma-Aldrich, #I1149) or FA and/or L-buthionine-sulfoximine (L-BSO, Sigma-Aldrich, #B2515) and/or antioxidants were added to a final volume of 200 µL per well. Three days later, resazurin (Sigma-Aldrich, #R7017) was added to a final concentration of 30 µmol L$^{-1}$ in the growing medium. Fluorescence ($\lambda_{ex}$ = 525 nm; $\lambda_{em}$ = 590 nm) was measured 3 h later in an Enspire Plate Reader (Perkin Elmer). For time-shifted experiments, HCT116 cells were seeded in 96-well plates (3000 cells/well) in DMEM medium containing NAC. Twenty-four hours later, the NAC-containing medium was washed off, and replaced with fresh medium containing the indicated FA concentration, to a final volume of 200 µL per well. For Nalm6 5000 cells per well were seeded into 96-well plates, the drugs to be tested added immediately afterwards. On day four, resazurin was added to a final concentration of 30 µmol L$^{-1}$ and fluorescence determined the next day. For DT40 and HepG2 cells, viability was determined on day three after plating by using the reagent CellTiter (MTS, Promega, #G3582)[3]. In all cases, the experiments were done by triplicate, and data represented as percentage of the fluorescence obtained with the untreated samples of the corresponding cell line.

The colony survival assay was done by seeding 600 cells per well in six-well plates. Immediately afterward, FA was added at the concentrations described in the text in a final volume of 2 mL (DMEM). Plates were maintained during 7–10 days at 37 °C, 5 % CO$_2$. Staining was done using a fixative/staining solution (0.5 % crystal violet, 6 % glutaraldehyde) for 30 min, following extensive rinse with tap water. Visible colonies were counted, and the results expressed as a percentage of the untreated wells. Experiments were done by duplicate and repeated the number of times indicated in each corresponding figure.

**ROS measurements.** For 2′,7′-dichlorodihydrofluorescein diacetate (H2DCFDA, Sigma-Aldrich #D6883) and pentafluorobenzenesulfonyl fluorescein (PBSF, CaymanChem #728912-45-6) assays, $4 \times 10^4$ HCT116 cells were seeded per well in 24-well plates and allowed to adhere overnight. Cells were then treated with 0, 60 and 150 µmol L$^{-1}$ of FA or 100 µmol L$^{-1}$ of L-BSO, for 48 h. After treatment, H2DCFDA or PBSF were added to each well at a final concentration of 10 µmol L$^{-1}$, kept in the dark at 37 °C for 30 min. Then, cells were lifted and transferred to flow cytometry tubes, which were kept at 4 °C until measuring was performed. Fluorescence ($\lambda_{ex} = 488$ nm; $\lambda_{em} = 530$ nm) was measured by flow cytometry using a Becton Dickinson's FACS Canto II Flow cytometer. For the dihydroethidium-based assay kit (DHE, CaymanChem #601290), cells were treated exactly as described for H2DCFDA. 10 µmol L$^{-1}$ Antimycin A was used as positive control and added with ROS staining buffer according to manufacturer's instructions. The DHE fluorescent signal ($\lambda_{ex} = 488$ nm and $\lambda_{em} = 564$–606 nm) was measured by flow cytometry using a Becton Dickinson's FACS Canto II Flow cytometer.

For ESR spectroscopy, HCT116 cells were exposed to 0, 60, and 150 µmol L$^{-1}$ FA and 100 µmol L$^{-1}$ L-BSO for 48 h. Then, cells were trypsinized and $2 \times 10^6$ cells incubated with 0.5 mmol L$^{-1}$ of the spin probe CMH (1-hydroxy-3-methoxycarbonyl-2,2,5,5-tetramethylpyrrolidine), which detects intracellular and extracellular superoxide ion, peroxyl radical, peroxynitrite, and nitrogen dioxide[28]. After incubation for 30 min at 37 °C, samples were shock-frozen in liquid nitrogen. ESR experiments were set up to obtain 6 independent measurements from each cell line and condition. After processing, four samples did not reach the $2 \times 10^6$ cells, which was used as a threshold number for experiments. ESR measurements were performed using the EMXmicro spectrometer (Bruker Biospin, Reinstetten, Germany) with microwave frequency 9.43 GHz, modulation frequency 100 kHz, modulation amplitude 3 G, and microwave power 2 mW.

**ROS and GSH redox status determination by genetic sensors.** The cytosolic roGFP2 sensor was cloned from Addgene 49435 into a retroviral backbone pLPCX by Gibson Assembly. Plasmid sequence was confirmed by sequencing. Retroviral infection was carried out transfecting HEK293T cells with pBS-CMV-gagpol (Addgene, #35614) and pCAG-VSVG (Addgene, #35616) vectors in addition to pLPCX cyto Grx1-roGFP2 (Addgene, #64975) and pLPCX myto Grx1-roGFP2 (Addgene, #64977). pLPCX cyto-roGFP was generated from Cyto-roGFP (Addgene, #49435). Conditioned medium was collected and recipient HCT116 cells infected adding Polybrene (Merk, #TR-1003-G) (1 µg µL$^{-1}$). Infection was confirmed by GFP-expression (90 % efficiency). Cells expressing the desired reporter were selected with 0.5 µg mL$^{-1}$ puromycin. No clonal selection was carried out to prevent single clone artefacts. For ROS measurement, cells were seeded into a 24-well plate at $3.5 \times 10^4$ cells per well and allowed to adhere overnight. Cells were then treated with 0, 60 and 150 µmol L$^{-1}$ of FA, and 0 or 100 µmol L$^{-1}$ of L-BSO for 48 h. H$_2$O$_2$ 500 µmol L$^{-1}$, which was used as a positive control, was added 15 min prior cell analysis. We also performed this experiment treating cells with 0.04 µmol L$^{-1}$ or 0.16 µmol L$^{-1}$ IAA for 48 h. After treatment, culture media was removed, and cells were washed with PBS. Cells were then trypsinized and transferred to clear flow cytometry tubes containing phosphate buffer saline (PBS) supplemented with 2 % FBS. Tubes were kept at 4 °C until measuring was performed. Fluorescence ($\lambda_{ex} = 405$ and 488 nm, $\lambda_{em} = 510$ nm) was measured by flow cytometry using a Becton Dickinson's FACS Aria II flow cytometer.

For determination of the glutathione (GSH) redox potential, cells expressing pLPCX cyto or mito Grx1-roGFP2 were exposed to 0, 60, and 150 µmol L$^{-1}$ of FA and/or L-BSO for 48 h. Then, cells were collected and fluorescence ($\lambda_{ex} = 405$ and 488 nm, $\lambda_{em} = 510$ nm) determined by flow cytometry as described above[35]. The fraction of oxidized Grx1-roGFP2 sensor was calculated using the Eq. 1.

$$OxD_{roGFP2} = \frac{R - R_{red}}{\left(\frac{I_{488min}}{I_{488max}}\right) x (R_{ox} - R) + (R - R_{red})} \tag{1}$$

**Metaphases analysis.** To assess single-chromosome damage, HCT116 Wild type (WT) and ΔADH5 cells were plated in P60 dishes, allowed to adhere and then treated with mitomycin C (MMC, Santa Cruz, #sc-3514) 0.5 µg mL$^{-1}$ during 24 h or with FA 150 µmol L$^{-1}$ during 48 h. 16 hours before harvesting the cells Colcemid (Gibco, #15212-012) was added at the concentration of 0.08 µg mL$^{-1}$ without replacing the culture medium. Cells were washed with PBS and trypsin was added to a final concentration of 0.125 %. Complete medium was added to stop trypsin reaction and clumps of cells disrupted by pipetting. Then, cells were centrifuged and resuspended into 2 mL of prewarmed hypotonic solution (KCl 0.075 mmol L$^{-1}$) and incubated in 14 mL of this solution for 15 min at 37 °C. Then, 1 mL of fixative solution was added (3:1 methanol:glacial acetic acid) dropwise. Cells were washed twice with fixative solution and then dropped onto chilled humid slides, where cells were left to dry overnight. The day after, slides were stained in 2 % Giemsa solution (Thermo Scientific, #10092013) prepared in Gur buffer (Gibco, #10582-013), left to dry and mounted using BC solution (Cicarelli, #891). Pictures were taken using a Zeiss Axiobserver Z1 microscope with a 40× oil-immersion objective and analysed using ImageJ[65]. To guarantee unbiased quantification, pictures were taken by a microscopy technician, who labelled the images with numbers. After scoring of chromosome damage, the identities of the images were revealed.

**3D-Spheroid assay.** 96-well plates were coated with 50 µL of 1.5 % sterile agarose 2 h before cell plating. 100 µL of HCT116 cells (WT, ΔADH5, ΔADH5 complemented, ΔGCLM, or ΔADH5 ΔGCLM) were seeded at a concentration of $2 \times 10^3$ or $4 \times 10^3$ cells/well. Immediately after seeding, cells were treated with 100 µL of DMEM 10 % FBS containing 2× concentrations of the drugs used. The final concentrations of the drugs were 0, 50, 100, and 150 µmol L$^{-1}$ FA; 0 and 100 µmol L$^{-1}$ L-BSO; 0 and 500 µmol L$^{-1}$ NAC; 0 and 1 mmol L$^{-1}$ Glutathione monoethyl ester (GSH-MEE, Santa Cruz, # sc-203974); 0 and 1 mmol L$^{-1}$ Trolox (Sigma-Aldrich, #238813). Plates were kept at 37 °C. Spheroid formation was assayed by microscopy (Zeiss Axio A1 inverted microscope) 5–7 days after seeding and registered using a CANON Rebel T3i camera attached to the microscope with an appropriate adaptor, at 40× magnification. For sphere-size quantification, ImageJ was used to measure the area of the formed sphere. Each condition was tested by technical triplicates. Spheroid formation was quantified by assessing the percentage of wells that formed spheroids.

**Phylogenetic analysis.** Eukaryote orthologs of ADH5 were obtained from NCBI, CLUSTAL at phylogeny.fr was used to align the sequences, and TreeDyn at phylogeny.fr for tree generation[66].

**Western blot analysis.** Cells were washed with PBS containing 1 mmol L$^{-1}$ N-Ethylmaleimide (NEM, Santa Cruz, #sc-202719), then lysed with Laemmli Sample buffer containing 2 % SDS, 4 % glycerol, 40 mmol L$^{-1}$ Tris-Cl (pH 6.8), 5 % 2-Mercaptoethanol, 0.01 % bromophenol blue, 1 mmol L$^{-1}$ NEM, 1 mmol L$^{-1}$ Phenylmethylsulfonyl fluoride (PMSF), protease inhibitor mixture (Roche, #COEDTAF-RO), and phosphatase inhibitor mixture (Roche, #04906837001). Samples were bath-sonicated (3 pulses 30 s ON 30 s OFF) and boiled for 10 min. Sample concentration was relativized by Coomassie Brilliant Blue staining. For separation, samples were loaded onto 12 % polyacrylamide gels and subjected to electrophoresis. Protein was transferred to nitrocellulose membranes, which were blocked with 2 % BSA in Tris-buffered saline (TBS) or 5 % non-fat milk in TBS. Membranes were incubated with primary antibodies overnight at 4 °C, followed by incubation with secondary antibodies conjugated with either horseradish peroxidase or fluorescent dye. DNA damaging agents were MMC, cisplatin (Santa Cruz, #sc-200896) and Hydroxyurea (Santa Cruz, #sc-29061) Proteins were visualized using ECL prime chemiluminescence reagent or fluorescence emission. Primary antibodies used were p53 (CST, #9282, 1:1000 dilution), phospho-p53 (CST, #9284, 1:1000 dilution), p21 (CST, #2947, 1:1000 dilution), phospho-histone H2A.X (CST, #9718, 1:1000 dilution), phospho-CHK1 (CST, #2348, 1:500 dilution), phospho-CHK2 (CST, #2661, 1:500 dilution), Vinculin (Santa Cruz, #sc-73614, 1:2000 dilution), alpha-tubulin (CST, #2144, 1:2000 dilution), and beta-actin (Santa Cruz, #sc-47778, 1:2000 dilution). Secondary antibodies used were horseradish peroxidase-linked anti-rabbit (CST, ##7074, 1:5000 dilution), horseradish peroxidase-linked anti-mouse (CST, #7076, 1:5000 dilution), DyLight-800 4× PEG-linked anti-rabbit (CST, #5151, 1:5000 dilution), and DyLight 680-linked anti-mouse (CST, #5470, 1:5000 dilution).

**Cell cycle assay and apoptosis determination.** Cells were plated at a final concentration of $3 \times 10^5$ cells per well in DMEM supplemented with 10 % FBS. After 24 h, cells were treated with 0, 60, and 150 µmol L$^{-1}$ FA for 24 h. After this period, cells were harvested by trypsinization and pelleted by centrifugation (5 min, 1000 × g). Cells were washed with cold PBS and then fixed with 70 % cold ethanol for 15 min on ice. Cells were washed twice with PBS and treated with 30 µg ribonuclease A and 15 µg of propidium iodide. Cells were run on a BD FACS Canto II flow cytometer and the data were analyzed with FlowJo 10.0.7 (Tree Star). The BD PE Annexin V Apoptosis Detection Kit was used for apoptosis determination (BD Pharmigen, #579563). Briefly, cells were plated and 24 h later exposed to the indicated concentrations of FA. Twenty-four hours later, cells were lifted, washed with cold PBS, and stained with PE-Annexin V antibody and 7-AAD. Samples were measured on a BD FACSAria II flow cytometer and data analysed with FlowJo 10.0.7 (Tree Star).

**GSH measurement.** GSH was determined using the GSH-Glo™ Glutathione Assay (Promega, #V6911). Briefly, 10.000 cells per well were seeded in a 96-well plate. A duplicated plate was prepared to determine viability. 48 h later GSH was determined following the instructions provided in the kit. In parallel, the viability was scored using resazurin, and the results adjusted for the percentage of viable cells relative to the GSH content of WT cells.

**Synthesis of S-hydroxymethyl-glutathione.** Reaction:

Procedure:

In a 50 mL-bottom flask, FA solution (12 µL, 37 wt. % in water, Sigma-Aldrich, #F8775) and HCl (0.5 mL, 36.5–38.0 %) were dissolved in 2 mL of dioxane (Sintorgan, #SIN-083003-63). The mixture was stirred at room temperature over 5 min and glutathione (Santa Cruz, #sc-29094) (50 mg, 0.16 mmol) was added in small portions. After stirring at room temperature over 48 h, the mixture was neutralized with saturated aqueous NaHCO$_3$ solution and partitioned between ethyl acetate and water. The aqueous phase was lyophilized (0.03 mBar, −80 °C, 72 h) to obtain a white solid using a Telstar LYOQuest-85 freeze dryer (Telstar, Madrid, Spain). A portion of 10 mg of the solid was resuspended in 1 mL of a CH$_3$OH:CH$_3$CN (1:1) mixture, centrifuged and the supernatant was diluted to be analysed by UPLC-HRMS. Estimated reaction yield 95.8 %[67].

*Sample preparation for UPLC-HRMS analysis.* HCT116 WT and ΔADH5 cells were counted and cultured in 100 cm plates at $1 \times 10^6$ cells/plate or $1.25 \times 10^6$ cells/plate for FA-treated samples. Two independent rounds of sample preparation were carried out in consecutive weeks. Six plates in the first week and five plates in the second week for each cell line were set up and allowed to grow for 24 h, then FA was subsequently added to the FA-treated samples that were allowed to grow for additional 48 h. One plate in each round was used for protein and cell count. Once 80 % confluence was reached, cells were gently washed with 5 mL of a 0.9 % NaCl aqueous solution at 0 °C. Subsequently, enzymatic activity was quenched by adding liquid N$_2$. Cells were scrapped immediately after with 1.4 mL of a cold (0 °C) CH$_3$OH:CH$_3$CN (50:50 v/v) solution and subsequently frozen using liquid N$_2$. After one freeze-thaw cycle, samples were vortex-mixed during 30 s and centrifuged at $5000 \times g$ for 5 min at 4 °C. Supernatants were collected and stored at −20 °C for 2 h and subsequently centrifuged at $15,000 \times g$ for 10 min at 4 °C. Afterwards, 1.4 mL of ultrapure water was added to supernatants and these solutions were immediately frozen and stored at −80 °C until lyophilization.

Process blanks consisting of incubating culture media in plates without cells were generated in parallel with samples, and followed the same protocol described above. For protein and cell count, cells were lifted and counted using trypan blue as viability marker. Afterwards, cells were lysed in a solution containing 1 µmol L$^{-1}$ EDTA; 10 µmol L$^{-1}$ Tris pH 8; 200 µmol L$^{-1}$ NaCl and 0.2 % Triton, and total protein was determined by the Bradford assay using BSA as standard. Samples were lyophilized at −80 °C and 50 mTorr for 48 h using a Telstar LYOQuest-85 freeze dryer (Telstar, Madrid, Spain) and stored at −80 °C until analysis by UPLC-HRMS. All sample residues from each batch were reconstituted the same day in a water: methanol (90/10 v/v) solution. Reconstitution factors were selected to reach the same protein content for all samples. After reconstitution, samples were vortex-mixed for 30 s and centrifuged at $21382 \times g$ for 20 min and 4 °C. Supernatants were stored until use at −80 °C. Quality control (QC) samples were prepared by pooling an aliquot of 15 µL from each sample, vortex-mixed for 30 s, split into four micro tubes, and stored at −80 °C until use for analysis.

A pooled QC sample spiked with GSH (14.3 µmol L$^{-1}$), GSSG (15.5 µmol L$^{-1}$), and HSMGSH (20 µmol L$^{-1}$) was used to verify the stability of retention times, peak shapes and areas during the analysis.

*UPLC-HRMS analysis.* UPLC-HRMS analyses were performed using a Waters ACQUITY UPLC I Class system fitted with a Waters ACQUITY UPLC BEH C$_{18}$ column (2.1 × 100 mm, 1.7 µm particle size, Waters Corporation, Milford, MA, USA, catalogue #186002352), and coupled to a Xevo G2S QTOF mass spectrometer (Waters Corporation, Manchester, UK, SN: YDA 375) with an electrospray ionization (ESI) source operated in ESI positive ionization mode. The typical resolving power and mass accuracy of the Xevo G2S QTOF mass spectrometer were 32,000 FWHM and 0.3 ppm at $m/z$ 556.2771, respectively. The mobile phase consisted of water with 0.1% formic acid (Fisher Chemical, #F/1900/PB15) (mobile phase A) and methanol (Fisher Chemical A454-4, (UN 1230-CL3)) (mobile phase B). The flow rate was constant at 400 µL min$^{-1}$, the elution gradient was set as follows: 0–1.6 min 0–0 % B; 1.6–2 min 0–20 % B; 2–6 min 20–70 % B; 6–7 min 70–70 % B; 7–14 min 70–90 % B; 14–17.5 min 90–90 % B; 17.5–18 min 90–95 % B; 18–21 min 95–95 % B. After each sample injection, the gradient was returned to its initial conditions in 9 min (total run time was 30 min). The eluates from the analytical column were diverted by automatically switching the valve to waste, except for the elution window from 0 to 8 min. The column and autosampler tray temperatures were set at 35 and 5 °C, respectively. The injection volume was 2 µL.

A solvent blank, which consisted of a water: methanol (90:10 v/v) solution, and a process blank were analysed at the beginning and end of each batch. Samples were randomly analysed within a defined template of spiked QC samples, and the analysis order was balanced based on sample classes. QC samples were used to condition the UPLC-HRMS system before sample analysis. A total of 30 randomized samples (WT $n = 10$, ΔADH5 $n = 10$, WT + 150 µmol L$^{-1}$ FA $n = 5$, ΔADH5 + FA $n = 5$) were analysed along 3 consecutive days. UPLC–HRMS sample lists were set up as follows (sample type (technical replicates)): zero consisting of mobile phase analysis without injection (1); solvent blank (2); process blank (2); QC samples (5); spiked QC sample (1); randomized, and balanced samples (12) with 1 spiked QC sample analysed every 4 samples; spiked QC sample (1); process blank (2); solvent blank (1).

The mass spectrometer was operated in positive ion mode with a probe capillary voltage of 2.5 kV and a sampling cone voltage of 30.0 V. The source and desolvation gas temperatures were set to 120 and 300 °C, respectively. The nitrogen gas desolvation flow rate was 600 L h$^{-1}$, and the cone desolvation flow rate was 10 L h$^{-1}$. The mass spectrometer was daily calibrated across the range of $m/z$ 50-1200 using a 0.5 mmol L-1 sodium formate solution prepared in 2-propanol/water (90:10 v/v). Data were drift corrected during acquisition using a leucine encephalin ($m/z$ 556.2771) reference spray (Waters cop, #700008842) infused at 5 µl min$^{-1}$, every 45 s. Data were acquired in MS continuum mode in the range of $m/z$ 50–1200, and the scan time was set to 0.5 s.

Principal component analysis (PCA) was conducted using MATLAB R2015a (The MathWorks, Natick, MA, USA) with the PLS Toolbox version 8.1 (Eigenvector Research, Inc., Manson, WA, USA). PCA was used to track data quality and to identify and remove outliers in the dataset. Two samples were identified as outliers by PCA, one from WT and one from ΔADH5 cells, and were not further considered for data analysis.

For UPLC-MS/MS experiments, the product ion mass spectra were acquired with collision cell voltages between 10 and 30 V, depending on the analyte. Ultra-high-purity argon (≥ 99.999 %) was used as the collision gas. Data acquisition and processing were carried out using MassLynx version 4.1 (Waters Corp., Milford, MA, USA).

Chemical standards were prepared in ultrapure water and were analysed under identical conditions as samples to validate metabolite identities by chromatographic retention time and MS/MS fragmentation pattern matching. Spiking experiments were also conducted with the authentic chemical standards on samples to address retention time differences caused by matrix effects.

Two different normalization strategies were independently used for sample analysis: data were normalized by the number of viable cells or by protein content.

*Quantification and statistical analysis.* Prism software package (GraphPad Software 7) was used for statistical analysis with the level of significance of 0.05 (95 % confidence). Additional information about statistical tests, sample number, and $P$-values are described in figure legends. The outcome of statistical analysis for relevant experiment and figures is included in the Source Data File. Unless otherwise stated, experiments were done using technical replicates (2 or 3 wells per condition) and repeated the $n$ times described in the figure legends with each symbol in a bar plot representing the average of the technical replicates for a given biological sample.

**Reporting summary**. Further information on research design is available in the Nature Research Reporting Summary linked to this article.

## Data availability

The main data supporting the findings of this study are available within the article and as Supplementary Figures. Uncropped/unprocessed blot images (Fig. 1b, f; Supplementary Figs. 1b, 2c, d, 5d) and source data are provided in the Source Data file. A reporting summary for this article is available as Supplementary Information file. Additional details on datasets and protocols that support the findings of this study will be made available upon reasonable request by the corresponding author. Reagents generated in this study can also be obtained upon request to L.B.P. Source data are provided with this paper.

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

## Acknowledgements

This work was supported by CONICET (PUE 2016 22920160100010CO), FOCEM MERCOSUR (COF 03/11), ANPyCT (PICT-PRH 2017-4668 and PICT-PRH 2015-0022). L.B.P. is also supported by MPI for Metabolism Research (Cologne, Germany) and MPI for Biophysical Chemistry (Göttingen, Germany). B.S. acknowledges

funding from the Deutsche Forschungsgemeinschaft (DFG, German Research Foundation, SCHU 2494/3-1, SCHU 2494/7-1, SCHU 2494/10-1, SCHU 2494/11-1, SFB 829, KFO 286, KFO 329, GRK 2407) and the Deutsche Krebshilfe (70112899). N.S. and O.P. are funded by DFG—Project number 268555672—SFB 1213, Project A06. G.C. is funded by Medical Research Council as part of UK Research and Innovation (file reference no. MC_UP_1201/18). We thank Eva Wiesner, Laetitia Graeser, Lisa Heidbüchel, and Uyen Pham for general support with *C. elegans* experiments, to Alejandra Atorressi and Cora Pollack for help with FACS and microscopy experiments. Also, to IBioBA staff for general support, to Ricardo Biondi, Alejandro Leroux, and Carolina Perez Castro for the critical reading of the manuscript. L.B.P. would like to thank Prof. Manel Esteller (Josep Carreras Leukaemia Research Institute, Barcelona, Spain) for hosting him in his lab during the COVID-19 pandemic. C.U., A.E.M., H.R., M.M. and A.F. are CONICET fellows. M.E.M., M.R.M., M.B. and L.B.P. are Research Staff Members from CONICET.

## Author contributions

C.U., A.E.M., M.S. and H.R. carried out cellular experiments assisted by L.B.P. M.R., K.K. and B.S. contributed with *C. elegans* data generation and analysis. M.R.M. and M.E.M. designed the sample preparation protocol for metabolite extraction, developed the UPLC-QTOF-MS-based method, and performed data analysis. M.R.M. conducted UPLC-HRMS experiments. G.A.F. and M.B. synthesized HSMGSH. O.P. and N.S. performed ESR spectroscopy. G.C. contributed with experiments in DNA repair-deficient cell lines. L.B.P. conceived the work and wrote the paper. All the authors revised the manuscript.

## Competing interests

The authors declare no competing interests.

## Additional information

**Peer review information** *Nature Communications* thanks Wei Du, Xiaoqing Tang, and the other anonymous reviewer(s) for their contribution to the peer review this work. Peer reviewer reports are available.

