## [Peer Review File · Nature Communications]

Reviewers' Comments:

Reviewer #1:

Remarks to the Author:

Formaldehyde (FA) is known to be highly reactive and to have cytotoxic and genotoxic effects. In addition to environmental sources, FA is produced inside cells during biochemical processes. FA reacts with GSH yielding the covalent product S-hydroxymethyl-GSH (HSMGSH). ADH5 is known to detoxify FA by metabolizing HSMGSH, yielding less toxic formate and during which reduced GSH is released back to GSH pool. This manuscript sought to test the sensible hypothesis that the reaction between FA and GSH affects endogenous GSH levels and redox balance, which, by triggering oxidative stress, accounts for part of FA toxicity. On the other hand, supplying GSH-precursors may help to limit FA toxicity, especially in Fanconi Anemia disease which results from impaired response to DNA damage. It also demonstrates that the function of ADH5 to help maintain the cellular GSH balance is conserved.

The key findings presented in favor of the hypothesis include: 1) Loss of ADH5 lowers GSH levels and sensitizes cells to FA cytotoxicity; 2) Exogenous glutathione precursors protect against FA-induced cell death and on the other hand inhibiting GSH production decreases the tolerance to FA cytotoxicity; 3) Worm mutants lacking H24K24.3 (the ortholog of the human ADH5 enzyme) show hypersensitivities to FA and paraquat, and NAC can restore the survival of the mutant.

However, I am not fully convinced by the conclusion drawn by the author, i.e., FA can cause cytotoxicity by triggering oxidative stress. To me, it is credibly shown that FA toxicity and tolerance are sensitive to cellular GSH content, but the argument for the underlying mechanism being oxidative stress is not well supported. The data suggesting an elevation of ROS level after FA treatment is not conclusive because it was measured on the cells treated with toxic FA doses. In addition, according to the data reported (fig. 3a-d), in wild-type cells the overall ROS level induced by FA treatment was lower than that by exposure to the glutathione (GSH) synthesis inhibitor L-BSO, but L-BSO at the test dose of 100uM does not affect cell viability (fig. S2a). In Δ ADH5 cells, it was shown that the overall ROS level is similar after treatment with FA and BSO, but at their respective test doses BSO was not toxic whereas FA shows marked toxicity. Furthermore, as shown in fig. 5g, very little GSH was detected after treatment with 100uM L-BSO, but as said above, no toxicity was observed. These call into question whether the possible effect of FA on GSH and the cellular redox state is sufficient to induce cell death. So, to better support their conclusion, convincing demonstrations regarding the induction of an oxidative stress level by FA that is detrimental to cell viability should be given, e.g., by quantifying the effects on oxidative stress markers and sensitivities to ROS inducers.

Specific comments:

- Have the authors checked whether and to what extent GSH level and GSH/GSSG ratio are affected by treatment with a toxic dose of FA? It will be an obvious question for people reading this paper.
- The rationale for mainly using HCT116 human colorectal carcinoma cells in the study is not clear. ADH5 KO mice are essentially wild type in phenotype. But combined deletion of Adh5 and Fancd2 (a DNA crosslink repair protein) was shown to lead to bone marrow failure and liver and renal dysfunction.
- PAGE 4 line 84-85: the figure citation is incorrect in the sentence that read "ADH5-deficient cells were not able to form tumor-spheroids in presence of FA, and they became sensitive to levels of FA near to those present in human blood (Extended Data Fig. 1c,d).
- Fig. 4C shows that inactivation of Gclm does not further exacerbate FA toxicity on Δ ADH5 HCT116 cells which was interpreted to indicate that ADH5 is the dominant factor in protecting HCT116 cells against FA. I am not convinced by this reasoning. According to fig. 5g, inactivation of Gclm does not further decrease GSH content in Δ ADH5 cells either. So, the data in Fig. 4C is consistent with the idea that FA toxicity and tolerance are closely linked to intracellular GSH levels.
- PAGE 10, line 235-236: the figure citation is incorrect in the sentence that reads: "At physiological FA concentrations as they occur in human blood, ADH5 is essential for cellular growth viability (Fig. 1c,d, 2c,d)".
- The authors acknowledge that since the total amounts of GSH and HSMGSH were lower in

Δ ADH5 cells, a role of efflux could not be ruled out. I wonder if the authors might find it relevant to cite Tulpule et al (<https://doi.org/10.1111/jnc.12170>) as well as other papers from these authors as they directly examine the effect of FA on GSH export from cultured neurons

Reviewer #2:

Remarks to the Author:

The manuscript by Umansky et al. aims at demonstrating that formaldehyde (FA) results in toxicity because of its capability to react with glutathione (GSH), which, in turn, leads to ROS accumulation and detrimental oxidative stress. They also suggest that this phenomenon is in part due to p53 activation, but does not apparently involve DNA damage as gamma-H2A.X are not modulated by the genetic ablation of FA-scavenging enzyme alcohol dehydrogenase class III (ADH5), and chromosomes do not show any significant aberration or structural damage. Pre-loading with sulfhydryl-containing molecules (i.e., NAC and GSH methyl ester, GSH-MEE), but not the ROS scavenger Trolox, partially rescues normal phenotype (i.e., prevents from cell death and reduces the disruption of tumor spheroids). Consistently, the irreversible inhibition of the rate limiting enzyme for GSH de novo synthesis, gamma-glutamyl cysteine synthetase (or glutamate:cysteine ligase, GCL), by buthionine-S,R-sulfoximine (BSO) recapitulates ADH5-null conditions in WT cells. Genetic ablation of the modulatory subunit of GCL (GCLM) show similar results with respect to those obtained with BSO, confirming the role of GSH in FA-mediated toxicity. Finally, adh-null worms show reduced life span when exposed to toxic concentrations of FA. Coherently, L1 larvae exposed to low doses of FA combined with the mitochondrial toxin paraquat (well documented ROS-producer) show defects in development.

GENERAL EVALUATION

Most of the findings of this study (e.g., those arguing for FA toxicity in ADH5-null experimental systems) are not innovative, as already reported elsewhere (Nat. Struct. Mol. Biol. 2011, 18:1432; Mol Cell. 2015, 60:177). Others are not informative (e.g., those reporting on the formation of S-hydroxymethyl-GSH, HSM-GSH) as already known, or not reliable (e.g., those based on the use of BSO and NAC). Moreover, the manuscript shows several critical lacking points. Experimental settings are usually not appropriate. Conclusions are not supported by the results obtained. ONLY some examples are as follows:

MAJOR POINTS:

- Experimental details (e.g., concentrations of different chemicals used, times of treatment, cells employed) are frequently missing throughout the text and not described in the Figure legends. Unfortunately, they are also difficult to be found in the Materials and Methods section, in which the authors dedicated 3 pages to describe Mass Spec studies and analyses and few (not informative) rows for the other techniques or protocols.
- The authors found out that FA concentration in the blood is about 10 μ M. Based on this result, the authors treat two different cell types with FA at a concentration of 50-to-150 μ M, which is 5-15 times higher. Looking at the dose-response curves, 10 μ M (blood concentrations) is not toxic in WT, neither in Δ ADH5 cells, indicating that, physiologically, endogenous FA is not toxic. The authors do not explain why they used higher concentration than the physiological ones, neither convinced this Reviewer that the experimental setting selected (i.e., cells subjected to FA) can recapitulate what happens inside a tissue. Are the authors sure that circulating FA is internalized by the cells? Why should it happen... if FA is actually extruded? Did the authors measure the intracellular FA concentration upon treatment?
- Considering that GSH is about 1000 higher (10 mM), it is hard to imagine how GSH pool can be decreased by 10 or 50 μ M FA. In the introduction, the authors make a parallel with haemolytic anaemia. However – at least in the form of the disease related to defective erythrocyte metabolism – GSH depletion and resulting oxidative stress are due to genetic defects (e.g., G6PDH mutations that impede NADP⁺-to-NADPH reduction) that affect GSH regeneration, this being lethal when hydrogen peroxide is physiologically over-produced in an environment, like red blood cells, where oxygen can be partially reduced in contact with heme-Fe of Hb. However, in principle, GSH de novo synthesis (catalysed by GCL and GS) and GSSG reduction (catalysed by glutathione disulphide reductase, GSR) are very efficient processes (the latter in particular) and can guarantee the maintenance of proper GSH levels even upon exposure to hydrogen peroxide and any other pro-oxidant/alkylating compound. In Δ ADH5 model, GSH de novo synthesis and GSSG

reduction are not affected and it is not reasonable how a single treatment of FA at a concentration of 10 (or even 150) μM can significantly alter GSH intracellular redox state. Namely it is not convincing how this condition can induce (after 3 days!) a decrease of viability (or tumor sphere formation capability, or any other phenotype related to cell healthy state).

- P.4, ll.97,98. Why do the authors say "inactivating p53 significantly suppressed the severe colony-forming phenotype detected in deltaADH5 cells"? DeltaADH5 cells do not efficiently form tumor spheres, or at least not as big as the WT counterparts. Moreover, "severe" is meaningless in this context.
- P.5, l.100. The effect of p53 ablation in cells and in spheroids seems contradictory. How do the authors, based on two different – and not directly linked – assays (viability and colony-forming) conclude that FA-induced cell death is p53 dependent and independent?
- How do the authors say that DNA damage is not involved? They should first evaluate the activation of ATR/Chk1 pathway. They should also explain why FA does not induce DNA damage. A huge amount of literature indicates it does, instead.
- P.5, l.105. The authors say that they could not detect a significant induction of gammaH2A.X by blood FA levels. Actually, they never used 10 μM , as they also detected in the blood, but concentrations 5-15 times higher.
- How do the authors explain that 50 μM of extracellular FA can block the antioxidant capability of GSH, whose intracellular concentrations is about 10 mM? Actually, FA can alkylate any reactive cysteine in the cell. Did the authors measure the activity of GST, as well as the other enzymes related to GSH homeostasis (e.g., GSR, GCL) upon FA treatment? All these enzymes have cysteines that can be targeted by FA, this making FA a broad "anti-GSH" molecule. Anyway, although this hypothesis is verified, it would even increase my concerns on how this phenomenon is selective for GSH (and the enzymes involved in its de novo synthesis and reduction) or, by contrast, completely non-specific.
- It is not reported that BSO induces oxidative stress (i.e., hydrogen peroxide production) if cells are maintained in resting conditions, even if it is used at 1 mM. Actually, BSO has been also reported to be administered in vivo, without any toxic side effects.
- Fig. 3 f,g. The authors' aim is to demonstrate that NAC and GSH-MEE sustain GSH intracellular levels (and, in turn, rescue cell viability) upon FA challenge. If so, they should also explain how 500 μM NAC, or 1 mM GSH-MEE are more effective in counteracting FA toxicity than endogenous GSH (10 mM), which is 10-50 times more concentrated. They should also provide a reason why they provide NAC and GSH-MEE so low, whereas they are commonly used at 10-30 mM in in vitro settings.
- Along the same line of reasoning, it is completely unexpected the results obtained with Trolox. In Fig. 3 c,d, the authors provide evidence that hydrogen peroxide is produced and GSH is oxidized upon FA treatment or BSO incubation, the latter condition recapitulating FA-mediated effects. If hydrogen peroxide is the effector of FA-toxicity, Trolox (being an antioxidant) should be even more active than NAC and GSH-MEE.
- The authors never considered, if not in three lines in the Conclusions, that ADH5 shows also an S-nitrosoglutathione reductase (GSNOR) activity. Actually, it has been reported that ADH5 activity towards GSNO is much greater than towards HSM-GSH, and the enzyme exhibits a k_{cat}/KM for GSNO close to the diffusion-limited rate. It is not a surprise, indeed, that thiol-containing molecules (NAC and GSH-MEE), which are also denitrosylating compounds able to complement GSNOR lack, counteract FA toxicity, whereas Trolox does not. Did the authors take into consideration the possibility that: i) FA induces NOS or NO generations? ii) GSNO is then formed? iii) GSNOR, more than ADH activity plays a role in this system?
- Consistently, it has been recently reported that impairment of GSNOR activity is related to aging because of mitochondrial alterations. Did the authors check for any mitochondrial damage occurring in their conditions?
- Fig. 3g, the column referring to cells treated with the combination FA/NAC/BSO show values which are clustered into two completely separated (very distant) groups, and statistical significance is not convincing. The authors should explain how s.e.m is so reduced and p is < 0.0001 .
- Fig. 4a and P.6, ll.155-160. How do the authors say that ADH5 and GCLM do not further affect viability if they never show a dose response effect for BSO? With the experiments shown, the authors cannot say that ADH5 is the dominant factor in protecting HCT116 cells against FA. Moreover, how do the authors reconcile the different effects of deltaGCL and BSO treatment?
- P.6, ll.165,166. It is chemically impossible that FA can limit GSH antioxidant properties in a ratio

of 1:1000.

- The formation of HSM-GSH has been known for long time. No need to be detected.
- P.8, l.178. DeltaADH5 cells contain 17.9% less reduced GSH than WT counterparts, which means about 8.5 mM instead of more than 10 mM. Do not the authors think these levels are still effective in keeping the right intracellular redox homeostasis? How do they explain those phenotypes observed in Fig. 1-3?
- P.9, l. 212. 10 uM NAC in worms could not be enough as previously argued in cells

MINOR POINTS:

- Figures 1C and D are never mentioned in the text
- P.4, l.85. What does "...prevented the early apoptotic markers Annexin V" mean? Annexin V is not an apoptotic marker. The early apoptotic marker is phosphatidylserine (PS) external translocation. Since Annexin V binds to PS, a FITC-conjugated form of this protein has been developed to measure the extent of apoptosis. Moreover, "prevention of a marker" is not appropriate to describe what the authors are looking at.
- Extended Figs. 1e,f do not refer to what is described in the text.
- Fig. 2f is not convincing. Images should be magnified to state that there is no chromosomal damage. Moreover, the authors should explain discrepancies with Posado et al., and the same Pontel et al. (Nat. Struct.Mol. Biol. and Mol. Cell, respectively).
- P.5, l.119-121. DCF fluorescence is a measure of hydrogen peroxide and, to some extent, of other ROS. It is not a measure of cellular oxidative status as the authors stated.
- P.7 l.152. BSO was never reported to have side-effects. BSO structure resembles gamma-glutamylcysteine. It binds to GCL as an irreversible inhibitor.
- Fig. 5a is not mentioned in the text.
- The authors should also show Western blot against p53 in Extended Fig. 1e.

Reviewer #3:

Remarks to the Author:

Manuscript Title: "Endogenous formaldehyde scavenges cellular glutathione resulting in cytotoxic redox disruption"

General comments: In the study by Umansky et al. entitled "Endogenous formaldehyde scavenges cellular glutathione resulting in cytotoxic redox disruption", the authors showed that formaldehyde (FA) causes cellular damage beyond genotoxicity by triggering oxidative stress, which is prevented by the enzyme alcohol dehydrogenase 5 (ADH5/GSNOR). Mechanistically, they demonstrated that endogenous FA reacts with the redox-active thiol group of glutathione (GSH) forming S-hydroxymethyl-GSH, which is metabolized by ADH5 yielding reduced GSH thus preventing redox disruption. They also identified the ADH5-ortholog gene in *Caenorhabditis elegans* and showed that oxidative stress also underlies FA toxicity in nematodes. In addition, they found that endogenous GSH protects cells lacking the Fanconi Anemia DNA repair pathway from FA. Therefore, they concluded that their findings establish a highly conserved mechanism through which endogenous FA disrupts the GSH-regulated cellular redox homeostasis that is critical during development and aging. Overall, this is an interesting paper which describes endogenous formaldehyde scavenges cellular glutathione resulting in cytotoxic redox disruption. However, there are several weaknesses, if fixed, will enhance the significance of the study.

Specific comments:

What is the rationale for measuring blood FA levels in mice? There is not in vivo experiments throughout the paper. Somehow, that part is disconnected from the whole story.

Is the observation specific to cancer cells but not normal cells? Normal cell line controls are preferred. In addition, most of the experiments except Fig 4, which also used Nalm6 cells (an ALL cell line), were done in HCT116 cancer cell lines. To make solid conclusion, other cancer cell lines are needed.

The authors attempted to make the notion that their findings are "highly conserved". However, they were only able to show the similar results in *C. elegans*. More evidence is needed to make the

conclusion.

The authors also emphasized that endogenous GSH can protect cells lacking the Fanconi anemia DNA repair pathway from FA, which might have broad implications for Fanconi anemia patients and for healthy BRCA2-mutation carriers. However, they only proved their notion in Δ FANCB cells, which only represents a very small portion of Fanconi patients. Other Fanconi anemia deficient cell lines are needed.

Minor:

2-segments Y-axis in Fig 2d should be re-arranged.

Quantification for Fig 3b is needed.

Dear Reviewers,

We acknowledge your helpful feedback, which has allowed us to truly improve our manuscript. This revised version includes results from additional experiments aimed to support our mechanism of FA-induced oxidative cytotoxicity when DNA repair fully operates. All the comments have been addressed and results from these additional experiments have been incorporated in the manuscript as new figures and also in the supplementary information. We have also modified typos and confusing sentences, overall addressing all your concerns in full.

Our point-by-point responses are listed below, and the changes made to the manuscript presented in blue letters.

Best regards,

Lucas.

Reviewers' comments:

Reviewer #1 (Remarks to the Author):

Formaldehyde (FA) is known to be highly reactive and to have cytotoxic and genotoxic effects. In addition to environmental sources, FA is produced inside cells during biochemical processes. FA reacts with GSH yielding the covalent product S-hydroxymethyl-GSH (HSMGSH). ADH5 is known to detoxify FA by metabolizing HSMGSH, yielding less toxic formate and during which reduced GSH is released back to GSH pool. This manuscript sought to test the sensible hypothesis that the reaction between FA and GSH affects endogenous GSH levels and redox balance, which, by triggering oxidative stress, accounts for part of FA toxicity. On the other hand, supplying GSH-precursors may help to limit FA toxicity, especially in Fanconi Anemia disease which results from impaired response to DNA damage. It also demonstrates that the function of ADH5 to help maintain the cellular GSH balance is conserved.

Our reply: We would like to thank R1 for recognizing the importance of our work.

The key findings presented in favor of the hypothesis include: 1) Loss of ADH5 lowers GSH levels and sensitizes cells to FA cytotoxicity; 2) Exogenous glutathione precursors protect against FA-induced cell death and on the other hand inhibiting GSH production decreases the tolerance to FA cytotoxicity; 3) Worm mutants lacking H24K24.3 (the ortholog of the human ADH5 enzyme) show hypersensitivities to FA and paraquat, and NAC can restore the survival of the mutant.

However, I am not fully convinced by the conclusion drawn by the author, i.e., FA can cause cytotoxicity by triggering oxidative stress. To me, it is credibly shown that FA toxicity and tolerance are sensitive to cellular GSH content, but the argument for the underlying mechanism being oxidative stress is not well supported. The data suggesting an elevation of ROS level after FA treatment is not conclusive because it was measured on the cells treated with toxic FA doses.

Our reply: Thanks for raising this concern. To further support that formaldehyde increases ROS levels we have now measured ROS by H2DCFDA at a non-toxic formaldehyde concentration (60 μ M) (Fig. 3a) and provided additional data supporting that formaldehyde leads to GSH:GSSG imbalance and oxidative stress (see below).

In addition, according to the data reported (fig. 3a-d), in wild-type cells the overall ROS level induced by FA treatment was lower than that by exposure to the glutathione (GSH) synthesis inhibitor L-BSO, but L-BSO at the test dose of 100 μ M does not affect cell viability (fig. S2a). In Δ ADH5 cells, it was shown that the overall ROS level is similar after treatment with FA and BSO, but at their respective test doses BSO was not toxic whereas FA shows marked toxicity. Furthermore, as shown in fig. 5g, very little GSH was detected after treatment with 100 μ M L-BSO, but as said above, no toxicity was observed. These call into question whether the possible effect of FA on GSH and the cellular redox state is sufficient to induce cell death.

Our reply: In Fig. 3a,b we used L-BSO (GSH synthesis inhibitor) as positive control of ROS induction in HCT116 cells. L-BSO causes an almost complete depletion of GSH (Extended Data Fig. 6k), which is likely to activate backup mechanisms. Indeed, in two recent reports (Harris, et al Cancer Cell 2015, and Harris, et al Cell Metabolism 2019) it was shown that some cancer cells overcome GSH depletion toxicity by expressing alternative antioxidant and protein-clearance mechanisms, which might explain the lack of L-BSO toxicity in HCT116 cells. We think it is unlikely that micromolar levels of formaldehyde concentrations will be able to fully deplete cellular GSH concentration, which is around 10 mM. However, a partial reduction of GSH pool would be sufficient to imbalance GSH:GSSG ratio, leading to redox homeostasis disruption, which can be deleterious as previously shown (Romero-Aristizabal, et al Nat Comm 2014) (Fig. 6i). Remarkably, the GSSG level was reported to be 200 nM (Morgan, et al, NCB 2013), thus the ratio GSH:GSSG (and EGSH) is expected to be very sensitive to small changes in GSH and GSSG endogenous levels. We have included the following sentences to clarify this point:

1- lines 130-131 "the abundance of GSH (1-10 mmol L⁻¹) might favor the spontaneous reaction between GSH and FA, which, if not limited, could alter the GSH:GSSG balance leading to oxidative stress"

2- lines 223-224 "The GSH:GSSG ratio was further affected by FA in Δ ADH5 but not in WT cells (Fig. 6i)"

Additionally, to address the reviewer's concern, we have:

a. Measured a dose-response curve of the toxicity of HCT116 and Nalm6 cells exposed to L-BSO (Extended Data Fig. 4a). HCT116 cells are completely resistant to GSH depletion, even to millimolar levels of the inhibitor, whereas Nalm6 cells are very sensitive to L-BSO. However, both cell lines are sensitive to formaldehyde (Fig. 1b,c,f), the toxicity is rescued by NAC (Fig. 4a,b) and it is exacerbated by GSH depletion using L-BSO (Fig. 5a,b). Moreover, GSH depletion exacerbates formaldehyde

toxicity in HepG2, HAP1 and chicken DT40 cell lines (**Extended Data Fig. 4c,d and Fig. 7a**). Overall supporting oxidative stress underlies formaldehyde cytotoxicity in our setting.

b. Determined GSH, GSSG and total GSH in cells exposed to formaldehyde by UPLC-HRMS (**Fig. 6f-h**), and measured the ratio GSH:GSSG upon formaldehyde exposure in WT and $\Delta ADH5$ cells (**Fig. 6i**), showing that the GSH:GSSG ratio is significantly affected by formaldehyde.

So, to better support their conclusion, convincing demonstrations regarding the induction of an oxidative stress level by FA that is detrimental to cell viability should be given, e.g., by quantifying the effects on oxidative stress markers and sensitivities to ROS inducers.

Our reply: We are providing new evidence supporting formaldehyde induces oxidative stress in human cells and in *C. elegans*:

a- We have determined that a mitochondrial GSH redox potential sensor (Grx1-roGFP2) is also induced upon formaldehyde exposure in HCT116 cells. Interestingly, this reporter is induced even in presence of ADH5, which is consistent with the cytoplasmic subcellular localization of ADH5, thus mitochondria would behave like an *ADH5*-deficient cell line (**Fig. 6k**).

b- We have measured two established oxidative stress markers in *C. elegans*: (i) *gst-4::GFP*, which is induced upon oxidative stress through the oxidative stress response transcription factor SKN-1 (the NRF2 ortholog), and (ii) HyPer, which is directly activated upon reaction with hydrogen peroxide via the formation of a disulfide bridge. We detected the induction of *Pgst-4::GFP* and HyPer, thus further supporting the generation of cytoplasmic ROS by formaldehyde, which leads to a SKN-1 (NRF2)-dependent response (**Fig. 3e,f, Extended Data Fig. 3a**).

c- We have moved formaldehyde plus paraquat (PQ) experiment to **Fig. 4h,i**, which shows that the ROS inducer PQ further exacerbates formaldehyde toxicity in *C. elegans*.

d. We have shown that the inhibition of GSH synthesis increases the toxicity of formaldehyde in human HepG2, HAP1, Nalm6, HCT116, and chicken DT40 cells (**Extended Data Fig. 4c,d; Fig. 5a,b; Fig. 7a,d,e**). We have also presented data in a cell line derived from a Fanconi Anemia patient (PD20), which showed a mild but significant decrease in the tolerance to formaldehyde in presence of L-BSO (**Fig. 7b,c**).

Specific comments:

- Have the authors checked whether and to what extent GSH level and GSH/GSSG ratio are affected by treatment with a toxic dose of FA? It will be an obvious question for people reading this paper.

Our reply: The revised manuscript includes results from measured GSH, GSSG and total GSH levels in WT and $\Delta ADH5$ cells exposed to formaldehyde (**Fig. 6f-h**), as well as the calculated GSH:GSSG ratio in cells exposed to formaldehyde (**Fig. 6i**). We detected a significant change in the GSH:GSSG ratio in *ADH5*-deficient cells exposed to formaldehyde. As discussed in the manuscript, formaldehyde might induce GSH efflux in $\Delta ADH5$ cells reducing its net levels. HSMGSH levels in $\Delta ADH5$ cells exposed to formaldehyde were below the detection limit of the UPLC-HRMS-based analytical method. This limitation prevented us from calculating the amount of HSMGSH relative to GSH in $\Delta ADH5$ cells exposed to formaldehyde. We are providing below the data for Wild type cells (not shown in the manuscript).

- The rationale for mainly using HCT116 human colorectal carcinoma cells in the study is not clear. *ADH5* KO mice are essentially wild type in phenotype. But combined deletion of *Adh5* and *Fancd2* (a DNA crosslink repair protein) was shown to lead to bone marrow failure and liver and renal dysfunction.

Our reply: *Adh5* $-/-$ *Fancd2* $-/-$ mice, indeed, presented a severe hematopoietic phenotype, and they also developed leukemia, hepatocellular carcinoma and widespread karyomegaly (Pontel, et al 2015, Mol Cell). We selected HCT116 -a non-blood cancer cell line- and Nalm6 -a leukemic-derived cell line, because they harbor wild type copies of p53, which in case of mutated or null might alter the cellular response to FA -a known DNA damaging metabolite. We have included a sentence explaining this selection and measured the formaldehyde toxicity in presence of L-BSO in HepG2 (hepatocellular carcinoma) and in HAP1 (chronic myeloid leukemia) (**Extended Data Fig. 4c,d**).

- PAGE 4 line 84-85: the figure citation is incorrect in the sentence that read “*ADH5*-deficient cells were not able to form tumor-spheroids in presence of FA, and they became sensitive to levels of FA near to those present in human blood (Extended Data Fig. 1c,d).

Our reply: We apologize for the mistake. This was accordingly modified in the revised manuscript.

- Fig. 4C shows that inactivation of *Gclm* does not further exacerbate FA toxicity on $\Delta ADH5$ HCT116 cells which was interpreted to indicate that *ADH5* is the dominant factor in protecting HCT116 cells against FA. I am not convinced by this reasoning. According to fig. 5g, inactivation of *Gclm* does not further decrease GSH content in $\Delta ADH5$ cells either. So, the data in Fig. 4C is consistent with the idea that FA toxicity and tolerance are closely linked to intracellular GSH levels.

Our reply: Based on the reviewer's concern, we have modified the text including the following sentence: "The simultaneous inactivation of *ADH5* and *GCLM* did not further affect viability compared to *ADH5*-single knockout cells (Fig. 5c). This observation indicates that *ADH5* is epistatic to *GCLM* in the formaldehyde viability assay performed in HCT116 cells".

- PAGE 10, line 235-236: the figure citation is incorrect in the sentence that reads: At physiological FA concentrations as they occur in human blood, *ADH5* is essential for cellular growth viability (Fig. 1c,d, 2c,d)".

Our reply: We have corrected this mistake in the revised manuscript.

- The authors acknowledge that since the total amounts of GSH and HSMGSH were lower in Δ *ADH5* cells, a role of efflux could not be ruled out. I wonder if the authors might find it relevant to cite Tulpule et al (<https://doi.org/10.1111/inc.12170>) as well as other papers from these authors as they directly examine the effect of FA on GSH export from cultured neurons

Our reply: We appreciate the reviewer's suggestion. We have included this citation in the revised manuscript.

Reviewer #2 (Remarks to the Author):

The manuscript by Umansky et al. aims at demonstrating that formaldehyde (FA) results in toxicity because of its capability to react with glutathione (GSH), which, in turn, leads to ROS accumulation and detrimental oxidative stress. They also suggest that this phenomenon is in part due to p53 activation, but does not apparently involve DNA damage as gamma-H2A.X are not modulated by the genetic ablation of FA-scavenging enzyme alcohol dehydrogenase class III (*ADH5*), and chromosomes do not show any significant aberration or structural damage. Pre-loading with sulfhydryl-containing molecules (i.e., NAC and GSH methyl ester, GSH-MEE), but not the ROS scavenger Trolox, partially rescues normal phenotype (i.e., prevents from cell death and reduces the disruption of tumor spheroids). Consistently, the irreversible inhibition of the rate limiting enzyme for GSH de novo synthesis, gamma-glutamyl cysteine synthetase (or glutamate:cysteine ligase, GCL), by buthionine-S,R-sulfoximine (BSO) recapitulates *ADH5*-null conditions in WT cells. Genetic ablation of the modulatory subunit of GCL (*GCLM*) show similar results with respect to those obtained with BSO, confirming the role of GSH in FA-mediated toxicity. Finally, *adh*-null worms show reduced life span when exposed to toxic concentrations of FA. Coherently, L1 larvae exposed to low doses of FA combined with the mitochondrial toxin paraquat (well documented ROS-producer) show defects in development.

Our reply: We acknowledge and see very positive the description of our results. However, this positive introduction seems contradictory to and is not consistent with the follow-up comments.

General evaluation (R2):

Most of the findings of this study (e.g., those arguing for FA toxicity in ADH5-null experimental systems) are not innovative, as already reported elsewhere (Nat. Struct. Mol. Biol. 2011, 18:1432; Mol Cell. 2015, 60:177). Others are not informative (e.g., those reporting on the formation of S-hydroxymethyl-GSH, HSM-GSH) as already known, or not reliable (e.g., those based on the use of BSO and NAC). Moreover, the manuscript shows several critical lacking points. Experimental settings are usually not appropriate. Conclusions are not supported by the results obtained.

Our reply:

- We agree on the fact that formaldehyde toxicity in *ADH5*-null systems was shown before, indeed we published this in 2015 (Pontel et al, Mol Cell). However, the main point of our work is to show **how formaldehyde toxicity is inflicted - revealing a novel toxic mechanism, and how ADH5 prevents it.**
- We disagree with the reviewer's claims regarding our results being not informative (S-hydroxymethyl-GSH). This is -to the best of our knowledge- **the first time this compound is detected in human cells** by ultraperformance liquid chromatography coupled to high resolution mass spectrometry (UPLC-HRMS), showing that cell metabolism can produce sufficient formaldehyde to react with GSH generating S-hydroxymethyl-GSH.
- R2 considers our results are not reliable because they are based on the chemical inhibitor BSO. We disagree with this comment because in fact **we have generated a mutant in the gene coding for GCLM -from the GSH synthetic pathway-, with the aim to genetically corroborate the findings obtained by using L-BSO.** Thus, both chemical and genetic inactivation data are provided to reliably support our findings.
- The final general negative statements are not in line with the reviewer's comments provided in the introductory paragraph. We have addressed below the specific points that were raised by this reviewer.

Major points (R2):

• *Experimental details (e.g., concentrations of different chemicals used, times of treatment, cells employed) are frequently missing throughout the text and not described in the Figure legends. Unfortunately, they are also difficult to be found in the Materials and Methods section, in which the authors dedicated 3 pages to describe Mass Spec studies and analyses and few (not informative) rows for the other techniques or protocols.*

Our reply: We strictly adhered to the high standards of the Nature Publishing Group to present our experimental procedures in great detail and comprehensively. We are happy to correct any specific mistakes that are pointed out by the reviewer as we constantly strive to improve our manuscript, but it is difficult to respond to such a general comment.

• *The authors found out that FA concentration in the blood is about 10 uM. Based on this result, the authors treat two different cell types with FA at a concentration of 50-to-150 uM, which is 5-15 times higher.*

Our reply: The experiments in Fig. 2b and 5e,f show HCT116 are affected by just 12.5 μ M formaldehyde. Moreover, preB-ALL Δ *ADH5* cells show significant formaldehyde sensitivity at a formaldehyde concentration of only 30 μ M (Fig. 1f).

On the other hand, we recognize that both R1 and R2 indicated that the level of blood formaldehyde levels in mice that we reported are not relevant to our work because they were measured in mice instead of human subjects. A recent report indicated human blood formaldehyde is around 45 μ M and can reach 90 μ M in patients with prostate cancer (Wei, Y, et al J. Chromatogr. B Anal. Technol. Biomed. Life Sci, 2019). We have included this information in the revised version of the manuscript and excluded data from the animal model, which was indeed generating confusion.

• *Looking at the dose-response curves, 10 uM (blood concentrations) is not toxic in WT, neither in deltaADH5 cells, indicating that, physiologically, endogenous FA is not toxic. The authors do not explain why they used higher concentration than the*

physiological ones, neither convinced this Reviewer that the experimental setting selected (i.e., cells subjected to FA) can recapitulate what happens inside a tissue. Are the authors sure that circulating FA is internalized by the cells? Why should it happen... if FA is actually extruded? Did the authors measure the intracellular FA concentration upon treatment?

Our reply: A formaldehyde concentration of 12.5 μM is sufficient to impair the formation of colonies in HCT116 cells lacking ADH5 (Fig. 2b and Fig. 5e,f), and Nalm6 cells lacking ADH5 presented a significantly less tolerance to formaldehyde at only 30 μM (Fig. 1f). In addition, R2 raised interesting questions that we are eager to address in the future. R2 also claimed that exposing cells to formaldehyde cannot recapitulate what happens inside cells. We indeed detected the product of formaldehyde and GSH in cells that were not exposed to formaldehyde (Fig. 6), and we presented evidence for formaldehyde cytotoxicity in *C. elegans* that further supports our proposed mechanism of action beyond human cancer cell lines. Moreover, we proposed the product of formaldehyde and GSH might be extruded, which was recognized by R1.

• Considering that GSH is about 1000 higher (10 mM), it is hard to imagine how GSH pool can be decreased by 10 or 50 μM FA. In the introduction, the authors make a parallel with haemolytic anaemia. However – at least in the form of the disease related to defective erythrocyte metabolism – GSH depletion and resulting oxidative stress are due to genetic defects (e.g., G6PDH mutations that impede NADP⁺-to-NADPH reduction) that affect GSH regeneration, this being lethal when hydrogen peroxide is physiologically over-produced in an environment, like red blood cells, where oxygen can be partially reduced in contact with heme-Fe of Hb. However, in principle, GSH de novo synthesis (catalysed by GCL and GS) and GSSG reduction (catalysed by glutathione disulphide reductase, GSR) are very efficient processes (the latter in particular) and can guarantee the maintenance of proper GSH levels even upon exposure to hydrogen peroxide and any other pro-oxidant/alkylating compound. In deltaADH5 model, GSH de novo synthesis and GSSG reduction are not affected and it is not reasonable how a single treatment of FA at a concentration of 10 (or even 150) μM can significantly alter GSH intracellular redox state. Namely it is not convincing how this condition can induce (after 3 days!) a decrease of viability (or tumor sphere formation capability, or any other phenotype related to cell healthy state).

Our reply: To address whether formaldehyde was affecting the intracellular redox state, we determined GSH and GSSG levels in samples from WT and ΔADH5 cells exposed to formaldehyde by means of UPLC-HRMS-based semi-targeted approach (Fig. 6f-i). Interestingly, in WT cells there was no significant change in GSH intracellular redox state (Fig. 6i), which however is significantly affected in absence of ADH5. The reversion of the death and 3D-sphere formation phenotypes by antioxidants NAC and GSH-MEE supports our hypothesis that an imbalance in GSH intracellular redox state underlies cytotoxicity. We have now provided further evidence showing the induction of the ROS-responsive gene *gst-4* and of the H₂O₂ sensor HyPer in *C. elegans* upon formaldehyde treatment (Fig 3e,f, Extended Data Fig. 3a), and of the mitochondrial GSH reporter Grx1-roGFP2 in cells (Fig. 6k).

• P.4, ll.97,98. Why do the authors say “inactivating p53 significantly suppressed the severe colony-forming phenotype detected in deltaADH5 cells”? DeltaADH5 cells do not efficiently form tumor spheres, or at least not as big as the WT counterparts. Moreover, “severe” is meaningless in this context.

Our reply: The colony survival assay (CSA) depicted in Fig. 2b shows that inactivating P53 restored the plate efficiency of ΔADH5 single knockout cells (compare red with yellow bars). Following the reviewer’s remark, we have replaced the word severe by significant.

• P.5, l.100. The effect of p53 ablation in cells and in spheroids seems contradictory. How do the authors, based on two different – and not directly linked – assays (viability and colony-forming) conclude that FA-induced cell death is p53 dependent and independent?

Our reply: We have not conducted experiments on spheroid formation in cells lacking P53. Thus, it is difficult to address the reviewer’s concern. In addition, the reviewer refers to viability vs colony-forming assays, which indeed measure different cell phenotypes. Colony-forming assays provide information about cell tolerance, adherence, cell-cell interactions, while an indirect readout of cell viability can be accomplished by measuring the ability of the cell to reduce resazurin. We interpretate that the different results are a consequence of different assays, revealing p53-dependent and independent effects. We have modified the text indicating “FA can impair cell growth by both p53-dependent and independent pathways” (line 107).

• How do the authors say that DNA damage is not involved? They should first evaluate the activation of ATR/Chk1 pathway. They should also explain why FA does not induce DNA damage. A huge amount of literature indicates it does, instead.

Our reply: We agree with the reviewer about the fact that formaldehyde can cause DNA damage. Indeed, we detected the phosphorylation of p53 at the Ser15, which is responsive to DNA damage. However, we did not detect the formation of DSB when measured indirectly by $\gamma\text{-H2Ax}$ nor a significant increase in broken chromosomes quantified by metaphase spreading in HCT116 cells (Fig 2c,d,f). We propose that there is no “lethal DNA damage” because likely cells with fully operating DNA repair systems prevent the formation of deleterious DSBs. To support our hypothesis, we have now included P-CHK1 and P-CHK2 in the supplementary information (Extended Data Fig. 2c,d). Furthermore, we have measured $\gamma\text{-H2Ax}$ in Nalm6 cells lacking *FANCB* - Fanconi Anemia DNA repair, detecting the induction of this marker upon formaldehyde, which indicates that in absence of DNA repair DSBs are generated by formaldehyde (Fig. 2e).

P.5, l.105. The authors say that they could not detect a significant induction of gammaH2A.X by blood FA levels. Actually, they never used 10 μM , as they also detected in the blood, but concentrations 5-15 times higher.

• How do the authors explain that 50 μM of extracellular FA can block the antioxidant capability of GSH, whose intracellular concentrations is about 10 mM? Actually, FA can alkylate any reactive cysteine in the cell. Did the authors measure the activity of GST, as well as the other enzymes related to GSH homeostasis (e.g., GSR, GCL) upon FA treatment? All these enzymes have cysteines that can be targeted by FA, this making FA a broad “anti-GSH” molecule. Anyway, although this hypothesis is verified, it would even increase my concerns on how this phenomenon is selective for GSH (and the enzymes involved in its de novo synthesis and reduction) or, by contrast, completely non-specific.

Our reply: The reviewer’s concern about blood formaldehyde levels was addressed above. We initially focused on GSH because it is one of the most abundant thiol-containing molecules in the cell and a putative target of formaldehyde reactivity. We agree with R2 that GSH might not be a unique thiol-containing compound affected by FA. Indeed, a recent report (Pietzke et al, Comm Chem 2020) shows cysteine is also affected by formaldehyde. We have incorporated this citation and discussed that GSH might not be a unique thiol target of formaldehyde in the revised version of the manuscript as follow “Remarkably, the strong electrophilicity of FA anticipates it might react with other abundant cellular thiols such as free cysteine and sulfhydryl groups in proteins, further affecting redox balance and metabolism” (Line 275-277).

• It is not reported that BSO induces oxidative stress (i.e., hydrogen peroxide production) if cells are maintained in resting conditions, even if it is used at 1 mM. Actually, BSO has been also reported to be administered *in vivo*, without any toxic side effects.

Our reply: We, respectfully, disagree with this statement of R2. BSO has been extensively used as a GSH synthesis inhibitor (since 1979), inducing ROS and oxidative stress, and it has been reported to be administered *in vivo* without major toxic side effects. Consistently, we have detected induction of ROS upon L-BSO treatment (Fig. 3a,b).

• Fig. 3 f,g. The authors' aim is to demonstrate that that NAC and GSH-MEE sustain GSH intracellular levels (and, in turn, rescue cell viability) upon FA challenge. If so, they should also explain how 500 μ M NAC, or 1 mM GSH-MEE are more effective in counteracting FA toxicity than endogenous GSH (10 mM), which is 10-50 times more concentrated. They should also provide a reason why they provide NAC and GSH-MEE so low, whereas they are commonly used at 10-30 mM in *in vitro* settings.

Our reply: Our experiments (Fig. 4b-d) indicate NAC is rescuing 3D-spheroid formation through supplying precursors for GSH synthesis. Moreover, we don't find any reason for using more NAC or GSH-MEE if we already detect a rescue by using a lower concentration.

• Along the same line of reasoning, it is completely unexpected the results obtained with Trolox. In Fig. 3 c,d, the authors provide evidence that hydrogen peroxide is produced and GSH is oxidized upon FA treatment or BSO incubation, the latter condition recapitulating FA-mediated effects. If hydrogen peroxide is the effector of FA-toxicity, Trolox (being an antioxidant) should be even more active than NAC and GSH-MEE.

Our reply: We interpreted that Trolox failed to rescue formaldehyde toxicity might be related to the fact that Trolox does not contain a thiol group. This is not the first time Trolox shows a different behavior than other widely used antioxidants such as NAC. In a recent report, it was shown that **NAC but not Trolox can rescue cell death** induced by the combination of L-BSO and MI2 or PR619, which are inhibitors of deubiquitinase (DUB) enzymes (see Harris et al Cell Metabolism 2019, Fig. 4g). The authors argue that L-BSO leads to oxidative damage of proteins that requires protein clearance (DUB). Thus, Trolox *per se* is not sufficient to rescue lethality. It would be interesting to understand whether DUB enzymes participate in formaldehyde tolerance.

• The authors never considered, if not in three lines in the Conclusions, that ADH5 shows also an S-nitrosogluthione reductase (GSNOR) activity. Actually, it has been reported that ADH5 activity towards GSNO is much greater than towards HSM-GSH, and the enzyme exhibits a kcat/KM for GSNO close to the diffusion-limited rate. It is not a surprise, indeed, that thiol-containing molecules (NAC and GSH-MEE), which are also denitrosylating compounds able to complement GSNOR lack, counteract FA toxicity, whereas Trolox does not. Did the authors take into consideration the possibility that: i) FA induces NOS or NO generations? ii) GSNO is then formed? iii) GSNOR, more than ADH activity plays a role in this system?

Our reply: ADH5/GSNOR is a complex enzyme and we understand that focusing on the nitroso-GSH (GSNO) metabolizing activity is beyond the scope of this work. However, we have discussed that both reactions (metabolization of HSMGSH and GSNO) generate a common product: reduced GSH, thus both reactions might contribute to GSH cellular redox balance in the same direction. We have cited a recent report showing that S-nitrosylation are not stable post-translational modifications but intermediates towards the formation of a disulfide covalent bond (Wolhuter et al Mol Cell, 2018) and expanded the discussion of GSNOR/ADH5 activities.

• Consistently, it has been recently reported that impairment of GSNOR activity is related to aging because of mitochondrial alterations. Did the authors check for any mitochondrial damage occurring in their conditions?

Our reply: We have measured mitochondrial membrane potential by JC1 (see below) and also mitochondrial E_{GSH} redox potential (Fig. 6k). As reported, ADH5-deficient cells present a lower mitochondrial potential than WT cell. However, we could not detect that formaldehyde further affects mitochondrial membrane potential. On the other hand, we found that formaldehyde increases the oxidation of the mitochondrial Grx1-roGFP2 sensor but independently of ADH5 (Fig. 6k). Thus, formaldehyde induces GSH:GSSG imbalance even in mitochondria, though in this organelle this redox imbalance is not prevented by ADH5.

• Fig. 3g, the column referring to cells treated with the combination FA/NAC/BSO show values which are clustered into two completely separated (very distant) groups, and statistical significance is not convincing. The authors should explain how s.e.m is so reduced and p is < 0.0001.

Our reply: The SEM and the p values for this experiment were calculated as explained in methods using the software Graphpad. In this platform, the option one-way ANOVA was selected, then no matching or pairing, and a multiple comparison test with a 0.05 confidence level (95% confidence interval) reporting multiplicity adjusted P values, and corrected with a Tukey test. We are providing the raw data for the area of the formed spheroids shown Fig. 3g (now Fig. 4c) in case the reviewer would like to conduct further statistical tests.

"WT	"WT FA 150"	"WT NAC"	"WT NAC LBSO"	"A5 UT"	"A5 FA 150"	"A5 NAC"	"A5 NAC LBSO"
6889	2669	11799	8614	2532	0	3211	3075
8633	2546	10058	8200	3286	0	3110	4143
9397	2369	9341	8380	2883	0	4106	3939
9510	2509	9341	7280	2807	0	4917	3294
9008	2411	10339	1120	2504	0	3803	3105

6867	802	8904	2167	1916	0	4545	3135
6805	1535	9419	1908	1900	0	2800	3197
5093	1623	8851	1830	1721	0	3415	3081
6232	1398	8918	2264	1886	0	3762	3563
6827	914	8463	1601	1490	0	4354	3151

• Fig. 4a and P.6, II.155-160. How do the authors say that ADH5 and GCLM do not further affect viability if they never show a dose response effect for BSO? With the experiments shown, the authors cannot say that ADH5 is the dominant factor in protecting HCT116 cells against FA. Moreover, how do the authors reconcile the different effects of deltaGCL and BSO treatment?

Our reply: We have described along the work the contribution of ADH5 and GCLM to the tolerance of formaldehyde; **not** of L-BSO. We understand that *ADH5* is epistatic to *GCLM* in the viability assay shown in Fig. 5c, and we have modified the text accordingly including the following statement: “This observation indicates that ADH5 is epistatic to GCLM in the FA viability assay (line 184)”. On the other hand, in the formation of spheres we cannot describe which factor is dominant. The different effects observed between $\Delta GCLM$ cells and WT cells treated with L-BSO might be a consequence of different cell lines (a leukemic vs a colorectal cancer), and of the fact that L-BSO further depletes GSH compared to $\Delta GCLM$ (**Extended Data Fig. 6k**).

• P.6, II.165,166. It is chemically impossible that FA can limit GSH antioxidant properties in a ratio of 1:1000.

Our reply: The reaction between formaldehyde and GSH produces HSMGSH, which consists of a formaldehyde molecule covalently bound to the redox-active thiol group of GSH. The continuous generation of formaldehyde from cellular metabolism might only mildly alter the net level of reduced GSH, but it does significantly alter the ratio GSH:GSSG (**Fig. 6i**), which was shown to affect the redox balance (Romero-Aristizabal, et al Nat Comm 2014).

• The formation of HSM-GSH has been known for long time. No need to be detected.

Our reply: We respectfully disagree with this comment. HSMGSH has been known for a long time but it has remained elusive for detection in human cells in part because of the lack of a chemical standard (not commercially available). We have synthesized HSMGSH, characterized the molecule, and used it to identify the endogenous HSMGSH by means of a UPLC-HRMS-based analytical method. Moreover, the detection of HSMGSH indicates that cellular metabolism continuously produces sufficient formaldehyde, which is a robust evidence for the endogenous generation of this aldehyde.

• P.8, I.178. DeltaADH5 cells contain 17.9% less reduced GSH than WT counterparts, which means about 8.5 mM instead of more than 10 mM. Do not the authors think these levels are still effective in keeping the right intracellular redox homeostasis? How do they explain those phenotypes observed in Fig. 1-3?

Our reply: We have addressed this comment above and included these sentences in the text to clarify this point:

1- lines 130-131 “the abundance of GSH (1-10 mmol L⁻¹) might favor the spontaneous reaction between GSH and FA, which, if not limited, could alter the GSH:GSSG balance leading to oxidative stress”

2- lines 223-224 “The GSH:GSSG ratio was further affected by FA in $\Delta ADH5$ but not in WT cells (Fig. 6i)”

It is likely that formaldehyde does not fully deplete GSH, but what it does is to alter the GSH:GSSG ratio and thus E_{GSH} redox potential, which is very sensitive to small changes in GSH cellular level (Romero-Aristizabal, et al Nat Comm 2014).

• P.9, I. 212. 10 uM NAC in worms could not be enough as previously argued in cells

Our reply: We apologize for the typo. In fact, we used a 10 mM NAC concentration in all *C. elegans* experiments.

MINOR POINTS:

• Figures 1C ad D are never mentioned in the text

Our reply: We appreciate the reviewer’s remark. We have modified the text accordingly.

• P.4, I.85. What does “..prevented the early apoptotic markers Annexin V” mean? Annexin V is not an apoptotic marker. The early apoptotic marker is phosphatidylserine (PS) exofacial translocation. Since Annexin V binds to PS, a FITC-conjugated form of this protein has been developed to measure the extent of apoptosis. Moreover, “prevention of a marker” is not appropriate to describe what the authors are looking at.

Our reply: We have modified the text including the following sentence: “in presence of FA, $\Delta ADH5$ cells showed the early apoptosis marker phosphatidylserine (measured by Annexin V)”.

• Extended Figs. 1e,f do not refer to what is described in the text.

Our reply: We have modified the text accordingly.

• Fig. 2f is not convincing. Images should be magnified to state that there is no chromosomal damage. Moreover, the authors should explain discrepancies with Posado et al., and the same Pontel et al. (Nat. Struct.Mol. Biol. and Mol. Cell, respectively).

Our reply: We have modified the text accordingly and magnified WT and $\Delta ADH5$ representative images in presence of formaldehyde. The rest of the images have been moved to the supplementary information. In Rosado et al NSMB 2011 and Pontel et al Mol Cell 2015, formaldehyde was shown to inflict DNA damage in cells and organisms lacking the Fanconi Anemia DNA repair pathway. In our revised manuscript, indeed, we corroborated data by Rosado et al showing that formaldehyde induces the DNA damage marker γ -H2Ax in cells lacking the Fanconi Anemia DNA repair pathway (**Fig. 2e**). We do not see any discrepancy with those papers, because our manuscript addresses how formaldehyde is causing cytotoxicity in DNA repair proficient cells and organisms.

• P.5, I.119-121. DCF fluorescence is a measure of hydrogen peroxide and, to some extent, of other ROS. It is not a measure of cellular oxidative status as the authors stated.

Our reply: We have modified the text accordingly.

• P.7 I.152. BSO was never reported to have side-effects. BSO structure resembles gamma-glutamylcysteine. It binds to GCL as an irreversible inhibitor.

Our reply: Although BSO is a very specific inhibitor of *GCL* with no reported side-effects, we understand in biology the fact that something has not been reported does not mean it does not occur. Therefore, we generated cell lacking *GCLM* as an alternative to the pharmacological drug and presented both approaches to enrich our manuscript.

• Fig. 5a is not mentioned in the text.

Our reply: We have modified the text accordingly (now Fig. 6a, line 193).

• The authors should also show Western blot against p53 in Extended Fig. 1e.

Our reply: The western blot against p53 was shown in Extended Fig. 1e (now Extended Data Fig. 2c).

Reviewer #3 (Remarks to the Author):

Manuscript Title: "Endogenous formaldehyde scavenges cellular glutathione resulting in cytotoxic redox disruption"

General comments: In the study by Umansky et al. entitled "Endogenous formaldehyde scavenges cellular glutathione resulting in cytotoxic redox disruption", the authors showed that formaldehyde (FA) causes cellular damage beyond genotoxicity by triggering oxidative stress, which is prevented by the enzyme alcohol dehydrogenase 5 (ADH5/GSNOR). Mechanistically, they demonstrated that endogenous FA reacts with the redox-active thiol group of glutathione (GSH) forming S-hydroxymethyl-GSH, which is metabolized by ADH5 yielding reduced GSH thus preventing redox disruption. They also identified the ADH5-ortholog gene in *Caenorhabditis elegans* and showed that oxidative stress also underlies FA toxicity in nematodes. In addition, they found that endogenous GSH protects cells lacking the Fanconi Anemia DNA repair pathway from FA. Therefore, they concluded that their findings establish a highly conserved mechanism through which endogenous FA disrupts the GSH-regulated cellular redox homeostasis that is critical during development and aging. Overall, this is an interesting paper which describes endogenous formaldehyde scavenges cellular glutathione resulting in cytotoxic redox disruption. However, there are several weakness, if fixed, will enhance the significance of the study.

Our reply: We appreciate the reviewer's enthusiasm and the recognition of the importance of our work.

Specific

comments:

What is the rationale for measuring blood FA levels in mice? There is not in vivo experiments throughout the paper. Somehow, that part is disconnected from the whole story.

Our reply: This point was also raised by reviewer #1, and we have excluded formaldehyde levels in mice, referring instead to a recent paper in which human blood concentration is determined to be around to 45 μ M for healthy donors (Wei, Y, et al *J. Chromatogr. B Anal. Technol. Biomed. Life Sci*, 2019).

Is the observation specific to cancer cells but not normal cells? Normal cell line controls are preferred. In addition, most of the experiments except Fig 4, which also used Nalm6 cells (an ALL cell line), were done in HCT116 cancer cell lines. To make solid conclusion, other cancer cell lines are needed.

Our reply: We have measured formaldehyde cytotoxicity in presence of the GSH synthesis inhibitor L-BSO in HepG2 (human hepatocellular carcinoma) and HAP1 (human chronic myeloid leukemia) (Extended Data Fig. 4c,d). We have also shown the sensitivity of formaldehyde in presence of L-BSO in chicken DT40 cells and in human fibroblasts isolated from Fanconi Anemia patients (PD20 cells) (Fig. 7a-c). In addition, we have included further experiments in *C. elegans* that show the induction of oxidative stress by formaldehyde (Fig. 3e,f, and Extended Data Fig. 4a), overall strongly suggesting the mechanism of formaldehyde toxicity is not specific to cancer cells and instead is evolutionary conserved from nematodes to mammals.

The authors attempted to make the notion that their findings are "highly conserved". However, they were only able to show the similar results in *c. elegans*. More evidence is needed to make the conclusion.

Our reply: We are presenting data in additional human cell lines (Extended Data Fig. 4c,d), in chicken DT40 cells (Fig. 7a) and in *C. elegans*. We show that ADH5 is phylogenetically conserved through eukarya (Fig. 1g), suggesting the mechanism of formaldehyde toxicity and the role of ADH5 is conserved. Given the evolutionary distance between *C. elegans* and mammals (estimated about 900 million years), it is generally considered evolutionary conserved if a biological mechanism exists in both and is executed by the same genetic pathways. The text has been modified accordingly.

The authors also emphasized that endogenous GSH can protect cells lacking the Fanconi anemia DNA repair pathway from FA, which might have broad implications for Fanconi anemia patients and for healthy BRCA2-mutation carriers. However, they only proved their notion in Δ FANCB cells, which only represents a very small portion of Fanconi patients. Other Fanconi anemia deficient cell lines are needed.

Our reply: We have now measured the sensitivity to formaldehyde in presence of the inhibitor of GSH synthesis L-BSO in DT40 cells deficient in *FANCI* and deficient in *BRCA2* (Fig. 7a). Δ *FANCI* and Δ *BRCA2* deficient cells are sensitive to formaldehyde, and this phenotype is exacerbated by inhibiting GSH synthesis. We are also providing evidences that fibroblasts derived from Fanconi Anemia patients belonging to the complementation group D2 (PD20, *FANCD2* $-/-$) can be sensitized to formaldehyde by L-BSO, in comparison to the isogenic PD20 control cells complemented with a wild type copy of *FANCD2* (Fig. 7b,c). Overall, we covered the core complex of Fanconi Anemia DNA repair (FANCB), the sensing/activating part (*FANCI* and *FANCD2*) and the homologous recombination step (*BRCA2*), which is downstream in the Fanconi Anemia DNA repair pathway.

Minor:

2-segments Y-axis in Fig 2d should be re-arranged.

Quantification for Fig 3b is needed.

Our reply:

- Fig. 2d Y-axis: We have changed the p-p53 quantification plot Y-axis to one segment. The quantification of H2Ax is shown with an Y-axis of 2-segments because having only 1 segment does not allow the reader to visualize the lack of induction of H2Ax signal upon formaldehyde.
- Fig. 3b is a representative plot of the data plotted and quantified in Fig. 3a.

Reviewers' Comments:

Reviewer #1:

Remarks to the Author:

The revised manuscript did not convince me more that the central argument that it is oxidative stress that underlies FA toxicity is correct. One piece of added data is that of the effect of FA treatment on GSH level and the GSH:GSSH ratio. It is shown that FA does not decrease but increases GSH levels and does not affect the GSH:GSSH ratio. I don't see how this fits particularly well in the narrative of the manuscript.

Reviewer #2:

Remarks to the Author:

Unfortunately, the authors did not address satisfactorily my points and neither convince me that the phenotype observed - which is evident, I agree on this - depends on the mechanism they propose. They did not clarify how the thiol antioxidant system (GSH and GSH-related enzymes), which is super efficient, catalitically driven, and 1000 times more concentrated than FA is not able to protect from the very mild oxidative insult produced by transient GSH:GSSG imbalance - which neither result in ROS production, as the authors state. They say no severe DNA damage, no ROS, nothing unless a change of GSH:GSSG ratio. Unfortunately, they contradict themselves when they show that HyPer and Gst4 are induced in *C. elegans* in response to ROS. So...Are ROS produced or not?

Along the same line, the authors continue to defend the thesis that BSO treatment results in ROS production (although they didn't provide references on that). Therefore, they should explain how this condition does not produce side-effects. How this two facts can be reconciled? They authors are not only asked to include further comments in the discussion. They should also prove their hypothesis.

Unfortunately, the authors did not provide a univocal proof that HSMGSH formed upon reaction of GSH with FA results in oxidative stress. I still believe (as I also previously suggested) that other ROS-independent mechanisms should be implicated and investigated.

Reviewer #3:

Remarks to the Author:

the authors have addressed all my concerns

Dear Reviewers,

We are pleased that Reviewer #3 is now fully satisfied with our revision. Unfortunately, Reviewers #1 and #2 felt that our updated manuscript and the new data did not provide enough support for our proposed mechanism of formaldehyde (FA) cytotoxicity. We have addressed most of the points raised during the first round of revision, although we might have not been clear enough to explain them. In this new communication, we would like to further explain our new data, providing a restructured manuscript that supports the following claims:

- (1) that FA can cause cytotoxicity even in presence of functional DNA repair.
- (2) that FA can induce the accumulation of reactive oxygen species (ROS).
- (3) that FA can alter GSH:GSSG balance.
- (4) that oxidative stress can drive FA cytotoxicity in presence of functional DNA repair.
- (5) that GSH synthesis limits FA cytotoxicity, which is particularly important for DNA-repair deficient cells.

Restructuring of the manuscript and layout

We have restructured our manuscript clarifying the narrative and starting with the key observation that Nalm6 pre-B leukemic cells lacking ADH5 (FA metabolism) or lacking the Fanconi Anemia DNA repair pathway ($\Delta FANCB$) are almost equally sensitive to FA (**Fig. 1a, see below**). DNA damage is only detected when the DNA repair pathway is inactive (**Fig. 1b,c, see below**). Next, we show that ADH5 also protects HCT116 cells against FA, and that despite cells being very sensitive to FA, the DNA damage marker H2AX is not detected. Afterwards, we show that the FA cytotoxicity cannot be reverted by the inactivation of p53 (**Fig. 1g**), raising the question, which other effect(s) than DNA damage might be driving FA cytotoxicity in the presence of functional DNA repair?. We introduce here the identification of the gene coding for ADH5 in *C. elegans* and show that it also protects nematodes against FA.

We have maintained the overall layout of the Fig. 2, 3 and 4, which show:

- (1) that FA can induce reactive oxygen species (ROS) in cells and in our animal model *C. elegans* (Fig.2).
- (2) that the antioxidants N-acetylcysteine (NAC) and GSH-MEE can rescue the toxicity caused by FA (Fig.3).
- (3) that inactivating GSH synthesis (or incubating with the inhibitor L-BSO) increases in the sensitivity of cells to FA (Fig. 4).

Figure 5 has been slightly simplified, moving former panels b and c to the supplementary, but showing the alteration of GSH:GSSG ratio by FA by two different techniques: high-resolution mass spectrometry and E_{GSH} genetically encoded sensors. The final figure (now Fig. 6) has not changed and shows that GSH synthesis can limit the toxicity of FA in cells lacking the tumor suppressor BRCA2 or the Fanconi Anemia DNA repair pathway.

In addition, we have modified the abstract and the introduction (in blue in the updated manuscript), highlighting data from two very recent papers showing that mutations in genes coding for FA metabolizing enzymes (ADH5 and ALDH2) -but not in DNA repair- can lead to a severe human condition characterized by bone marrow failure, mental retardation and dwarfism(1, 2). This syndrome was termed AMeD or IBMFS (for aplastic anemia, mental retardation and dwarfism, or inherited bone marrow failure syndrome). Our data

showing that the very common antioxidant NAC (and dietary supplement) can suppress FA toxicity might be a therapeutical intervention for those patients.

We have also introduced major changes into the discussion, describing the importance of our findings for AMeD/IBMFS patients, but also alternative explanations for our results, including limitations of our study.

Please find below a point-by-point to your specific points.

Reviewers' comments:

Reviewer #1 (Remarks to the Author):

The revised manuscript did not convince me more that the central argument that it is oxidative stress that underlies FA toxicity is correct.

We are sorry to hear our updated manuscript did not convince you more. We have done our best efforts to address all the points you raised during the first round of revision, providing new experiments for every suggestion (quantified the ROS markers *Pgst4* and HyPer, determined FA toxicity in presence of L-BSO in more cell lines (HAP1, HepG2, DT40, PD20), measured GSH and GSSG in presence of FA by UPLC-HMRS, determined ROS at non-toxic FA concentrations, shown that L-BSO is cytotoxic for Nalm6 and discussed why L-BSO is not cytotoxic for HCT116).

One piece of added data is that of the effect of FA treatment on GSH level and the GSH:GSSG ratio. It is shown that FA does not decrease but increases GSH levels and does not affect the GSH:GSSG ratio. I don't see how this fits particularly well in the narrative of the manuscript.

1- Regarding the GSH:GSSG ratio, we found that indeed it decreases in $\Delta ADH5$ cells when they are exposed to FA (**Fig. 5g, see below**), which perfectly fits with our proposed mechanism. We posit that in absence of ADH5, GSH cannot be salvaged from FA-GSH products, thus causing GSH:GSSG imbalance. The slight increase in GSH net level you refer to, is only seen in wild type (WT) cells and might be a consequence of induction of GSH synthesis through NRF2 (ROS-response factor). Indeed, this observation is consistent with the induction of the *Pgst-4::GFP* reporter in *C. elegans* (**Fig. 2e,f**). In absence of ADH5, this likely induction of GSH synthesis is not able to compensate for GSH imbalance. We have also measured the ratiometric induction of HyPer in *C. elegans*, confirming the elevation of ROS levels (**Extended Data Fig. 3a**).

Overall, we are supporting that FA induces oxidative stress by several different techniques:

- 1- ROS (H_2O_2) determination by DHFCDA probe.
- 2- ROS determination by roGFP genetically encoded sensor.
- 3- ROS (H_2O_2) determination by the HyPer sensor in *C. elegans*.
- 4- Determination of the ROS-responsive *Pgst-4::GFP* reporter in *C. elegans*.

We also show that FA can alter GSH:GSSG balance by two different and independent approaches:

- 1- Measurement of GSH and GSSG by high-resolution mass spectrometry (UPLC-HRMS).
- 2- By assessing the response of cytoplasmic and mitochondrial E_{GSH} genetically encoded sensors (Grx1-roGFP).

2- What evidence we provide to support that oxidative stress underlies FA toxicity? Our data indicate that in presence of functional DNA repair, FA is still cytotoxic when FA metabolism fails ($\Delta ADH5$ cells/animals). Taking into account the strong induction of ROS by FA shown in Fig. 2, we posit that oxidative stress might underlie FA cytotoxicity. To support this claim, we are showing the following independent results:

1- The antioxidants NAC and GSH-MEE can completely revert FA cytotoxicity in human cells and in *C. elegans* (**Fig. 3**). The reversion of FA cytotoxicity in cells and in *C. elegans* by NAC suggests this widely used food additive can be repurposed for therapeutic intervention in AMeD/IBMFS patients.

2- FA cytotoxicity is exacerbated by the ROS generator paraquat (PQ) in *C. elegans* (**Fig. 3h,i**).

3- FA cytotoxicity is also increased in cells depleted for GSH ($\Delta GCLM$) (**Fig. 4**) or in cells treated with the inhibitor of GSH synthesis L-BSO (**Fig. 4**), which induces ROS as seen in **Fig. 2a,b**.

We think the next step will be to reveal the signalling pathways that controls cell death in response to FA. In this regard, we are showing that p53 is not fully involved FA-death in HCT116 cells (**Fig. 1g**), because the inactivation of this factor ($\Delta ADH5 \Delta TP53$) does not rescue the cellular viability of those cells.

3- One of the critical points raised by Reviewer #1 during the first round of the revision was why L-BSO is not cytotoxic to HCT116 cells, considering that it generates comparable quantities of ROS as FA. We have provided new data showing that L-BSO is cytotoxic to Nalm6 cells (**Extended Data Fig. 4a, see below**), which are also very sensitive to FA, and that the combination of L-BSO and FA further exacerbates cytotoxicity in these cells (**Fig. 4a**). The explanation for the lack of toxicity in HCT116 cells can be found in the recent papers from the laboratory of Joan S. Brugge showing that despite of generating oxidative stress, L-BSO or GSH synthesis inactivation are not cytotoxic for a particular subset of cancer cell lines. This lack of toxicity of L-BSO originates because some cancer cells (including HCT116) compensated the lack of GSH synthesis by two alternative mechanisms: the thioredoxin and the DUBs proteins(3, 4). We believe these papers show why L-BSO is not cytotoxic to HCT116, making it a very useful positive control for the experiments we are showing in Fig 2, simply because L-BSO still can induce ROS without affecting cell health(5). We have incorporated a full paragraph in the discussion to explain L-BSO lack of toxicity in HCT116 cells.

Reviewer #2 (Remarks to the Author):

Unfortunately, the authors did not address satisfactorily my points and neither convince me that the phenotype observed - which is evident, I agree on this - depends on the mechanism they propose.

We appreciate Reviewer #2 is convinced about the phenotypes we show, and we will try to explain better the mechanism we proposed and the data that support it.

They did not clarify how the thiol antioxidant system (GSH and GSH-related enzymes), which is super efficient, catalitically driven, and 1000 times more concentrated than FA is not able to protect from the very mild oxidative insult produced by transient GSH:GSSG imbalance

The amount of intracellular FA is still unknown. Very recently, a paper in *Sci Adv* showed that nuclear FA can rise during differentiation of hematopoietic cells causing DNA damage in cells deficient for the Fanconi Anemia DNA repair pathway(6). Interestingly, the damage caused by endogenous FA in those cells is comparable to the one caused by exposure to 0.5 mM FA. Thus, we cannot rule out that locally intracellular concentrations of FA can reach concentrations able to disrupt GSH:GSSG balance. Indeed, our data indicate that GSH:GSSG ratio can be significantly altered by 150 μ M FA. We have incorporated the following sentences in the discussion:

“Although we were using concentrations near those reported for human blood, the amount of FA that effectively reaches the intracellular space is unknown and becomes a limitation of our study. Remarkably, It was recently reported that in differentiating hematopoietic cells there is a sharp increase in nuclear FA(6). In this system, the DNA damage detected in absence of the Fanconi Anemia DNA repair pathway is comparable to that detected by incubating cells with 0.5 mM FA, which is close to the level reported for cellular GSH. Thus, it is plausible that a local rise in endogenous FA might alter intracellular GSH leading to redox imbalance.”

produced by transient GSH:GSSG imbalance - which neither result in ROS production, as the authors state.

We apologize if our manuscript was not clear enough and we hope the restructured version of our manuscript helps to clarify our findings. We think our data support the following:

1- that FA results in ROS production (**Fig. 2a-d, see below**), as shown by different techniques in cells and in *C. elegans*.

2- that FA can alter GSH:GSSG imbalance particular in absence of ADH5 (**Fig. 5g, see below**) - shown by different techniques: the E_{GSH} sensor (Grx1-roGFP) (**Fig. 5i**) and by Mass spectrometry (**Fig. 5g**). The alteration of the GSH:GSSG balance correlates with ROS generation measured by roGFP genetically encoded sensors, by DHFCDA chemical probe, and in *C. elegans* by HyPer and *gst4::GFP* induction. We posit FA-induced GSH:GSSG imbalance is causing cytotoxicity and provide as evidences: **(1)** that we are able to rescue with the antioxidants N-acetylcysteine and GSH-MEE, and **(2)** we can further increase FA cytotoxicity by co-incubating with paraquat or L-BSO (**Fig. 4**). The alteration of GSH:GSSG ratio by FA might not be the unique mechanism that contributes to an increase in ROS, and future research should reveal other mechanisms that lead ROS accumulation by FA, and to FA cytotoxicity independently of DNA damage. The following sentences were incorporated in the discussion to tone down the claim of GSH:GSSG disruption as unique mechanism:

*“We proposed this GSH:GSSG imbalance can lead to ROS accumulation and cytotoxicity. However, we also detected a significant drop in total GSH in Δ ADH5 cells, which might indicate that the active efflux of HSMGSH could contribute to a drop in the net cellular GSH thus also affecting cellular redox homeostasis (**Fig. 5d-f**). Accordingly, in a recent genome-wide CRISPR-screening, it was reported that the inactivation of the gene coding for the ABC-transporter ABCC1, which pumps GSH-conjugates out of the cell, favors cell growth in the presence of FA(7). Alternatively, due to its strong electrophilicity, FA might react with other abundant cellular thiols such as free cysteine(8) and with sulfhydryl groups present in proteins involved in redox control, thus also contributing to cellular redox disruption. Further research should establish the contribution of those mechanisms to cellular oxidative imbalance triggered by FA.”*

They say no severe DNA damage, no ROS, nothing unless a change of GSH:GSSG ratio

We were not able to detect chromosome damage nor the DNA damage marker H2AX but we **did detected ROS accumulation** and a change in GSH:GSSG ratio (**new Fig. 1** for no DNA damage, and **Fig. 2 and 5** for ROS and GSH:GSSG ratio).

Unfortunately, they contradict themselves when they show that HyPer and Gst4 are induced in *C. elegans* in response to ROS. So...Are ROS produced or not?

We cannot see a contradiction here, the data in *C. elegans* clearly support the ROS generation by FA. Indeed, we have shown ROS production in cells by using a roGFP genetically encoded sensor, by DHFCDA chemical probe, and in *C. elegans* by HyPer and *gst4* induction. Interestingly, our *C. elegans* data even indicate that the cellular response to ROS is orchestrated by SKN-1/Nrf2, which is the oxidative stress response transcription factor activating glutathione and other factors upon ROS, which we might subject in future investigations. Research in the nematode has been exemplary in contributing to a better understanding of the relationship between ROS, oxidative stress response and their impact on development and ageing, and brought forth a well-applied extensive tool kit for quantifying ROS formation and dynamics *in vivo* (9).

Along the same line, the authors continue to defend the thesis that BSO treatment results in ROS production (although they didn't provide references on that).

We are indeed showing that BSO treatment results in ROS production (please see histogram below corresponding to **Fig. 2b**, L-BSO signal is depicted with orange dotted line). Moreover, L-BSO is frequently used as an inducer of ferroptosis -a cellular death caused by iron-catalysed oxidative stress(3, 10). And, L-BSO has been widely used as GSH depletor (11, 12) and to induce oxidative damage to DNA and proteins, and ROS induction (3, 5, 13, 14).

Therefore, they should explain how this condition does not produce side-effects. How this two facts can be reconciled? They authors are not only asked to include further comments in the discussion. They should also prove their hypothesis.

We understand the R3 refers to the fact that L-BSO is not cytotoxic for HCT116, but it does cause a comparable induction of ROS as the one observed with FA. Regarding this point, we have provided new data showing that L-BSO is cytotoxic to Nalm6 cells, which are also very sensitive to FA. As explained in the response to R1 (point 3), the explanation for the lack of toxicity in HCT116 cells can be found in the recent papers from the laboratory of Joan S. Brugge showing that despite of generating oxidative stress, L-BSO or GSH synthesis inactivation is compensated in a subset of cell lines by two alternative mechanism: the thioredoxin and the DUBs proteins. We believe these papers show why L-BSO is not cytotoxic to HCT116, making it a very useful positive control, because it still can induce ROS without affecting cell health. We have incorporated a full paragraph in the discussion to address this point.

We consider that our hypothesis is that FA causes cytotoxicity through inducing oxidative damage. We focus our work on FA toxicity instead of L-BSO because several papers have already described the cell death pathways and compensating mechanisms that operate in response to GSH depletion (and to L-BSO). We acknowledge this suggestion, but we consider exploring L-BSO toxicity might lack of novelty.

Unfortunately, the authors did not provide a univocal proof that HSMGSH formed upon reaction of GSH with FA results in oxidative stress. I still believe (as I also previously suggested) that other ROS-independent mechanisms should be implicated and investigated.

We agree that other mechanisms might contribute to FA cytotoxicity, and we have incorporated a full paragraph in the discussion about the limitations of our studies and other possible interpretation of the data. One important point about detecting HSMGSH is that it indicates that cellular metabolism generates sufficient FA to react with GSH resulting in this compound. HSMGSH has indeed remained elusive from previous studies because of the lack of a commercial HSMGSH that can be bought and used as reference standard in mass spectrometry assays. We synthesized *in-house* this compound and detected it even in absence of supplemented FA. We think the detection of HSMGSH (**Fig. 5**) -which is the substrate of ADH5-; the increased FA toxicity in $\Delta ADH5$ cells and organisms (**Fig. 1**); the alteration in GSH:GSSG ratio (**Fig. 5**); the fact that inactivating GSH-synthesis reduces the tolerance of cells to FA (**Fig. 4**); the generation of ROS upon FA exposure (**Fig. 2**); and the rescue of FA-toxicity by antioxidants (**Fig. 3**), all cumulatively support our mechanism of GSH:GSSG imbalance as FA cytotoxic alteration. However, and as R3 describes, these data do not exclude alternative explanations, which will be matter of future research.

Reviewer #3 (Remarks to the Author):

the authors have addressed all my concerns

References

1. F. A. Dingler, M. Wang, A. Mu, C. L. Millington, N. Oberbeck, S. Watcham, L. B. Pontel, A. N. Kamimae-Lanning, F. Langevin, C. Nadler, R. L. Cordell, P. S. Monks, R. Yu, N. K. Wilson, A. Hira, K. Yoshida, M. Mori, Y. Okamoto, Y. Okuno, H. Muramatsu, Y. Shiraishi, M. Kobayashi, T.

- Moriguchi, T. Osumi, M. Kato, S. Miyano, E. Ito, S. Kojima, H. Yabe, M. Yabe, K. Matsuo, S. Ogawa, B. Göttgens, M. R. G. Hodkinson, M. Takata, K. J. Patel, Two Aldehyde Clearance Systems Are Essential to Prevent Lethal Formaldehyde Accumulation in Mice and Humans. *Mol. Cell* (2020), doi:10.1016/j.molcel.2020.10.012.
2. Y. Oka, M. Hamada, Y. Nakazawa, H. Muramatsu, Y. Okuno, K. Higasa, M. Shimada, H. Takeshima, K. Hanada, T. Hirano, T. Kawakita, H. Sakaguchi, T. Ichimura, S. Ozono, K. Yuge, Y. Watanabe, Y. Kotani, M. Yamane, Y. Kasugai, M. Tanaka, T. Suganami, S. Nakada, N. Mitsutake, Y. Hara, K. Kato, S. Mizuno, N. Miyake, Y. Kawai, K. Tokunaga, M. Nagasaki, S. Kito, K. Isoyama, M. Onodera, H. Kaneko, N. Matsumoto, F. Matsuda, K. Matsuo, Y. Takahashi, T. Mashimo, S. Kojima, T. Ogi, Digenic mutations in *ALDH2* and *ADH5* impair formaldehyde clearance and cause a multisystem disorder, AMeD syndrome. *Sci. Adv.* **6**, eabd7197 (2020).
 3. I. S. Harris, J. E. Endress, J. L. Coloff, G. M. Denicola, W. G. Kaelin, J. S. Brugge, I. S. Harris, J. E. Endress, J. L. Coloff, L. M. Selfors, S. K. Mcbrayer, Deubiquitinases Maintain Protein Homeostasis and Survival of Cancer Cells upon Glutathione Depletion Article Deubiquitinases Maintain Protein Homeostasis and Survival of Cancer Cells upon Glutathione Depletion. *Cell Metab.*, 1–16 (2019).
 4. I. S. Harris, A. E. Treloar, S. Inoue, M. Sasaki, C. Gorrini, K. C. Lee, K. Y. Yung, D. Brenner, C. B. Knobbe-Thomsen, M. A. Cox, A. Elia, T. Berger, D. W. Cescon, A. Adeoye, A. Brüstle, S. D. Molyneux, J. M. Mason, W. Y. Li, K. Yamamoto, A. Wakeham, H. K. Berman, R. Khokha, S. J. Done, T. J. Kavanagh, C. W. Lam, T. W. Mak, Glutathione and Thioredoxin Antioxidant Pathways Synergize to Drive Cancer Initiation and Progression. *Cancer Cell.* **27**, 211–222 (2015).
 5. H. M. Lee, D. H. Kim, H. L. Lee, B. Cha, D. H. Kang, Y.-I. Jeong, Synergistic effect of buthionine sulfoximine on the chlorin e6-based photodynamic treatment of cancer cells. *Arch. Pharm. Res.* **42**, 990–999 (2019).
 6. X. Shen, R. Wang, M. J. Kim, Q. Hu, C.-C. Hsu, J. Yao, N. Klages-Mundt, Y. Tian, E. Lynn, T. F. Brewer, Y. Zhang, B. Arun, B. Gan, M. Andreeff, S. Takeda, J. Chen, J. Park, X. Shi, C. J. Chang, S. Y. Jung, J. Qin, L. Li, A Surge of DNA Damage Links Transcriptional Reprogramming and Hematopoietic Deficit in Fanconi Anemia. *Mol. Cell.* **80**, 1013-1024.e6 (2020).
 7. M. Olivieri, T. Cho, A. Álvarez-Quilón, K. Li, M. J. Schellenberg, M. Zimmermann, N. Hustedt, S. E. Rossi, S. Adam, H. Melo, A. M. Heijink, G. Sastre-Moreno, N. Moatti, R. K. Szilard, A. McEwan, A. K. Ling, A. Serrano-Benitez, T. Ubhi, S. Feng, J. Pawling, I. Delgado-Sainz, M. W. Ferguson, J. W. Dennis, G. W. Brown, F. Cortés-Ledesma, R. S. Williams, A. Martin, D. Xu, D. Durocher, A Genetic Map of the Response to DNA Damage in Human Cells. *Cell.* **182**, 481-496.e21 (2020).
 8. M. Pietzke, G. Burgos-Barragan, N. Wit, J. Tait-Mulder, D. Sumpton, G. M. Mackay, K. J. Patel, A. Vazquez, Amino acid dependent formaldehyde metabolism in mammals. *Commun. Chem.* **3**, 78 (2020).
 9. A. Miranda-Vizueté, E. A. Veal, *Caenorhabditis elegans* as a model for understanding ROS function in physiology and disease. *Redox Biol.* **11** (2017), pp. 708–714.
 10. Y. Sun, Y. Zheng, C. Wang, Y. Liu, Glutathione depletion induces ferroptosis, autophagy, and premature cell senescence in retinal pigment epithelial cells article. *Cell Death Dis.* **9** (2018), doi:10.1038/s41419-018-0794-4.
 11. O. W. Griffith, “Mechanism of Action, Metabolism, and Toxicity of Buthionine Sulfoximine and Its Higher Homologs, Potent Inhibitors of Glutathione Synthesis*” (1982).
 12. O. W. Griffith, A. Meister, Potent and specific inhibition of glutathione synthesis by buthionine sulfoximine (S-n-butyl homocysteine sulfoximine). *J. Biol. Chem.* **254**, 7558–7560 (1979).
 13. G. Gokce, G. Ozsarlak-Sozer, G. Oktay, G. Kirkali, P. Jaruga, M. Dizdaroglu, Z. Kerry, Glutathione depletion by buthionine sulfoximine induces oxidative damage to DNA in organs of rabbits in vivo. *Biochemistry.* **48**, 4980–4987 (2009).
 14. R. Reliene, R. H. Schiestl, Glutathione depletion by buthionine sulfoximine induces DNA deletions in mice. *Carcinogenesis.* **27**, 240–244 (2006).

Reviewers' Comments:

Reviewer #4:

Remarks to the Author:

The manuscript I was sent for review is a revised version, so I will not reiterate the major findings. First off, I would like to stress that some of the data the authors report are novel and exciting: among these, I would like to mention (i) the identification of hydroxymethyl-GSH in formaldehyde-exposed cells and (ii) the identification of an ADH5 ortholog in *C. elegans*, providing convincing evidence for the ortholog being involved in formaldehyde metabolism.

However, I have the following two major issues that prevent me from wholeheartedly supporting the manuscript:

(A) I do concur with a major issue both reviewer 1 and reviewer 2 still see: the role of ROS in formaldehyde toxicity is not yet convincingly demonstrated, and ROS-independent. The authors (in their rebuttal) claim to have provided evidence in favor of ROS being generated: (1) "ROS (H₂O₂) determination by DHFCDA probe" – (2) "ROS determination by roGFP genetically encoded sensor", (3) "ROS (H₂O₂) determination by the HyPer sensor in *C. elegans*.", (4) "Determination of the ROS-responsive Pgst-4::GFP reporter in *C. elegans*." Unfortunately, this evidence is still not convincing – owing in large part to the reactivity of formaldehyde. (ad 1) It needs to be stressed that H₂DCFDA is not the probe per se (as also claimed in the text), but rather its hydrolysis product, H₂DCF, which, upon oxidation, results in fluorescent DCF. This is not correctly indicated in text and figures. Moreover, DCF generation is not at all proof for the generation of hydrogen peroxide; in fact, DCF generation is unspecific to an extent that the only claim that should be made is that some oxidizing species is generated, resulting in H₂DCF oxidation. (ad 2, 3) It is, therefore, highly appreciated that the authors employ additional ways to assess oxidant generation. Normally, roGFP would be an excellent choice – but considering the thiol reactivity of formaldehyde, the question should be how a direct interaction of FA with redox-sensitive thiols can be excluded; in other words: can roGFP fluorescence changes be brought about in an ROS-independent manner through thiol-reactive compounds? (ad 4) Similarly, I contend that it is misleading by the authors to state that the Pgst4::GFP reporter is ROS-responsive – which is, of course, true, but as the reporter will also be affected by other (xenobiotic) compounds (in this case: maybe FA or some FA reaction product other than ROS?), the evidence for ROS generation is no more than circumstantial if based on activation of this reporter.

(B) A major issue that I did not see pointed out by the previous reviewers is the following: In Figure 3, the authors intend to provide evidence for the important role of GSH in FA toxicity. In their description of the experimental setup (lines 439-442 and lines 542-544) the authors state that NAC and formaldehyde (for *C. elegans*) and NAC or GSHmethyl ester and formaldehyde (for cultured cells) were present in parallel during incubation; in fact, thiols were partly present in molar excess over FA. Considering the reactivity of FA with thiols it is therefore expected and not at all surprising that NAC or GSH ester prevent any FA effect.

Reviewer #5:

Remarks to the Author:

Review comments

The study demonstrated that formaldehyde (FA) reacts with the redox-active thiol group in glutathione (GSH), leading to an imbalance of the ratio between GSH and GSSG, which in turn induces the accumulation of ROS and cytotoxicity. The cytotoxicity of FA is inhibited by ADH5 and enhancing GSH synthesis.

Some issues listed below should be addressed to strengthen the paper.

Major comments

1. The article lacks novelty. Many studies have reported that formaldehyde reduces intracellular glutathione and causes oxidative stress, such as Formaldehyde-releasing prodrugs specifically affect cancer cells by depletion of intracellular glutathione and augmentation of reactive oxygen

species (PMID 18030472).

2. Because the study emphasizes the toxic effect of endogenous formaldehyde, the experimental design should start with the increase of endogenous formaldehyde in some disease or pathological state, then correct the downstream disorders through intervention to reduce endogenous formaldehyde, and finally alleviate the disease.

3. The article highlights Fanconi Anemia and human inherited bone marrow failure syndrome (IBMFS) several times, but the study did not include these disease models.

4. In the introduction section, line 46-49, the link of ADH5 and FA is not clearly described. So, the conclusion that "endogenous FA can drive cancer initiation and Fanconi Anaemia phenotypes through causing genome instability" is inappropriate.

5. In the result section, line 103-105, according to the data "despite neither WT nor Δ ADH5 cells presented a significant phosphorylation of the upstream kinases CHK1 and CHK2 in response to FA, we detected a mild phosphorylation of p53 (Ser 15) in Δ ADH5 but not in WT HCT116 cells", the scientific significance should be described.

6. In the result 3 section, why choose HepG2 and HAP1 cell lines. This study detected that the L-BSO were not cytotoxic to the human cancer cells Nalm6 and HCT116. Also, but did not demonstrate whether L-BSO similarly were not cytotoxic to HepG2 and HAP1 cell lines. Also, what is the purpose of selecting multiple cell lines.

7. In the result 6 section, why not directly increase the content of GSH or up-regulate GSH generating enzyme to enhance the tolerance of FANCD2-deficient cells for FA.

8. In the figure 2 a, why didn't the experiment design the concentration of 60 in the Δ ADH5/pA5 group. At the same time, concentration units are not indicated in the figure.

9. In line 90, " γ -H2AX and p-P53 were only detected in Δ FANCB cells", Fig. 1b and extended Fig. 1a cannot explain that p-P53 were only detected in Δ FANCB cells.

10. In Fig. 3f, in the absence of formaldehyde (0mM), why does adh-5 mutation decrease the survival rate of *C. elegans*?

11. In the result "GSH biosynthesis limits FA toxicity"(line 174), in addition to loss of function, gain of function experiments should also be use to demonstrate the conclusion.

12. In Fig. 5d, FA treatment increased GSH levels compared to WT. Why?

13. In result "HSMGSH metabolism prevents GSH:GSSG imbalance"(line 227), the study did not detect the changes of HSMGSH in WT and Δ ADH-5 cells after FA treatment.

14. Under the same experimental conditions, why are the results in Fig. 4b and Fig. 4d so different.

Minor comments

1. There are too many keywords to be precise enough.

2. In Fig. 2e, adh-5 should be changed to adh-5 (RNAi).

3. Line 43, what is the full name for BRCA2? There were many similar errors in the manuscript.

4. The description of "p53" should be uniform. Line 87 is p53, however, line 90 P53.

5. Line 142-144, "Exposure to FA induced a population of cells in which the sensor is oxidized in the absence of ADH5. These results indicate that FA detoxification is necessary to prevent accumulation of ROS (Fig. 2c,d). " should be changed as "Exposure to FA induced a population of cells in which the sensor is oxidized in the absence of ADH5 (Fig. 2c,d). These results indicate that

FA detoxification is necessary to prevent accumulation of ROS. " There were many similar errors in the manuscript.

6. Line 158-159, the sentence "In the same study, NAC was shown to effectively rescue the cell death phenotype caused by L-BSO and MI2 or PR619" lack literature or data support.

7. The concentration of primary antibody should be supplemented in the methods section of manuscript.

8. For figure 4 b, "+100 μ M L-BSO" don't match the picture.

Dear Reviewers,

First of all, I would like to thank you for your comments and constructive suggestions about our manuscript. We have taken on board your suggestions and carried out the experiments requested to address the concerns raised in the last round of revision. Briefly, we have performed measurements of oxidative stress using additional chemical probes and the state-of-the-art Electron Spin Resonance (ESR) spectroscopy. Overall, showing that FA induces the accumulation of reactive oxygen species (ROS) and that ADH5 prevents formaldehyde (FA)-induced oxidative stress. We have also carried out control experiments using a thiol-reactive compound other than FA (iodoacetamide, IAA), showing it does not induce the genetic sensor roGFP. To further evaluate the causative relationship between FA and oxidative stress, we have performed time-shifted viability experiments, which, together with the data showing induction of ROS, cumulatively support oxidative stress contributes to FA cytotoxicity. However, we acknowledge that due to the strong reactivity of FA any result obtained with antioxidants might also be interpreted as quenching of FA instead of ROS. To reflect more thoroughly our findings, we have modified the title that now reads as “*Endogenous formaldehyde scavenges glutathione causing redox disruption and cytotoxicity*” and updated the text discussing the strong reactivity of FA and how this might impact on the interpretation of the data presented in this manuscript.

The new version includes the following additional data:

1- Regarding the induction of oxidative stress and oxidant species:

- a. Measured oxidant species by additional probes, having now 5 different approaches, indicating oxidative stress is generated in response to FA (H2DFCDA, DHE, PBSF, roGFP and HyPer).
- b. Determined that FA induces ROS accumulation in $\Delta ADH5$ cells by ESR.
- c. Measured roGFP in response to the thiol-reactive agent IAA, which shows that a non-specific thiol reactive compound is not able to activate roGFP signal.
- d. Measured the response of *Pgst-4* (SKN-1-dependent) in adults *C. elegans* upon FA treatment in presence of N-acetylcysteine (NAC), indicating that this thiol-rich compound is able to restrict *Pgst-4::GFP* induction in response to FA.

2- Regarding the causative relationship between oxidative stress and cytotoxicity

- a. Determined the toxicity of FA in time-shifted experiments pre-treating with the antioxidant NAC in HCT116 cells, showing a partial reversion of the phenotype.
- b. Showed that a thiol-reactive compound (IAA) other than FA is equally toxic to wild-type and to *ADH5*-deficient cells, supporting the fact that not any thiol-reactive molecule will cause cytotoxicity in absence of *ADH5*.
- c. In adult *C. elegans*, we were not able to detect a significant reversion of the toxicity phenotype when pre-treating animals with NAC for 2 h. These data suggest that *C. elegans* might not accumulate and maintain sufficient NAC after to be able to counteract FA, which is supported by a failure to prevent *Pgst-4::GFP* activation when animals are pre-loaded for 2-h with NAC (see figure in the reply to this specific query by Reviewer 4).

We have also included additional experiments complementing $\Delta GCLM$ strain as requested by Reviewer 5 and updated our manuscript accordingly. Please find below more specific replies to each point raised in the last round of revision.

REVIEWER COMMENTS

Reviewer #4 (Remarks to the Author):

The manuscript I was sent for review is a revised version, so I will not reiterate the major findings. First off, I would like to stress that some of the data the authors report are novel and exciting: among these, I would like to mention (i) the identification of hydroxymethyl-GSH in formaldehyde-exposed cells and (ii) the identification of an *ADH5* ortholog in *C. elegans*, providing convincing evidence for the ortholog being involved in formaldehyde metabolism.

Our reply: we acknowledge these kind words about our data, we are convinced that this work will be the launching platform for several projects in the future linking GSH metabolism, FA metabolism and redox biology.

However, I have the following two major issues that prevent me from wholeheartedly supporting the manuscript:

(A) I do concur with a major issue both reviewer 1 and reviewer 2 still see: the role of ROS in formaldehyde toxicity is not yet convincingly demonstrated, and ROS-independent. The authors (in their rebuttal) claim to have provided evidence in favor of ROS being generated: (1) "ROS (H₂O₂) determination by DHFCDA probe" – (2) "ROS determination by roGFP genetically encoded sensor", (3) "ROS (H₂O₂) determination by the HyPer sensor in *C. elegans*.", (4) "Determination of the ROS-responsive Pgst-4::GFP reporter in *C. elegans*." Unfortunately, this evidence is still not convincing – owing in large part to the reactivity of formaldehyde. (ad 1)

Our reply: We appreciate this concern, and we have performed additional experiments to further demonstrate oxidative stress induction by FA, and to address that the signal detected using the genetic ROS probe (roGFP) is not an artifact of the strong reactivity of FA. Please find a more detailed explanation below and in the reply to the comment "A" found at the end of this letter.

It needs to be stressed that H₂DCFDA is not the probe per se (as also claimed in the text), but rather its hydrolysis product, H₂DCF, which, upon oxidation, results in fluorescent DCF. This is not correctly indicated in text and figures. Moreover, DCF generation is not at all proof for the generation of hydrogen peroxide; in fact, DCF generation is unspecific to an extent that the only claim that should be made is that some oxidizing species is generated, resulting in H₂DCF oxidation. (ad 2, 3)

Our reply: We have modified the text and figures, including a more precise description of H₂DCFDA:

We therefore measured the level of oxidant species by quantifying the oxidation of the probe 2',7'-dichlorodihydrofluorescein diacetate (H₂DCFDA). In the cells, this probe is converted into dichlorodihydrofluorescein (H₂DCF) that in presence of oxidant species is oxidized into fluorescent dichlorofluorescein (DCF).

Please refer to our reply to comment A at the end of this letter.

It is, therefore, highly appreciated that the authors employ additional ways to assess oxidant generation. Normally, roGFP would be an excellent choice – but considering the thiol reactivity of formaldehyde, the question should be how a direct interaction of FA with redox-sensitive thiols can be excluded; in other words: can roGFP fluorescence changes be brought about in an ROS-independent manner through thiol-reactive compounds? (ad 4)

Our reply: To address this comment, we have carried out a kill curve and measured roGFP with the compound Iodoacetamide (IAA), which reacts with thiol-groups, mainly cysteines, but whose product is not a substrate of ADH5 (see figure below). Indeed, IAA is used in proteomics to block reduced cysteine residues for protein characterization and peptide mapping. Conversely, IAA is very toxic for

HCT116 cells, but it does not differentially affect wild type and *ADH5*-deficient cells. Using non-toxic and sublethal concentrations of IAA we were not able to detect the induction of roGFP signal, suggesting that roGFP cannot be activated by any thiol-reactive molecule (**Extended Data Fig. 3a-c**). Moreover, we detected a reduction in the roGFP signal, likely due to the alkylation and inactivation of roGFP by IAA, something opposite to what we detected using FA.

Similarly, I contend that it is misleading by the authors to state that the *Pgst4::GFP* reporter is ROS-responsive – which is, of course, true, but as the reporter will also be affected by other (xenobiotic) compounds (in this case: maybe FA or some FA reaction product other than ROS?), the evidence for ROS generation is no more than circumstantial if based on activation of this reporter.

Our reply: We have modified the text, taken out the sentence that refers to *Pgst4::GFP* as ROS-responsive. Regarding ROS generation, we have incorporated additional chemical probes and ESR spectroscopy data (see reply to comment A at the end of this letter). Considering these new data further supporting the induction of ROS upon FA by alternative techniques, we think that it is likely that SKN-1 is inducing the expression of *Pgst4::GFP* upon detection of ROS. To gain insights into the activation of SKN-1 in response to FA, we determined the induction of *Pgst4::GFP* in adult *C. elegans* in presence of NAC. Interestingly, the co-treatment of NAC and FA showed a significant reversion of *Pgst4::GFP* signal, still the signal was not reverted to the basal level (**Fig. 2h**). This observation indicates that FA triggers a persistent induction of SKN-1 response that can be significantly reduced, although not completely abrogated by a thiol-rich antioxidant compound. This experiment is still not able to distinguish whether *Pgst4::GFP* activation is due to ROS generated upon FA exposure or due to some FA-reaction product, because NAC could be quenching ROS or quenching FA. However, the data strongly support that FA underlies the *Pgst4::GFP* induction.

Interestingly, SKN-1 shares similarities with the mammalian factor NRF2, however worms lack KEAP1, the partner of NRF2 that senses oxidative stress through cysteines. In mammals, when the cysteines of KEAP1 are oxidized, NRF2 is released and translocates into the nucleus to activate the antioxidant response. In *C. elegans*, the oxidative-stress dependent activation of SKN-1 depends more on its phosphorylation mediated by the MAPK p38. Our data showing activation of SKN-1 might indicate that FA response originates in p38. In mammals, p38 has been reported to play a dual role. On one side, it acts as a pro-apoptotic factor but also it has opposite role in protecting cells from oxidative stress through the induction of antioxidant genes (Gutierrez-Uzquiza, et al JBC, 2012). In the upcoming years we expect to address the role of p38 in FA tolerance, resolving also which is the molecule that triggers the activation of this signalling -a ROS or directly FA.

Gutierrez-Uzquiza, et al p38α mediates cell survival in response to oxidative stress via induction of antioxidant genes: effect on the p70S6K pathway. J. Biol. Chem. 287, 2632–2642 (2012).

(B) A major issue that I did not see pointed out by the previous reviewers is the following: In Figure 3, the authors intend to provide evidence for the important role of GSH in FA toxicity. In their description

of the experimental setup (lines 439-442 and lines 542-544) the authors state that NAC and formaldehyde (for *C. elegans*) and NAC or GSHmethyl ester and formaldehyde (for cultured cells) were present in parallel during incubation; in fact, thiols were partly present in molar excess over FA. Considering the reactivity of FA with thiols it is therefore expected and not at all surprising that NAC or GSH ester prevent any FA effect.

Our reply: We thank the Reviewer for raising this concern. Indeed, we show that FA and GSH can react forming S-hydroxymethyl-GSH both in cell-free systems (**Extended Fig. 6a,b**) and in cells (**Fig. 5**). It is therefore possible that NAC or GSH-monoethylester (GSH-MEE) can quench at least part of the FA added to the culture medium, or eventually FA inside the cells. The product formed by GSH-MEE or NAC with FA may still reach the cellular lumen. In fact, Nalm6 culture medium contains 50 μ M b-mercaptoethanol (a thiol able to quench FA), nonetheless Nalm6 cells are sensitive to concentrations as low as 30 μ M FA (**Fig. 1**). We think this interpretation still does not invalidate our proposal of using NAC as a therapeutic agent in patients harbouring mutations in *ADH5* and *ALDH2* (IBMFS/AMeD syndrome), who present high levels of circulating FA. In these patients, independently of the mechanisms of reversion of FA toxicity, an increase in circulating thiols might significantly improve their outcome.

Being able to discern between the ability of a thiol-rich molecule such as NAC of quenching FA or quenching of a ROS formed upon FA exposure is technically challenging and might not be feasible due to the strong reactivity of FA. For example, the partial reversion of FA toxicity we observed pre-loading HCT116 cells with NAC (**Fig. 3a**) could still be because FA is quenched intracellularly, and not because ROS are neutralized. Thus, we think that any experiment combining NAC and FA might still be dually interpreted as quenching of FA or neutralization of ROS. Nonetheless, we have performed the experiments suggested in the additional comment B at the end of this letter. (A) We performed time-shifted experiments using NAC and FA in HCT116 cells (WT and $\Delta ADH5$) (**Fig. 3a**) and (B) time-shifted experiments using adult *C. elegans* (**Extended Data Fig. 4e**). A pre-treatment of HCT116 cells with NAC partially reverted the toxicity phenotype in HCT116 but has no effect in *C. elegans*. It might be possible that *C. elegans* is not able to maintain elevated intracellular NAC, and that after washing off NAC, the intracellular level of NAC also decays. Consistently, the 2-h pre-treatment with NAC cannot significantly limit the activation of *Pgst4::GFP* in *adh-5*(RNAi) worms, indicating that loading of worms with NAC is not sufficient to prevent the FA-dependent *Pgst4::GFP* induction nor the toxicity of FA (**see figure below, and Extended Data Fig. 4e**)

Reviewer #5 (Remarks to the Author):

Review comments

The study demonstrated that formaldehyde (FA) reacts with the redox-active thiol group in glutathione (GSH), leading to an imbalance of the ratio between GSH and GSSG, which in turn induces the accumulation of ROS and cytotoxicity. The cytotoxicity of FA is inhibited by ADH5 and enhancing GSH synthesis.

Some issues listed below should be addressed to strengthen the paper.

Major comments

1. The article lacks novelty. Many studies have reported that formaldehyde reduces intracellular glutathione and causes oxidative stress, such as Formaldehyde-releasing prodrugs specifically affect cancer cells by depletion of intracellular glutathione and augmentation of reactive oxygen species (PMID 18030472).

Our reply: We believe that the novelty in our manuscript consists of several findings such as the characterization of the ADH5 ortholog in *C. elegans*; the role of GSH synthesis in the protection against FA toxicity; the identification of S-hydroxymethyl-GSH in cells unexposed to FA, which indicates that metabolism generates sufficient FA; and the role of ADH5 in sustaining GSH:GSSG balance. In addition, in our revised manuscript we present additional evidence indicating that ADH5 prevents oxidative stress triggered by FA. The reference mentioned (PMID 18030472) points towards the same direction as our manuscript, however the drugs used in that paper are histone deacetylase inhibitors that might be altering cell health independently of the proposed release of FA.

2. Because the study emphasizes the toxic effect of endogenous formaldehyde, the experimental design should start with the increase of endogenous formaldehyde in some disease or pathological state, then correct the downstream disorders through intervention to reduce endogenous formaldehyde, and finally alleviate the disease.

Our reply: We truly appreciate this comment, and indeed in our previous version we have incorporated in the introduction and in the discussion the fact that mutations in FA metabolism can lead to the human inherited bone marrow failure syndrome (IBMFS). It was recently shown that an increase in endogenous FA can drive IBMFS (Dingler et al 2020), thus we have focused our current manuscript on (1st) confirming that FA metabolism controls the cellular tolerance to FA, (2nd) addressing a mechanism of how that toxicity is inflicted in organisms without mutations in DNA repair, and (3rd) proposing an intervention to alleviate FA toxicity. We feel that the manuscript is organized not very differently to what the Reviewer is proposing, and that further organization would require major changes that may not further clarify the findings reported here.

3. The article highlights Fanconi Anemia and human inherited bone marrow failure syndrome (IBMFS) several times, but the study did not include these disease models.

Our reply: We have not included any disease model of Fanconi Anemia, because our initial aim was to address the toxic mechanism of FA in presence of DNA repair. This approach allowed us to uncover that FA induces the accumulation of ROS, and that GSH prevents FA toxicity. Moreover, we are showing for the first time that H24K24.3 is the ortholog of *ADH5* in *C. elegans*. In the recent paper by Dingler et al, it is shown that mice lacking *Adh5* exposed to a FA precursor (methanol) developed similar phenotypes as *Adh5*^{-/-} *Aldh2*^{-/-} mice not exposed to any additional source of FA (a model of IBMFS/AMeD syndrome). Thus, we are using cells lacking ADH5 exposed to FA as a model of overwhelmed FA metabolism. Our next step would be to develop *ADH5 ALDH2* deficient worms and characterize this disease model. This model would be helpful for the understanding of IBMFS/AMeD phenotypes, mainly because breeding mice lacking *Adh5* and *Aldh2* is challenging due to the severe postnatal lethality (Dingler et al 2020).

4. In the introduction section, line 46-49, the link of ADH5 and FA is not clearly described. So, the conclusion that “endogenous FA can drive cancer initiation and Fanconi Anaemia phenotypes through causing genome instability” is inappropriate.

Our reply: We have included a more extensive explanation of ADH5, Fanconi Anemia and FA that reads as follow:

This disease is caused by mutations in genes coding for factors that conform the Fanconi Anaemia DNA repair pathway. When this DNA repair pathway is not functional, cells are more sensitive to compounds that can damage the DNA such as DNA interstrand-crosslinking (ICL) agents and simple aldehydes¹¹. Mice with inactivating mutations in the gene coding for the enzyme alcohol dehydrogenase 5 (Adh5) and in the gene coding for the factor Fanconi Anaemia Complementation Group D2 (Fancd2) accumulate FA-DNA adducts and present hematopoietic stem cell attrition, which correlates with genome instability in the hematopoietic compartment⁹. Moreover, these mice develop leukaemia and liver cancer and present widespread karyomegaly⁹, overall indicating that endogenous FA can drive cancer initiation and Fanconi Anaemia phenotypes through causing genome instability.

5. In the result section, line 103-105, according to the data “despite neither WT nor Δ ADH5 cells presented a significant phosphorylation of the upstream kinases CHK1 and CHK2 in response to FA, we detected a mild phosphorylation of p53 (Ser 15) in Δ ADH5 but not in WT HCT116 cells”, the scientific significance should be described.

Our reply: we have included a paragraph discussing that despite the phosphorylation detected in CHK2 is not significant (**Extended Data Fig. 2c**), it might lead to a mild phosphorylation of p53 at Ser 15. However, this mild activation of p53 in Δ ADH5 HCT116 cells is not sufficient to trigger p53-dependent cell death. Accordingly, the inactivation of p53 does not rescue the toxicity of FA (**Fig. 1g**). It reads as follow:

*Accordingly, we detected a mild phosphorylation of p53 in Δ ADH5 HCT116 cells exposed to FA (**Fig. 1f**), and a non-significant increase in p-CHK2 (**Extended Data Fig. 2c**), suggesting that in presence of DNA repair, FA-caused DNA damage can only mildly activate a CHK2-dependent DNA damage response. However, this DNA damage is still not sufficient to significantly damage chromosomes (**Fig. 1h**), which would trigger p53-dependent apoptosis¹³. Concordantly, the inactivation of p53 did not rescue the viability of Δ ADH5 cells in presence of FA (**Fig. 1g**).*

6. In the result 3 section, why choose HepG2 and HAP1 cell lines. This study detected that the L-BSO were not cytotoxic to the human cancer cells Nalm6 and HCT116. Also, but did not demonstrate whether L-BSO similarly were not cytotoxic to HepG2 and HAP1 cell lines. Also, what is the purpose of selecting multiple cell lines.

Our reply: We initially focused our study on a blood-derived cell line (Nalm6, acute lymphoblastic leukemia) and HCT116 as a non-blood model (colorectalcarcinoma). In the previous round of revision, we were asked for additional cell lines related to the phenotypes developed by mice lacking *Adh5* and the Fanconi Anemia DNA repair pathway. These animals develop bone marrow failure, liver failure, hepatocarcinoma and leukemia. Thus, we decided to include HepG2 (a liver-cancer derived cell line) and HAP1 (a chronic myeloid leukaemia-derived cell line) to further strengthen our manuscript. We have included the L-BSO toxicity data for HepG2 and HAP1 at 100 μ M L-BSO, which is the concentration used in our experiments (**Extended Data Fig. 5b**).

7. In the result 6 section, why not directly increase the content of GSH or up-regulate GSH

generating enzyme to enhance the tolerance of FANCD2-deficient cells for FA.

Our reply: We thank the Reviewer for this suggestion, and indeed early papers suggested that exposing lymphocytes from Fanconi Anemia patients to the GSH precursor N-acetylcysteine was able to revert genome instability (Ponte et al, 2012). We are planning to carry out these interventional experiments as part of the future projects.

8. In the figure 2 a, why didn't the experiment design the concentration of 60 in the $\Delta ADH5/pA5$ group. At the same time, concentration units are not indicated in the figure.

Our reply: we apologize for the omission. We have reorganized figure 2 including a complete set of data for WT, $\Delta ADH5$ and for $\Delta ADH5/pADH5$, not only in H2DFCDA but also for the additional oxidative stress probe DHE.

9. In line 90, “ γ -H2AX and p-P53 were only detected in $\Delta FANCB$ cells”, Fig. 1b and extended Fig. 1a cannot explain that p-P53 were only detected in $\Delta FANCB$ cells.

Our reply: we acknowledge that this sentence did not clearly explain the results presented in Fig 1. Indeed, FA was unable to induce p-p53 in our setting using $\Delta ADH5$ or $\Delta FANCB$ Nalm6 cells. We have modified the text that now read as follow:

Despite that $\Delta ADH5$ and $\Delta FANCB$ cells were significantly sensitive to physiological levels of FA, γ -H2AX was detected in the presence of FA in $\Delta FANCB$ but not in $\Delta ADH5$ cells (Fig. 1b,c), indicating that FA also impairs cell viability when DNA repair is intact, but without triggering the DNA damage marker γ -H2AX. In contrast, p-p53 and γ -H2AX were significantly induced in $\Delta ADH5$ and $\Delta FANCB$ cells by the ICL-DNA damaging drug cisplatin (Cpt) (Fig. 1b,c).

10. In Fig. 3f, in the absence of formaldehyde (0mM), why does *adh-5* mutation decrease the survival rate of *C. elegans*?

Our reply: The reduced survival rate in this graph is observed in the NAC treated worms only. We sometimes observe a mild effect on animal survival in wt or *adh-5* mutants due to the NAC treatment. Although NAC is known to have an overall pro-survival effect on *C. elegans*, some studies show the induction of a mild stress response (induction of protein-folding stress and others, e.g. doi: [10.6061/clinics/2015\(05\)13](https://doi.org/10.6061/clinics/2015(05)13)) and the suppression of endogenous ROS, which could have mild adverse effects on the worms.

11. In the result “GSH biosynthesis limits FA toxicity”(line 174), in addition to loss of function, gain of function experiments should also be use to demonstrate the conclusion.

Our reply: We have shown that GSH biosynthesis limits FA toxicity by two independent approaches (pharmacological using L-BSO and by genetic inactivation of GSH synthesis ($\Delta GCLM$)). We are now including viability data for the $\Delta GCLM/pGCLM$ and $\Delta GCLM \Delta ADH5/pGCLM$ (Fig. 4b). To maintain the symmetry of the figure, we have moved the panel with 3D sphere formation in presence of L-BSO to **Extended Data Fig. 5g**. In addition, we are also showing the rescue of FA toxicity in adult *adh-5* worms by expressing the transgene *adh-5::GFP* (**Extended Data Fig. 2g**).

12. In Fig. 5d, FA treatment increased GSH levels compared to WT. Why?

Our reply: We think this interesting observation suggests that a transient increase in oxidative stress might be triggering the activation of NRF2, which induces GSH synthesis and consequently an

increase of net GSH. In absence of ADH5, the formation of HSMGSH might outcompete this induction, affecting GSH:GSSG balance and GSH level. Accordingly, the data in *C. elegans* indicate that FA can induce the SKN-1 (the NRF2-ortholog)-dependent gene *gst-4* (Fig. 2g,h).

13. In result “HSMGSH metabolism prevents GSH:GSSG imbalance”(line 227), the study did not detect the changes of HSMGSH in WT and Δ ADH-5 cells after FA treatment.

Our reply: Our data show that in absence of any added FA, HSMGSH accumulates relative to GSH in cells lacking ADH5 (Fig. 5b). HSMGSH is metabolized by ADH5, and we detected a significant imbalance in GSH:GSSG ratio in Δ ADH5 cells even in absence of exogenous FA (Fig. 5g). Thus, cumulative supporting that HSMGSH metabolism is necessary to maintain GSH:GSSG balance.

When exposing Δ ADH5 cells to 150 μ M FA, HSMGSH levels were below the detection limit of the UPLC-HRMS-based analytical method, which might be because extrusion of GSH, as suggested by Tulpule et al (<https://doi.org/10.1111/jnc.12170>). This limitation prevented us from calculating the amount of HSMGSH relative to GSH in Δ ADH5 cells exposed to FA. Please find below the measurement of HSMGSH in WT cells exposed to FA (not included in the manuscript).

14. Under the same experimental conditions, why are the results in Fig. 4b and Fig. 4d so different.

Our reply: We apologize for the labelling of Fig. 4d, which was confusing. In Fig. 4b (now Extended Data Fig. 5g) and d (now Fig. 4c), we are detecting similar results by blocking GSH synthesis using the inhibitor L-BSO (Extended Data Fig. 5g) and by genetically inactivating GSH synthesis (Δ GCLM) (Fig. 4c). In both cases, cells can hardly form spheres when GSH synthesis is inactive, although the phenotypes using the inhibitor L-BSO are more significant. We have reorganized the labelling of the Extended Data Fig. 5g to clarify the treatments shown in each panel.

Minor comments

1. There are too many keywords to be precise enough.

Our reply: We have taken out some of the keywords as suggested by the Reviewer.

2. In Fig. 2e, *adh-5* should be changed to *adh-5* (RNAi).

Our reply: The nomenclature was corrected.

3. Line 43, what is the full name for BRCA2? There were many similar errors in the manuscript.

Our reply: The full name was included. Also the full names for ADH5, ALDH2, FANCD2, SKN-1 and NRF2.

4. The description of “p53” should be uniform. Line 87 is p53, however, line 90 P53.

Our reply: The nomenclature was corrected.

5. Line 142-144, “Exposure to FA induced a population of cells in which the sensor is oxidized in the absence of ADH5. These results indicate that FA detoxification is necessary to prevent accumulation of ROS (Fig. 2c,d).” should be changed as “Exposure to FA induced a population of cells in which the sensor is oxidized in the absence of ADH5 (Fig. 2c,d). These results indicate that FA detoxification is necessary to prevent accumulation of ROS.” There were many similar errors in the manuscript.

Our reply: The sentence was corrected.

6. Line 158-159, the sentence “In the same study, NAC was shown to effectively rescue the cell death phenotype caused by L-BSO and MI2 or PR619” lack literature or data support.

Our reply: The data reported by Harris et al 2019 (DOI: 10.1016/j.cmet.2019.01.020) show that NAC can rescue cell death caused by a combination of L-BSO and MI-2 or EERI. We have corrected the sentence that was originally referring to MI2 and PR619, replacing it for MI-2 and EERI. We have also included the reference to the paper that support that sentence. Please find below the figure from Harris et al 2019 Cell Met, where authors show that NAC but not Trolox rescued cell death caused by a combination of L-BSO and MI-2 or EERI (Fig. S4G in the manuscript by Harris et al).

7. The concentration of primary antibody should be supplemented in the methods section of manuscript.

Our reply: The concentration used are now included in the method section.

8. For figure 4 b, “+100 μM L-BSO” don't match the picture.

Our reply: We have changed the labelling of this panel that is now in Extended Data Fig. 5g

Additional comments from reviewer #4

Regarding my concerns under A, the authors should use assays assessing the formation of reactive oxygen species such as superoxide or hydrogen peroxide that are both specific and not relying on components directly interacting with formaldehyde. Examples would include superoxide-specific probes such as dihydroethidine, which would react with superoxide to generate a hydroxyethidine product that can then be identified by HPLC. Regarding hydrogen peroxide formation, peroxidase-based assays could be employed for the detection of H_2O_2 , but direct interference with formaldehyde would have to be excluded beforehand.

Our reply: We have included additional chemical probes to further support the specific generation of ROS, indicating cellular peroxides accumulate upon FA:

1- DHE (dihydroethidium), which can form two fluorescent products. One is ethidium (ex 480 nm/em 576 nm), which is formed by nonspecific redox reactions, while the other is 2-hydroxyethidium (2-OH-E+), a specific adduct of $O_2^{\bullet-}$ (ex 500-530 nm/em 590-620 nm). We assayed DHE by flow cytometry using the PE channel (Excitation at 488 nm, Emission 575/26), and detected a significant induction of fluorescence, comparable to the one induced by the ROS generator antimycin A (**Fig. 2c**). Although we cannot distinguish whether this signal originated from Ethidium or 2-OH-E+, this assay supports that FA can induce the accumulation of oxidant species, likely ROS. Unfortunately, an HPLC device was not available for detecting the superoxide-DHE product.

2- To gain insight into the identity of the ROS induced by FA in $\Delta ADH5$ cells, we performed Electron Spin resonance spectroscopy using the CMH spin probe, which detects the presence of superoxide ion, peroxy radical, peroxyxynitrite and nitrogen dioxide (**Fig. 2f, and Extended Data Fig. 3d**). We detected a significant induction of ROS in $\Delta ADH5$ exposed to 60 μM and a further increase in combination of L-BSO and FA. Interestingly, in WT cells neither L-BSO nor FA alone were able to induce ROS, but the ESR signal was detected when FA and L-BSO were combined in WT cells. 150 μM FA showed the same trend as 60 μM , though the signal does not further increase compared to 60 μM .

3- We have incorporated the peroxide-specific probe pentafluorobenzenesulfonyl fluorescein (PBFS) (**Extended Data Fig. 3e**). This probe is selective for cellular peroxides (although the original paper claims it is specific for H_2O_2 , it could be peroxidised by other cellular peroxides than H_2O_2). PBFS reports peroxides through a non-oxidative mechanism. Briefly, the perhydrolysis of a sulfonyl linkage releases a fluorescein moiety that can be detected by flow cytometry (see below).

(Adapted from Maeda et al 2004).

The peroxidase-based assays suggested would have also been a good option, however we chose a chemical probe (PBFS) that does not depend on oxidation for detecting cellular peroxides, and that allows us to track peroxide accumulation by flow cytometry. Any peroxidase-based assay would have required cell lysis and determination in the supernatant, which may be less sensitive, and the peroxidase susceptible to be damaged by FA.

Maeda, H., Fukuyasu, Y., Yoshida, S., et al. Fluorescent probes for hydrogen peroxide based on a non-oxidative mechanism. *Angew Chem. Int. Ed. Engl.* 43(18), 2389-2391 (2004).

4- We have shown that a thiol-reactive drug (iodoacetamide, IAA) is not able to induce roGFP, suggesting that the signal detected in presence of FA is indeed related to the ability of FA to induce ROS accumulation and not because FA directly reacts with roGFP. Even more so, IAA seems to inactivate roGFP fluorescence, indicating that a direct reaction with its thiols do not cause an increase in roGFP signal (**Extended Data Fig. 3a-c**).

As for the concerns under B, these are really major in my eyes, as they question the findings on the role of GSH in the observed effects. Here, the authors should either show that there is no direct interaction between formaldehyde and the used GSH or thiol. Or they should perform incubations in a manner that avoids direct interaction, for example by time-shifted incubation (for example, preincubate with thiol to load cells or worms, wash off supernatant, then incubate with formaldehyde).”

Our reply: We consider that we are presenting several independent experiments supporting the role of GSH in preventing FA cytotoxicity, including loss -by genetic inactivation and chemical inhibition; and gain of function experiments of GSH synthesis ((**Fig. 4b**), as requested by Reviewer 5). However, as discussed above, we understand that distinguishing whether GSH is quenching FA or if it is quenching ROS formed upon FA, is technically challenging, and any result might be still dually interpreted. Independently of this limitation, we have performed the experiments suggested (see our reply to the original comment (B) above) and we believe our data cumulatively support that FA induces oxidative stress and that GSH can limit FA toxicity, suggesting a causative relation between ROS and FA cytotoxicity. We are including the following paragraph in the discussion to highlight the limitation of the interpretation of our results:

This strong FA electrophilicity also limits the interpretation of the experiments shown in Fig. 3 using antioxidants to suppress FA cytotoxicity. The rescue of cytotoxicity might be because oxidative stress is reduced, or because FA is being quenched by NAC/GSH-MEE, or both. Our data cumulatively support oxidative stress induction upon FA and suggest that cytotoxicity is, at least in part, caused by the accumulation of oxidant species.

Reviewers' Comments:

Reviewer #5:

Remarks to the Author:

Dear Editor and author

We have carefully considered the author's responses for reviewers' comments of manuscript (Manuscript#: NCOMMS-20-16485C). The author has given reasonable explanations and effective evidence for the reviewer's comments. There are no flaws in the data analysis, interpretation and conclusions. This work is of great significance to this field and related fields. I would suggest to accept it.

Best wishes

Reviewer #6:

Remarks to the Author:

A major issue raised by Reviewer 4 is whether this study provides sufficient evidence of ROS involvement in the reported effects of endogenous formaldehyde. The new EPR findings included by the authors in response do show strong circumstantial evidence of ROS involvement in the FA effects.

Reviewer #4 correctly questions the validity of the roGFP data on the grounds that electrophilic adduction by FA could conceivably lead to a spectral change in the fluorogenic center that may be mistaken for that which occurs when the vicinal cysteines on the surface of the beta barrel of roGFP are oxidized to form a disulfide.

To the Reviewer's point, roGFP is not a direct sensor of ROS. Rather, roGFP reports on the glutathione redox potential (EGSH) established by the ratio GSH/GSSG in the Nernst equation (it has been demonstrated that EroGFP equals EGSH, See Meyer and Dick, 2010

DOI:10.1089/ars.2009.2948) through a redox relay involving glutathione peroxidase and glutaredoxin.

Unfortunately, the iodoacetic acid experiment performed by the authors is not all that informative because the results are not generalizable to other electrophiles including those that can induce dithiol formation. Specifically, FA has been described to form a hemithioacetal with cysteine (<https://doi.org/10.1038/s42004-019-0224-2>), which could condense into a disulfide with a neighboring thiol. Thus, it is possible that FA induces the same disulfide on roGFP that results from oxidation through its redox relay. To validate the use of roGFP with FA as a stimulus, the integrity of this redox relay has to be demonstrated.

On a related note, the authors' description of the finding is also confusing. That "...roGFP cannot be activated by any thiol-reactive molecule" is factually incorrect as stated (e.g., diamide, H₂O₂) but also, I suspect, not what the authors intend to say. They may mean to say that an oxidative (electrophilic) attack on roGFP is not sufficient to cause a spectral change consistent with its oxidation. The Spanish "cualquier" does not accurately translate to "any" in this instance, "not just any" may be their intended meaning here.

The equilibration of GSH/GSSG with roGFP is mediated by its glutathionylation by glutaredoxin. This affords at least two opportunities to determine whether the change in the ratio of fluorescence intensity induced by 405 and 488 excitation is secondary to a FA attack on the sensor.

If ROS are involved, inhibition of glutaredoxin activity (e.g., 2-AAPA) should completely ablate the roGFP signal induced by FA, as it does when H₂O₂ is given to the cells. Secondly, a comparison of the rate of the FA-induced response of roGFP and the chimeric glutaredoxin-roGFP version (which the authors also report using in this study!) would be expected to show a faster (though not necessarily a stronger) response in Grx-roGFP expressing cells.

Confirmation that the FA-induced roGFP response depends on glutaredoxin would establish that FA treatment induces an increase in the intracellular EGSH, which would support the authors' thesis. However, by itself, it would not prove the involvement of ROS.

The logical next step would be to test the role of glutathione peroxidase activity in the roGFP relay. Supplementing the cells with sodium selenite overnight would likely increase the expression of the selenium-containing Gpxs nonspecifically, which would be expected to alter the kinetics of the intracellular roGFP response to FA treatment relative to that in control unsupplemented cells. A

knock down/knockout of Gpx would be convincing as well, but it would hinge on targeting the correct Gpx isoform to distinguish between H₂O₂ and lipid peroxide involvement. Perhaps a simpler approach at this point would be to test whether the FA effects require H₂O₂. Exogenous catalase and/or endogenous catalase overexpression, with appropriate controls, could be an easy thing to try.

Reviewer #5 (Remarks to the Author):

Dear Editor and author

We have carefully considered the author's responses for reviewers' comments of manuscript (Manuscript#: NCOMMS-20-16485C). The author has given reasonable explanations and effective evidence for the reviewer's comments. There are no flaws in the data analysis, interpretation and conclusions. This work is of great significance to this field and related fields. I would suggest to accept it.

Best wishes

Our reply: We acknowledge #R5 evaluation and valuable comments on our manuscript.

Reviewer #6 (Remarks to the Author):

A major issue raised by Reviewer 4 is whether this study provides sufficient evidence of ROS involvement in the reported effects of endogenous formaldehyde. The new EPR findings included by the authors in response do show strong circumstantial evidence of ROS involvement in the FA effects.

Our reply: We thank R#6 for this positive commentary about the use of EPR (or ESR) to address ROS involvement in FA effects. In our last revision, we have approached R#4 issue by incorporating techniques that report ROS through different mechanisms, as well as performed survival (Fig. 4) and complementation assays (Fig. 5) to further validate the requirement of GSH synthesis for cellular FA tolerance. We would really like to acknowledge his/her efforts to comment on the comments raised by the original R#4.

Reviewer #4 correctly questions the validity of the roGFP data on the grounds that electrophilic adduction by FA could conceivably lead to a spectral change in the fluorogenic center that may be mistaken for that which occurs when the vicinal cysteines on the surface of the beta barrel of roGFP are oxidized to form a disulfide.

To the Reviewer's point, roGFP is not a direct sensor of ROS. Rather, roGFP reports on the glutathione redox potential (EGSH) established by the ratio GSH/GSSG in the Nernst equation (it has been demonstrated that EroGFP equals EGSH, See Meyer and Dick, 2010 DOI:10.1089/ars.2009.2948) through a redox relay involving glutathione peroxidase and glutathione reductase.

Our reply: We appreciate the clarification of the roGFP mechanism, and we are now including a more precise description of roGFP mechanism of action. In our manuscript, we have calculated the oxidized fraction of Grx1-roGFP (Fig. 6i) using the equation provided in the Meyer and Dick paper referred by the reviewer. We described this calculation in the methods section (lines 582-585, reference 35 -the original 2008 paper from T. Dick lab (Gutcher et al).

Unfortunately, the iodoacetic acid experiment performed by the authors is not all that informative because the results are not generalizable to other electrophiles including those that can induce dithiol formation. Specifically, FA has been described to form a hemithioacetal with cysteine (<https://doi.org/10.1038/s42004-019-0224-2>), which could condense into a disulfide with a neighboring thiol. Thus, it is possible that FA induces the same disulfide on roGFP that results from oxidation through its redox relay. To validate the use of roGFP with FA as a stimulus, the integrity of this redox relay has to be demonstrated.

Our reply: When we designed the experiment using iodoacetamide (IAA), our intention was to address R#4 point, who stated that "*Normally, roGFP would be an excellent choice – but considering the thiol reactivity of formaldehyde, the question should be how a direct interaction of FA with redox-sensitive thiols can be excluded; in other words: can roGFP fluorescence changes be brought about in an ROS-independent manner through thiol-reactive compounds?*"

In response to this comment from R#4, we selected one thiol-reactive compound (IAA) that reacts in a non-specific way with thiol groups, detecting no induction of roGFP signal (Supplementary Fig. 3a-c). In addition, IAA -which is very toxic -is not more toxic to $\Delta ADH5$ than to WT cells (Supplementary Fig. 3b), which is in stark contrast to FA, highly toxic for $\Delta ADH5$ cells.

The mechanism by which FA reacts with free cysteines described by Kamps et al (<https://doi.org/10.1038/s42004-019-0224-2>) consists of the formation of the hemithioacetal, which last for less than 40 minutes at neutral pHs. The thermodynamic product (more stable) is a thiazolidine, which consists of the cyclization of the thiol group and the amine group in cysteine. In this same study, the authors addressed whether FA could induce the formation of disulfide bonds from the hemithioacetal formed with cysteine in tripeptides such as glutathione. They were not able to detect this phenomena as stated in the text: "*As shown by studies with glutathione and another Cys containing tripeptide, we observed that under the tested conditions thiols do not form disulfides when treated with HCHO (Supplementary Figs. 42 and 43).*" In light of this published result, we consider the formation of a disulfide bond in roGFP catalysed by FA instead of ROS an unlikely event that, if occurring, would perhaps only marginally contribute to roGFP signal detected in presence of FA.

To further support ROS induction upon FA, we have included additional probes that do not rely on cysteine (PBSF -fluoresces upon peroxidation of a sulfonyl bond; H2DCFDA -fluoresces upon oxidation of the derivative dichlorodihydrofluorescein; DHE -oxidized to 2-hydroxyethidium by superoxide and non-specifically to ethidium, and ESR -which detects superoxide ion, peroxy radical, peroxy nitrite and nitrogen dioxide. In the updated manuscript, we are including the following statement to clarify the potential issues of roGFP in the context of FA studies:

"This sensor reports on the GSH redox potential (E_{GSH}) established by the ratio GSH:GSSG through a redox relay involving cysteines in cellular glutaredoxins, as an indirect measurement of ROS"

"In order to test the induction of oxidative stress by FA more thoroughly, we incorporated the genetically-encoded cytosolic sensor roGFP. This sensor reports on the GSH redox potential (E_{GSH}) established by the ratio GSH:GSSG through a redox relay involving cysteines in cellular glutaredoxins, as an indirect measurement of ROS²⁷. Exposure to FA induced a population of cells in which the sensor is oxidized in the absence of ADH5 (Fig. 3d,e). Importantly, iodoacetamide (IAA, a known thiol-reactive compound) was not able to oxidize roGFP when cells were exposed to non-lethal or subtoxic concentrations of this compound, indicating that an electrophilic attack on roGFP is not sufficient to cause a spectral change consistent with its oxidation (Supplementary Fig. 3a-c)."

On a related note, the authors' description of the finding is also confusing. That "...roGFP cannot be activated by any thiol-reactive molecule" is factually incorrect as stated (e.g., diamide, H₂O₂) but also, I suspect, not what the authors intend to say. They may mean to say that an oxidative (electrophilic) attack on roGFP is not sufficient to cause a spectral change consistent with its oxidation. The Spanish "cualquier" does not accurately translate to "any" in this instance, "not just any" may be their intended meaning here.

Our reply: we apologize for this mistake. Indeed, R#6 is right, we mean to say that an oxidative (electrophilic) attack on roGFP is not sufficient to cause a spectral change consistent with its oxidation. We can modify the text accordingly.

The equilibration of GSH/GSSG with roGFP is mediated by its glutathionylation by glutaredoxin. This affords at least two opportunities to determine whether the change in the ratio of fluorescence intensity induced by 405 and 488 excitation is secondary to a FA attack on the sensor.

If ROS are involved, inhibition of glutaredoxin activity (e.g., 2-AAPA) should completely ablate the roGFP signal induced by FA, as it does when H₂O₂ is given to the cells. Secondly, a comparison of the rate of the FA-induced response of roGFP and the chimeric glutaredoxin-roGFP version (which the authors also report using in this study!) would be expected to show a faster (though not necessarily a stronger) response in Grx-roGFP expressing cells.

Our reply: We agree that these experiments could address the integrity of roGFP signalling in the context of FA exposure. However, we consider the alteration of GSH/GSSG ratio by FA is independently supported by UPLC-HRMS data (Fig. 6g). In addition, roGFP induction upon FA is supported by additional probes that do not rely on glutaredoxins or on cysteine-reactive moieties as described above.

Confirmation that the FA-induced roGFP response depends on glutaredoxin would establish that FA treatment induces an increase in the intracellular EGSH, which would support the authors' thesis. However, by itself, it would not prove the involvement of ROS.

Our reply: Independently of roGFP data, we are showing that FA induces an increase in the intracellular EGSH by UPLC-HRMS (EGSH estimated from the reduction in the ratio of the GSH/GSSG) (Fig. 6g). UPLC-HRMS is a high-resolution mass spectrometry technique that allows a direct measurement of cellular GSH and GSSG, from which we have calculated the GSH/GSSG ratio. These data agree with the EGSH increase detected by Grx1-roGFP (Fig. 6i -estimated from the fraction of OxD-Grx1-roGFP).

The logical next step would be to test the role of glutathione peroxidase activity in the roGFP relay.

Supplementing the cells with sodium selenite overnight would likely increase the expression of the selenium-containing Gpxs nonspecifically, which would be expected to alter the kinetics of the intracellular roGFP response to FA treatment relative to that in control unsupplemented cells. A knock down/knockout of Gpx would be convincing as well, but it would hinge on targeting the correct Gpx isoform to distinguish between H₂O₂ and lipid peroxide involvement.

Perhaps a simpler approach at this point would be to test whether the FA effects require H₂O₂. Exogenous catalase and/or endogenous catalase overexpression, with appropriate controls, could be an easy thing to try

Our reply: We acknowledge the experimental advice from R#6 aiming at the integral characterization of roGFP as previously shown in excellent papers from Dr. James Samet lab (Wages et al 2014, Redox Biol for the characterization of zinc exposure (DOI: 10.1016/j.redox.2014.10.005; and doi: [10.1289/ehp.1206039](https://doi.org/10.1289/ehp.1206039), for ozone-exposed epithelial cells), among others. In these works, the authors performed the selenium, catalase and 2-

AAPA exposure experiments to validate the specificity of the signal detected by roGFP sensors. These validations were both appropriate and necessary given that roGFP was the unique approach used to address oxidative stress in those papers. In our manuscript, we are addressing oxidative stress and GSH/GSSG ratio by multiple and alternative chemical probes, and by UPLC-HRMS, overall supporting the findings beyond the use of roGFP.

To distinguish whether roGFP relay induction by FA depends on H₂O₂ or on lipid peroxides (or other ROS) might be truly challenging. As R#6 recognized, the knockdown/out of Gpxs would hinge on targeting the correct Gpx isoform (there are at least 8 isoforms coded in the human genome (GPx1-8)). We appreciate the R#6's suggestion of overexpressing catalase. Indeed, a reduction of roGFP signal in conditions of catalase overexpression would eventually address whether H₂O₂ is involved in these effects. However, the ESR spectroscopy uses the spin probe CMH (1-hydroxy-3-methoxycarbonyl-2,2,5,5-tetramethylpyrrolidine) and detects superoxide ion, peroxy radical, peroxy nitrite and nitrogen dioxide but not H₂O₂ (Fig. 3f) (<https://doi.org/10.1371/journal.pone.0090964>). This result does not mean that H₂O₂ is not involved in FA effects, but it anticipates that overexpressing catalase might not be conclusive because other peroxides accumulate in response to FA (as it would be expected from a general disruption of GSH/GSSG balance). In our manuscript, we have used the word "oxidative stress" instead of specifically indicating H₂O₂ or lipid peroxides, because our data support the accumulation of ROS, but we are not able to specifically identify if this ROS is H₂O₂, a lipid peroxide or other cellular peroxide.

I have to say that it is in our best interest to identify the reactive oxygen specie(s) that accumulate upon FA and that can contribute to FA toxicity when DNA repair is intact. And part of the experiments we are planning include not only the overexpression of catalase, but also SOD, some peroxiredoxins, and proteins that counteract lipid peroxides independently of GSH (FSP1, <https://doi.org/10.1038/s41586-019-1707-0> or DHODH, <https://doi.org/10.1038/s41586-021-03539-7>), among others. Moreover, we are directing these proteins to different cellular organelles that might be relevant to FA toxicity (endoplasmic reticulum, cytoplasm, nucleus, and mitochondria). We hope to advance on these experiments in the upcoming years.